# Benchmarking Uncertainty Disentanglement: Specialized Uncertainties for Specialized Tasks

**Bálint Mucsányi**
University of Tübingen
b.h.mucsanyi@gmail.com

**Michael Kirchhof**
University of Tübingen

**Seong Joon Oh**
University of Tübingen
Tübingen AI Center

## Abstract

Uncertainty quantification, once a singular task, has evolved into a spectrum of tasks, including abstained prediction, out-of-distribution detection, and aleatoric uncertainty quantification. The latest goal is disentanglement: the construction of multiple estimators that are each tailored to one and only one source of uncertainty. This paper presents the first benchmark of uncertainty disentanglement. We reimplement and evaluate a comprehensive range of uncertainty estimators, from Bayesian over evidential to deterministic ones, across a diverse range of uncertainty tasks on ImageNet. We find that, despite recent theoretical endeavors, no existing approach provides pairs of disentangled uncertainty estimators in practice. We further find that specialized uncertainty tasks are harder than predictive uncertainty tasks, where we observe saturating performance. Our results provide both practical advice for which uncertainty estimators to use for which specific task, and reveal opportunities for future research toward task-centric and disentangled uncertainties. All our reimplementations and Weights & Biases logs are available at https://github.com/bmucsanyi/untangle.

## 1 Introduction

When uncertainty quantification methods were first pioneered for deep learning [12, 29], their task was simple: giving one total uncertainty estimate. The recent demand for trustworthy machine learning [36] created new requirements, mostly centering around disentangling the above predictive uncertainty into aleatoric (data-inherent and irreducible) and epistemic (model-centric and reducible) components [11, 52, 47]. Such disentangled estimators are needed for multiple modern applications: Out-of-distribution detection needs to filter unseen samples with high epistemic uncertainty without being confounded with seen samples with high aleatoric uncertainty [37], and active learning uses individual aleatoric and epistemic estimates to select the most efficient samples to learn from [28, 35].

However, recent advances towards such disentangled uncertainties are primarily theoretical and supported by only small-scale experiments [47, 53, 37]. Conversely, larger-scale benchmarks evaluate methods w.r.t. only one uncertainty component and do not test for undesirable side effects on other components [13, 40]. There is currently no study that evaluates which component(s) each method captures in practice and which it does not – which is often contrary to their original intuition.

Our work provides a comprehensive benchmark of the vast recent landscape of uncertainty methods and tasks. We reimplement nineteen uncertainty quantification methods in up to fourteen ways and evaluate each on thirteen practically defined tasks on ImageNet-1k [10] and CIFAR-10 [27]. This includes recent information-theoretical and Bregman decomposition formulas that intend to disentangle total uncertainties into aleatoric and epistemic components [57, 42, 11]. We reveal that none of the existing approaches achieve disentanglement in practice. Most proposed pairs of estimators are highly internally correlated (rank corr. $\geq 0.78$) and fail to unmix aleatoric and epistemic uncertainty (Section 3.1). We also find that specialized tasks (Sections 3.2 and 3.3) are harder to solve

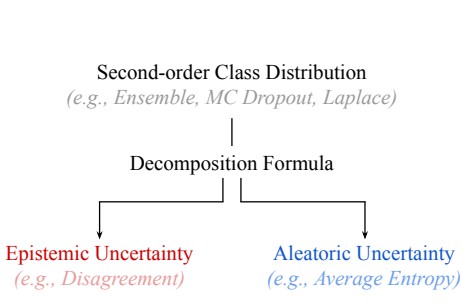
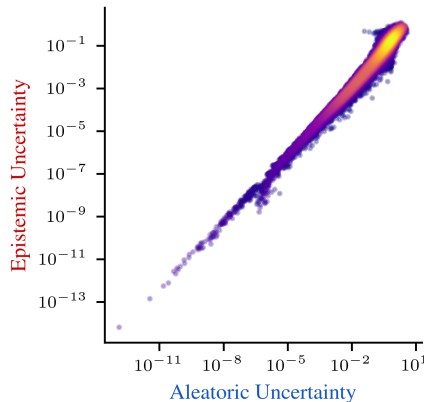

Figure 1: Decomposition formulas like in Eq. (1) decompose second-order distributions into individual estimates for epistemic and aleatoric uncertainties. However, we find that the estimates are internally highly correlated. The density plot on the right shows this for the epistemic and aleatoric uncertainty estimates obtained from decomposing deep ensemble uncertainties on ImageNet-1k. This means that they capture the same notion of uncertainty in practice as opposed to two disentangled ones.

than previous predictive uncertainty tasks, on which we observe saturating performance (Section 3.4). Based on these insights, we uncover a promising path for future disentangled uncertainty estimates: combining individual estimators that strongly reflect one type of uncertainty while being (almost) unrelated to the other.

These findings emphasize the importance of clearly specifying the task one wants to solve with an uncertainty estimator and tailoring the estimator to it. We anticipate that our quantitative insights will drive the field toward developing more disentangled and specialized uncertainty estimators.

## 2    Benchmarked Methods

This section provides an overview of the benchmarked uncertainty estimators and disentanglement formulas. We reimplement all nineteen methods and explain implementation details in Appendix A.

### 2.1    Uncertainty Quantification Methods

We consider a classification setting with a discrete label space of $C$ classes. On top of the eight supervised uncertainty quantification methods from Kirchhof et al. [25], we reimplement another eleven methods to encourage diversity and general applicability of our findings. Below, we categorize the benchmarked approaches into distributional and deterministic methods.

#### 2.1.1    Distributional methods

Distributional methods model a second-order predictive distribution $q(\pi \mid x)$ over class probability vectors $\pi \in \Delta^{C-1}$ for an input $x \in \mathcal{X}$. For example, $q(\pi \mid x)$ can correspond to a Bayesian posterior on the simplex, $p(\pi \mid x, \mathcal{D})$, induced by a weight-space posterior $p(\theta \mid \mathcal{D}) \propto p(\mathcal{D} \mid \theta) \, p(\theta)$ when training on dataset $\mathcal{D}$.

**Spectral-Normalized Gaussian Processes (SNGP)** [32] represent the $q(\pi \mid x)$ distributions by approximating a Gaussian process (GP) over the classifier *output*, aided by spectral normalization. We also benchmark the last-layer **GP** without spectral normalization. The last-layer **Laplace Approximation** [8] and **Stochastic Weight Averaging – Gaussian (SWAG)** [34] both model a Gaussian parameter distribution in a post-hoc fashion that induces the $q(\pi \mid x)$ distributions. The Laplace approximation does so by fitting the parameter-space Gaussian w.r.t. the local curvature around the MAP estimate, whereas SWAG samples model weights via checkpointing and fits an empirical distribution. Similarly, **Heteroscedastic Classifiers (HET)** [5] and **Latent Heteroscedastic Classifiers (HET-XL)** [6] predict a heteroscedastic Gaussian distribution over the *logits* and pre-logit *embeddings*, respectively. Evidential deep learning methods for classification [45, 4] directly learn a

Dirichlet distribution over the output probability vectors. Following Ulmer et al. [50], we refer to the method of Sensoy et al. [45] as **Evidential Deep Learning (EDL)** and that of Charpentier et al. [4] as the **Posterior Network (PostNet)**.

**MC Dropout** [12, 48] and **Deep Ensemble** [29] do not construct second-order predictive distributions $q(\boldsymbol{\pi} \mid \boldsymbol{x})$ explicitly. Instead, they sample from them by $M$ repeated forward passes with randomly switched off activations or by training $M$ models, respectively. The **Heteroscedastic Classification Neural Network (HetClassNN)** [23] uses the uncertainties from MC Dropout for epistemic uncertainty and models an input-conditional heteroscedastic logit variance for aleatoric uncertainty. The **Shallow Ensemble** [31] is a lightweight approximation of the Deep Ensemble with a shared backbone and $M$ output heads.

Practical tasks like threshold-based rejection often need a scalar uncertainty value $u(\boldsymbol{x}) \in \mathbb{R}$ instead of a second-order predictive distribution $q(\boldsymbol{\pi} \mid \boldsymbol{x})$. To this end, **uncertainty aggregators** compile the above distributions into scalar uncertainty estimates. Several methods exist for this aggregation, such as calculating the Bayesian Model Average (BMA) $\bar{\boldsymbol{\pi}}(\boldsymbol{x}) := \mathbb{E}_{q(\boldsymbol{\pi}|\boldsymbol{x})}[\boldsymbol{\pi}]$ and using its entropy as the uncertainty estimate $u(\boldsymbol{x})$ or quantifying the variance of $q(\boldsymbol{\pi} \mid \boldsymbol{x})$, as often seen in ensembles. While many distributional methods are proposed with a specific aggregator, we show in Appendix D.5 that they do not always behave as expected and limit performance. To remove this confounder, we consider fourteen aggregators (Appendix D) for distributional methods and use the best-performing one.

### 2.1.2 Deterministic Methods

Deterministic methods [43] directly output scalar uncertainty estimates $u(\boldsymbol{x}) \in \mathbb{R}$ instead of modeling a second-order predictive distribution $q(\boldsymbol{\pi} \mid \boldsymbol{x})$ over class probability vectors.

**Loss Prediction** [59, 28, 25] employs an additional MLP head for $u(\boldsymbol{x})$ that estimates the loss of the network's prediction $\boldsymbol{\pi}(\boldsymbol{x}) \in \Delta^{C-1}$, reflecting a notion of (in-)correctness. **Correctness Prediction** is a special variant for classification where $u(\boldsymbol{x})$ predicts how likely the predicted class $\hat{y} := \arg\max_{c \in \{1, \ldots, C\}} \pi_c(\boldsymbol{x})$ is to be the correct class $y$, i.e., $p(\hat{y} = y \mid \boldsymbol{x})$.

**Deterministic Uncertainty Quantification (DUQ)** [53] learns a latent mixture-of-RBF density on the training dataset and outputs as $u(\boldsymbol{x})$ how close an input's embedding is to the mixture means. The **Mahalanobis** method [30] builds a similar latent mixture of Gaussians in a post-hoc fashion. It also perturbs the inputs adversarially to separate in-distribution (ID) and out-of-distribution (OOD) samples. The **Deep Deterministic Uncertainty (DDU)** method [37] combines the spectral normalization of SNGPs with the latent density of the Mahalanobis method. **Temperature Scaling** [18] post-hoc calibrates the predicted probability vectors with a temperature scalar.[1] As a **Baseline**, we use a deterministic single-point network trained with the cross-entropy loss.

### 2.2 Uncertainty Decomposition Formulas

So far, we only considered uncertainty estimators that (sometimes after aggregating) output a single estimate $u(\boldsymbol{x})$. A second strain of literature outputs not only one estimate but decomposes the $q(\boldsymbol{\pi} \mid \boldsymbol{x})$ of distributional methods into multiple estimators that each intend to quantify one source of uncertainty, such as epistemic and aleatoric uncertainty [22, 36]. We benchmark two prominent approaches to obtain such pairs of estimators: the **information-theoretical (IT)** [11, 37, 57] and the **Bregman** decomposition [42, 19, 17]. In the main paper, we focus on the IT decomposition due to its widespread use. The definition and results of the Bregman decomposition are shown in Appendix B.

The IT decomposition decomposes the entropy of the predictive distribution $p(y \mid \boldsymbol{x}) = \int p(y \mid \boldsymbol{\pi}, \boldsymbol{x}) \, \mathrm{d}q(\boldsymbol{\pi} \mid \boldsymbol{x})$ into an aleatoric and an epistemic component:

$$\underbrace{\mathbb{H}_{p(y|\boldsymbol{x})}(y)}_{\text{predictive}} = \underbrace{\mathbb{E}_{q(\boldsymbol{\pi}|\boldsymbol{x})}\left[\mathbb{H}_{p(y|\boldsymbol{\pi},\boldsymbol{x})}(y)\right]}_{\text{aleatoric}} + \underbrace{\mathbb{I}_{p(y,\boldsymbol{\pi}|\boldsymbol{x})}(y;\boldsymbol{\pi})}_{\text{epistemic}}, \tag{1}$$

where $p(y \mid \boldsymbol{\pi}, \boldsymbol{x}) = \mathrm{Cat}(y; \boldsymbol{\pi}) = \pi_y$, $p(y, \boldsymbol{\pi} \mid \boldsymbol{x}) = p(y \mid \boldsymbol{\pi}, \boldsymbol{x})q(\boldsymbol{\pi} \mid \boldsymbol{x})$, $\mathbb{H}_{p(y|\boldsymbol{x})}(y)$ is the entropy, and $\mathbb{I}_{p(y,\boldsymbol{\pi}|\boldsymbol{x})}(y; \boldsymbol{\pi})$ is the mutual information. Intuitively, the aleatoric component represents the

---

[1]The DDU, temperature scaling, Laplace, and Mahalanobis methods are the only ones in our benchmark that require a validation set for training their uncertainty modules.

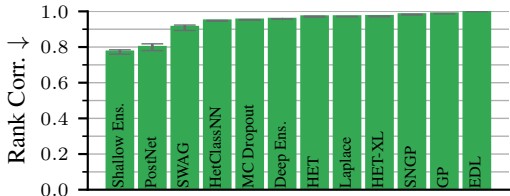 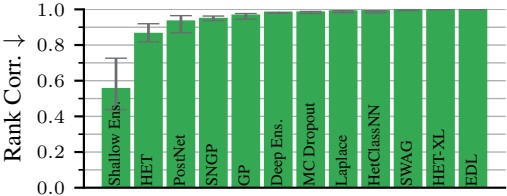

(a) ImageNet results. All twelve distributional methods exhibit a high rank corr. ($\geq 0.78$).

(b) CIFAR-10 results. Eleven out of twelve distributional methods exhibit a strong rank corr. ($\geq 0.88$).

Figure 2: Rank correlation between the aleatoric and epistemic estimates obtained by the IT decomposition on ImageNet (left) and CIFAR-10 (right). The two uncertainty components are strongly correlated for most methods, violating a necessary condition of their disentanglement.

spread of the labels that the plausible predictions in the posterior have on average. In contrast, the epistemic component only captures the disagreement of the predictions $p(y \mid \boldsymbol{\pi}, \boldsymbol{x})$ in the second-order predictive distribution $q(\boldsymbol{\pi} \mid \boldsymbol{x})$. For evidential deep learning methods with Dirichlet $q(\boldsymbol{\pi} \mid \boldsymbol{x})$ distributions, closed-form expressions exist for each term of Eq. (1), whereas other approaches require Monte Carlo approximations [50].

The key goal behind these decompositions is **uncertainty disentanglement**: The aleatoric component should capture aleatoric and only aleatoric uncertainty, and the epistemic estimator should reflect epistemic and only epistemic uncertainty. In particular, this entails that both components need to be sufficiently uncorrelated. See Appendix E for more details and a formal definition.

## 3 Experiments

We now investigate our main research question: Does any approach give disentangled uncertainty estimators (Section 3.1)? Then, we go into each individual type of uncertainty and investigate which estimator practically performs the best on epistemic (Section 3.2), aleatoric (Section 3.3), and predictive uncertainty tasks (Section 3.4). Lastly, we draw conclusions across all tasks (Section 3.5) and benchmark the robustness of current uncertainty estimators (Section 3.6).

To provide even grounds, we reimplement each method and provide it as an easy-to-use uncertainty wrapper that can be added to arbitrary `timm` [56] models[2]. In this paper, we use pretrained ResNet-50 backbones and train each approach for 50 ImageNet-1k [10] epochs with a training pipeline following Tran et al. [49]. The CE baseline converges to an accuracy of 0.785 with this strategy. Since the DUQ method has memory and stability issues on ImageNet, in Section 3.7, we repeat all experiments on CIFAR-10 [27] with the WideResNet 28-10 architecture, following Liu et al. [32]. We only report the other 18 methods on ImageNet. We search for ideal hyperparameters and an early stopping checkpoint for each method by tracking the validation performance. We then run the best hyperparameters across five seeds and report mean, minimum, and maximum test performance. This overall takes 1.5 GPU years on RTX 2080 Ti GPUs. We report the main results in the paper and go into more detail for, e.g., different uncertainty aggregators in the appendix. We also publish all of these metrics and their logs.[3]

### 3.1 Decomposition Formulas Fail to Disentangle Aleatoric and Epistemic Uncertainty

We first study if decomposition formulas, IT or Bregman, yield disentangled estimators. Since they decompose second-order predictive distributions $q(\boldsymbol{\pi} \mid \boldsymbol{x})$, we analyze distributional methods and no deterministic methods in this section.

Fig. 1 reveals a simple failure: The decomposed aleatoric and epistemic uncertainty estimates are strongly correlated, being high or low iff the other component is high or low. These severe internal correlations prohibit the estimators from capturing semantically different sources of uncertainty and hinder applications that require unconfounded uncertainty estimates, such as active learning.

---

[2]https://github.com/bmucsanyi/untangle
[3]https://wandb.ai/bmucsanyi/untangle

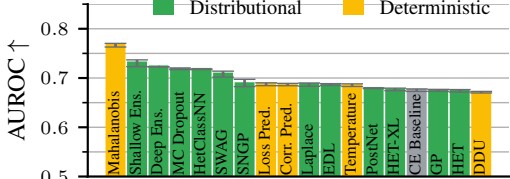

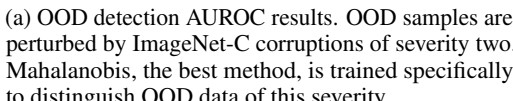

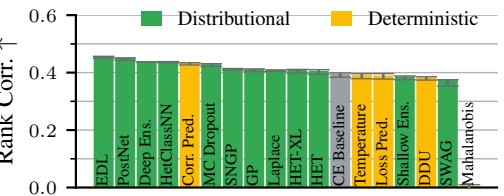

(a) OOD detection AUROC results. OOD samples are perturbed by ImageNet-C corruptions of severity two. Mahalanobis, the best method, is trained specifically to distinguish OOD data of this severity.

(b) Rank correlation of uncertainty estimators and the GT aleatoric uncertainty on ImageNet. The entropy of the ImageNet-ReaL label distributions is used as GT aleatoric uncertainty.

Figure 3: Performance of uncertainty quantification methods on epistemic (left) and aleatoric (right) uncertainty tasks on the ImageNet validation dataset.

The behavior of deep ensembles is just one example. Figure 2a shows that aleatoric and epistemic estimates obtained via the IT decomposition are highly rank correlated (rank corr. $\in [0.78, 0.99]$) for all distributional methods that we benchmark. This holds similarly on CIFAR-10 (Figure 2b), as well as for the Bregman decomposition (Appendix B) and also does not considerably lower when we artificially add more epistemic uncertainty into the dataset, see Appendix C.2. Often, the components are even linearly correlated; see the Pearson correlation results in Appendix C.4.

A part of these correlations is inevitable: On ImageNet, regions with aleatorically uncertain images are undersampled compared to regions without aleatoric uncertainty and thus also more epistemically uncertain (see Fig. J.3). This means that ImageNet has a level of inevitable correlation between epistemic and aleatoric uncertainty estimates. We quantify this inevitable correlation via the rank correlation between the GT aleatoric uncertainty (i.e., the entropy of the GT label distribution) and the models' epistemic uncertainty given by the Bregman decomposition in Appendix B.4. This gives levels of inevitable correlation for the Bregman decomposition that are at most $0.45$. Further, we show in Section 3.3 that there are pairs of uncertainty estimators where one performs well on aleatoric and the other on epistemic uncertainty, with a notably low rank correlation of $0.15 \pm 0.01$. Thus, the severe correlations exceeding $0.78$ are shortcomings of the decomposition formulas and not inherent properties of the ImageNet dataset.

In conclusion, decomposition formulas of various forms applied to various second-order distributions produce uncertainty estimators that are so highly correlated that they hardly capture the different individual notions of aleatoric and epistemic uncertainty that they are intended to capture.

## 3.2 Epistemic Uncertainty: Specialized Uncertainty Esimators Detect OOD Inputs the Best

If decomposition formulas cannot yield epistemic and aleatoric uncertainty estimates, which methods can? For the rest of the paper, we widen the scope and include not only the aleatoric and epistemic estimators defined by the decomposition formulas but also arbitrary aggregators of second-order distributions, as well as deterministic methods. This section tests which of these estimators represents epistemic uncertainty, measured by an out-of-distribution (OOD) detection task [17, 37]. We create a 50/50 dataset of in-distribution (ID) and OOD samples, with ID samples getting class 0 and OOD samples getting class 1. We quantify via a binary classification AUROC if uncertainty estimates are higher on OOD samples than on ID samples. We use ImageNet-C [21] with all of its corruptions of severity level two as OOD data. The severity level two is far enough out-of-distribution to deteriorate the ImageNet accuracy by 27% (Section 3.6).

Fig. 3a shows that the methods differ greatly in their ability to detect OOD samples and, thus, in their alignment with epistemic uncertainty. The Mahalanobis method performs best. This is likely because it is the only method trained specifically for OOD detection with ImageNet-C corruptions of severity level two. We find its advantage already vanishes when changing the task to severity level three (Appendix H.3). A method with a similar latent density intuition and distance-awareness induced by spectral normalization, DDU, is the worst at telling ID and OOD samples apart (AUROC $= 0.675$). Additionally, the best-performing aggregators for the second-order distributions, the performance of which Fig. 3a shows, are often not the disagreement-based aggregators that decomposition formulas propose for epistemic uncertainty tasks (Appendix D.5). These insights highlight that the practical

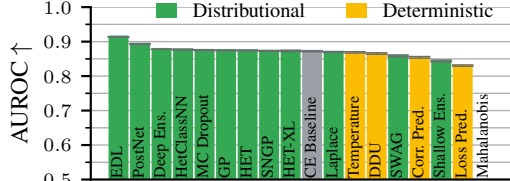

(a) ID correctness prediction results measured by the AUROC w.r.t. model correctness. The evidential deep learning methods, EDL and PostNet, capture predictive uncertainty remarkably well.

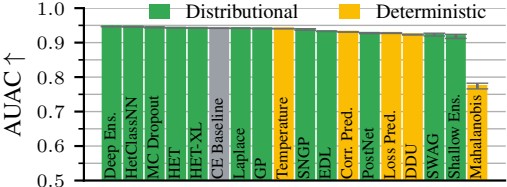

(b) Abstained prediction results using the AUAC metric. Most methods are within a 0.03 AUAC band. EDL and PostNet lose their advantage as their accuracy is lower.

Figure 4: ID predictive uncertainty evaluation on the ImageNet validation dataset. The Mahalanobis method is a specialized OOD detector that cannot differentiate between ID samples.

tailoring of uncertainties to a specific uncertainty task of interest, as done by the Mahalanobis method, weighs more than high-level intuitions, which, e.g., decomposition formulas are based on.

### 3.3 Aleatoric Uncertainty: No Method With Outstanding Performance

The previous experiment isolated the epistemic capabilities of uncertainty estimates. We now evaluate how well the benchmarked models predict aleatoric uncertainty. We follow Tran et al. [49] and Kirchhof et al. [24, 25] and use the disagreement of human annotators as ground truths for aleatoric uncertainty. ImageNet-ReaL [2] (and CIFAR-10H [41]) queries multiple annotators for labels on each image. We showcase some examples in Appendix J. We use the entropy of the soft-label distributions per image as GT aleatoric uncertainties. We then calculate the rank correlation between the methods' uncertainty estimates and the GT label entropies across all images. We do not use an AUROC here because the GT values are continuous, but provide binarized AUROCs for direct comparability of aleatoric and epistemic uncertainty performance results in Appendix H.4.

Fig. 3b shows that almost all methods lie within a correlation of $[0, 37, 0.46]$. Note that the best achievable rank correlation is not one since the GT aleatoric data contains ties. While it is unknown how high the best achievable rank correlation is, the fact that there are consistent improvements across the methods hints at the fact that further performance gains are far from saturated. The method that sticks out on the low end of the spectrum is Mahalanobis, which is uncorrelated with the GT aleatoric uncertainty. This is, in fact, a strength: Mahalanobis estimates reflect epistemic uncertainty while being non-informative of aleatoric uncertainty. Combining this with a second estimator for aleatoric uncertainty can pave the way for a pair of disentangled uncertainty estimators. As a simple start, combining it with the CE baseline achieves a low rank correlation of $0.15 \pm 0.01$ between the two. We see this as a promising pathway to disentangled uncertainty estimators in the future.

With the aleatoric and epistemic tasks introduced, we can take a final look at the epistemic and aleatoric estimators proposed by the IT decomposition. In Appendix C.3, using them instead of the best ones reduces Shallow Ensemble's performance, which was the best distributional method on OOD detection, to the CE baseline level. It again shows that the theoretically intuitive estimators underperform in practice.

### 3.4 Predictive Uncertainty: The Best Method Depends on the Precise Task

Let us now broaden the view beyond disentanglement to benchmark how well uncertainty estimators solve other practically relevant tasks. We start with correctness prediction, where the AUROC quantifies whether wrong predictions generally have higher uncertainties than correct predictions.

Fig. 4a shows that most uncertainty estimators perform within $\pm 0.014$ of the cross-entropy baseline when predicting correctness. Modern methods like HET-XL do not outperform older methods like the Deep Ensemble or MC Dropout. Evidential deep learning methods, like EDL and PostNet, are an exception to this. They are considerably better at predicting correctness. This also holds when mixing the datasets with OOD data in Appendix H.1. However, their better performance comes at a cost. Evidential methods have a trade-off between the quality of their uncertainty estimates and the

classification accuracy. When demanding similar classification accuracies, we find that they lose their advantage.

A related task to correctness prediction is abstained prediction. It involves refusing to predict on the $x\%$ most uncertain examples and calculating the model's accuracy on the remaining samples. We use the Accuracy-Coverage (AC) curve [15] that plots increasing fractions of abstained samples from $0\%$ to $100\%$ on the $x$-axis against the accuracy on the non-abstained portion. Following the conventions of Galil et al. [14], we denote this metric as the area under the accuracy coverage curve (AUAC). In Appendix H.5, we also evaluate methods on the rAULC and E-AURC metrics that normalize the AUAC by the accuracy of the underlying model [43, 14].

Fig. 4b shows that this predictive uncertainty task is saturated. All uncertainty methods apart from Mahalanobis obtain an AUAC score greater than $0.91$, and only a few outperform the CE baseline. Since the AUAC depends on accuracy, EDL and PostNet perform worse in this metric, although both AUROC and AUAC target predictive uncertainty. This demonstrates that practitioners need to carefully specify an uncertainty estimator's overall goal. Designing a system that can detect errors is not the same as designing a system that reduces errors.

The same holds for calibration. While this is also a predictive uncertainty task, the goal is slightly different. It is not to provide a good *ranking* of uncertain images but to give correctness *probabilities* that are close to the true frequentist probabilities.

Fig. 5 shows that different methods excel at this task than at the AUROC correctness task (Fig. 4a). In particular, Laplace and temperature scaling, which were only as good as the CE baseline in terms of AUROC, show drastically improved performance in terms of the ECE. Note that these two use a validation dataset to become better calibrated, similar to how Mahalanobis specialized and outperformed on the epistemic uncertainty task.

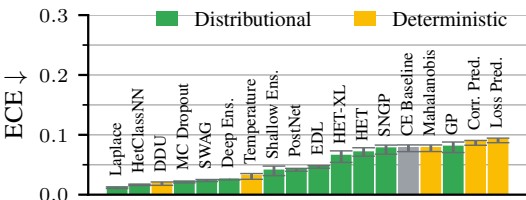

Figure 5: Expected calibration error on ImageNet.

In conclusion, the predictive uncertainty results show that the exact definition of the task one intends to solve with uncertainty estimators matters because different estimators specialize in different notions of uncertainty.

## 3.5 Different Tasks Require Different Methods

The previous sections suggest that uncertainty tasks are not all solved by the same best method. In this section, we investigate how the performance of methods on different tasks is correlated. In particular, we use the previous practical tasks along with further popular metrics and measure the between-task Pearson correlations of the performance of all benchmarked methods. The correlations of the rankings of the methods are similar; see Appendix H.7.

Fig. 6 shows barely any recognizable clusters, except that AUAC is confounded by accuracy. While all other metrics, except OOD AUROC, correlate to some extent, their correlation is not perfect, once again demonstrating that the performances of the uncertainty estimators depend on the exact task. These findings further corroborate that there is no one-fits-all uncertainty estimator, but there are multiple tasks with nuanced differences to which an uncertainty estimator can be tailored.

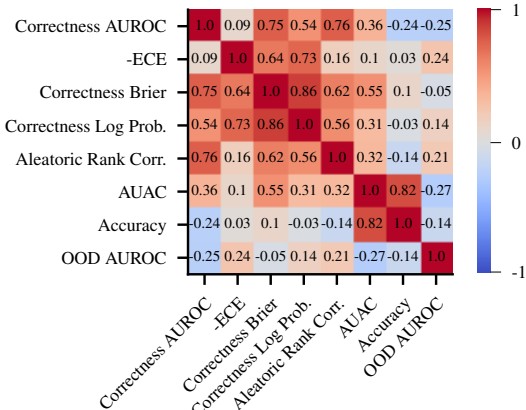

Figure 6: The Pearson correlation of metric pairs across methods and aggregators on the ImageNet validation dataset is only medium. Most capture different aspects of uncertainty methods.

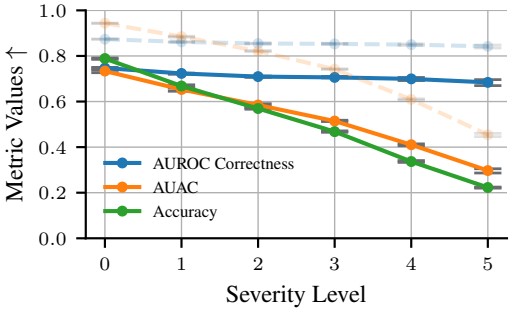 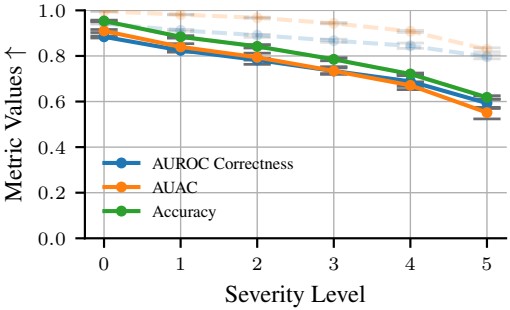

(a) ImageNet results. The uncertainty estimators' performance in terms of the AUROC degrades much slower than the model's accuracy.

(b) CIFAR-10 results. Model accuracy and the performance of the uncertainty method degrade together: OOD, uncertainty estimates are not to be trusted.

Figure 7: Degradation of correctness prediction, abstained prediction, and accuracy metrics with increasingly severe ImageNet-C (left) and CIFAR-10C (right) corruptions. The shown MC Dropout results are typical for all methods (except Mahalanobis). Solid lines: metrics normalized to the $[0, 1]$ range w.r.t. corresponding random and oracle predictors. Dashed lines: unnormalized values.

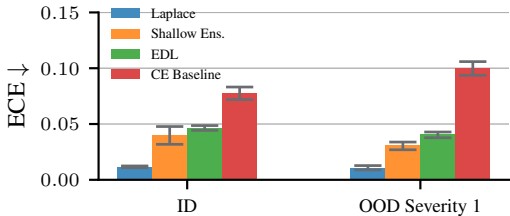 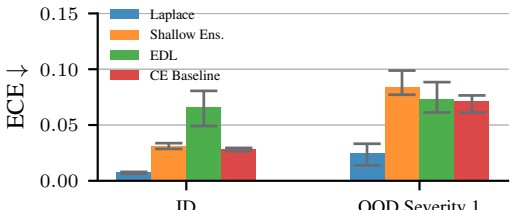

(a) ImageNet results. Methods are generally more robust to OOD perturbations than on CIFAR-10. EDL and Shallow Ensemble even become better calibrated OOD.

(b) CIFAR-10 results. Methods are much less robust to OOD perturbations than on ImageNet: Shallow Ensemble and CE Baseline degrade significantly.

Figure 8: ECE results on ImageNet (left) and CIFAR-10 (right). Methods display drastically different behavior on ImageNet and CIFAR-10 regarding the robustness of their calibration. OOD samples are perturbed with ImageNet-C (left) and CIFAR-10C (right) corruptions of severity level one.

## 3.6 Uncertainties are Robust to Distribution Shifts

As uncertainty estimates are often intended to increase the reliability of systems, one necessity is that they remain robust when a system faces unforeseen inputs. We test this by checking if their previous abstinence and correctness performances are preserved even when the model's accuracy drops with increasing ImageNet-C perturbation levels. Only then can we trust them and, e.g., base the abstinence from prediction on these uncertainty estimates.

Fig. 7a shows the correctness prediction AUROC, AUAC, and model accuracy as we increasingly perturb the ImageNet validation samples and go OOD. The correctness prediction performance in terms of the AUROC remains almost constant, whereas the accuracy degrades to less than $25\%$ at severity level five. The AUAC performance degrades together with accuracy, which is a fundamental property of the metric itself since the area under the accuracy is lower-bounded by the baseline accuracy. The AUAC gain (i.e., AUAC − Accuracy) increases with the perturbation severity, showing that the uncertainty estimators even become relatively *better* on the abstinence task as the severity increases. The tendencies are maintained when we normalize the metrics (solid lines) according to their random predictive performance (see Appendix H.2 for details). This observation holds for all methods except Mahalanobis, see Appendix H.2. Figure 8a shows that the methods' ECE also remains robust to perturbations on ImageNet. These results underline the trustworthiness of existing uncertainty quantification methods as we go OOD on ImageNet.

### 3.7 CIFAR-10 Results Do Not Always Transfer to ImageNet

We conclude our experiments with a word of caution. Appendix G repeats all experiments on CIFAR-10, which is widely used in the uncertainty quantification literature [53, 37, 17]. While some conclusions from CIFAR-10 experiments replicate on ImageNet, like the correlated aleatoric and epistemic estimators, the larger-scale ImageNet often shows different behavior.

**Robustness.** Uncertainty estimates are far less robust on CIFAR-10 than on the ImageNet scale, even though the drop in classification accuracy is very similar. Unlike on ImageNet, where the uncertainty estimators maintain a close to constant performance in predicting correctness as we go OOD (Fig. 7a), on CIFAR-10, correctness estimators deteriorate together with the model's accuracy (Fig. 7b). The same holds for the ECE (Fig. 8a vs. Fig. 8b). So, while robustness appears to be a striking problem on CIFAR-10, it gets resolved by scaling to a larger dataset.

**Method rankings.** Nine out of thirteen tasks exhibit substantially different rankings (rank corr. $< 0.5$) between CIFAR-10 and ImageNet. See Table H.1 for details. This indicates that performance on CIFAR-10 should not be taken as an estimate for ImageNet performance.

These experiments underline that methods might show substantially different behaviors on large-scale datasets. As best practice, we encourage to first scale the approaches to the final deployment domain (and define a precise task) instead of making fundamental design choices on toy datasets.

## 4  Connections Between Our Findings and Related Works

**Uncertainty Disentanglement.** The decomposition of aleatoric and epistemic uncertainties [42, 11] has recently been shown to have failure cases. The disentanglement is usually analyzed theoretically [57, 1, 16] or with qualitative plots (Kirchhof et al. [26], Fig. 6-9; Mukhoti et al. [37], Fig. 2; Valdenegro-Toro and Mori [52], Fig. 8-10). Our results support this discussion with a practical and quantitative perspective. To the best of our knowledge, we are the first to quantify the uncertainty disentanglement. We find that no tested decomposition formula works for any tested second-order distribution, neither on ImageNet-1k nor CIFAR-10. Our findings encourage combining separate methods instead, such as the CE baseline's predictive entropy and the Mahalanobis values, where each method handles a specific type of uncertainty. This is similar to the recent work of Mukhoti et al. [37]. We expect that our quantitative benchmarking methods help develop this field further.

**Robustness.** Recent benchmarks on OOD detection and robustness [38, 40, 43, 13] have first highlighted robustness issues of uncertainty estimates. Our benchmark supports these findings on CIFAR-10, especially in the region that is slightly OOD yet already causes degradation of both the main task and the uncertainty estimator. The latter implies that uncertainty estimators either need to become more robust to distribution shifts [25] or be better able to detect subtle epistemic uncertainties. However, our experiments on ImageNet do not show robustness issues. It is possible that the vast space of natural images that the ImageNet training dataset covers resolves this issue. We encourage repeating our experiments and testing the uncertainty estimation not just on test data but also on perturbed test data for future large-scale uncertainty estimators.

**Aleatoric uncertainty.** While epistemic uncertainty is widely evaluated on the OOD detection proxy task [37, 49, 17], aleatoric uncertainty still lacks a standardized testing protocol. The current approaches seem to converge to soft labels, but nuances in how they are collected still need discussion (compare, for example, CIFAR-10H [41] to CIFAR-10S [7] and CIFAR-10N [55]). An increasing number of uncertainty quantification approaches compare to such human GT notions of aleatoric uncertainty [49, 24, 25, 26], indicating the interest in the field. Our benchmark shows that no method can give highly accurate aleatoric uncertainty estimates yet, stressing the need for benchmarks, methods, and training resources to develop along.

**Predictive uncertainty and calibration.** Contrary to aleatoric uncertainty alignment, calibration and predictive uncertainty benchmarks are starting to become saturated and, according to our experiments, the top performers are ready for deployment. This corroborates recent findings by Galil et al. [14]. In comparison to this benchmark that compared model architectures, we compared nineteen different approaches on the same backbone with a wide range of aggregator functions.

# 5    Conclusions, Limitations, and Outlooks

We study how a diverse spectrum of uncertainty estimators and decomposition formulas perform on a comprehensive set of uncertainty quantification tasks. Our quantitative findings bring an empirical foundation to recent discussions in the field, namely that 1) the aleatoric and epistemic uncertainty components of decomposition formulas are highly correlated and not disentangled, 2) epistemic and aleatoric tasks are best solved by practically tailored methods, whereas methods relying on intuitions often underperform, and 3) there is no one-fits-all uncertainty estimate. On a brighter side, our experiments also reveal the important fact for practitioners that 4) predictive uncertainty estimation achieve a high, saturating performance across almost all methods, and 5) uncertainty estimates, when trained on large amounts of data, stay robust to perturbations longer than the classifiers whose uncertainties they predict, hence enabling to safeguard the classifiers to some extent.

A limitation of our disentanglement benchmark is that we tested on two datasets, which are both classification tasks. This is because we require ground truths for aleatoric uncertainty. Currently, the only larger-scale datasets with such ground truths, in the form of multiple annotations per input, are the two classification datasets we base our analysis on [2, 41]. Further aleatoric uncertainty ground truths are an ongoing effort [44, 7]. We encourage the expansion of the set of datasets, both within classification and to fields like regression [51] or unsupervised learning [24], to expand our uncertainty disentanglement investigations. A second limitation is that we focus on models that have converged after training on the large-scale ImageNet dataset. A different interesting setup is models trained on small amounts of data, where epistemic uncertainty may be further from convergence. For example, there is a follow-up investigation of our work by de Jong et al. [9] that undersamples the train data. We replicate parts of their main experiment results on CIFAR-10 in Appendix H.9. We encourage future works to evaluate uncertainties on an as broad array of tasks as possible to refine the understanding of which specific uncertainty tasks individual uncertainty estimators excel at.

This last suggestion is a corollary of how our findings changed our perspective on uncertainty quantification. There is no general uncertainty; instead, uncertainty quantification covers a spectrum of tasks where the definition of the exact task heavily influences the optimal method and performance. Such a precise definition of tasks per estimator would help construct disentangled uncertainties and could lead to the alignment of theoretical developments and intuitive descriptions about what particular types of uncertainty methods aim to capture. This pragmatic reassessment of the field could overcome the traditional one-fits-all view of uncertainty and even the more recent epistemic vs. aleatoric dichotomy and uncover the full variety of uncertainty estimates that are tailored to nuanced, practical tasks.

## Acknowledgements

This work was funded by the Deutsche Forschungsgemeinschaft (DFG, German Research Foundation) under Germany's Excellence Strategy – EXC number 2064/1 – Project number 390727645. It also received funding from the DFG via the Priority Programme DFG SPP 2298-2. The authors thank the International Max Planck Research School for Intelligent Systems (IMPRS-IS) for supporting Bálint Mucsányi and Michael Kirchhof.

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

# A  Details of Benchmarked Methods

We consider a classification setting with discrete label space $\{1, \ldots, C\}$ of $C$ classes.

We evaluate two classes of methods: deterministic methods and distributional methods. Distributional methods output a second-order predictive distribution $q(\boldsymbol{\pi} \mid \boldsymbol{x})$ for input $\boldsymbol{x} \in \mathcal{X}$. Deterministic methods output a single probability vector $\boldsymbol{\pi}(\boldsymbol{x}) \in \Delta^{C-1}$ and additional uncertainty estimates detailed below. The (pre-softmax) logits of the models are denoted by $\boldsymbol{f}(\boldsymbol{x}) \in \mathbb{R}^C$. Therefore, it holds that $\boldsymbol{\pi}(\boldsymbol{x}) = \mathbf{softmax}(\boldsymbol{f}(\boldsymbol{x}))$. The activations of layer $\ell \in \{1, \ldots, L\}$ in the model is denoted by $\boldsymbol{f}^\ell(\boldsymbol{x}) \in \mathbb{R}^{D^\ell}$ with $\boldsymbol{f}^L(\boldsymbol{x}) = \boldsymbol{f}(\boldsymbol{x}), D^\ell = C$. The one_hot function converts a label $y \in \{1, \ldots, C\}$ into a vector with only zero entries except for the $y$th one, which is one.

## A.1  Deterministic Methods

Deterministic methods output an uncertainty estimate $u(\boldsymbol{x})$ for input $\boldsymbol{x} \in \mathcal{X}$, such as the estimated probability of the model's prediction to be correct.

### A.1.1  Loss Prediction

Loss prediction [51, 28, 25] employs an additional output head $u^{\mathrm{lp}}$ connected to the pre-logit layer that predicts the loss of the network's prediction on input $\boldsymbol{x} \in \mathcal{X}$. The loss predictor head is trained in a supervised fashion by making $u^{\mathrm{lp}}(\boldsymbol{x})$, the predicted loss, closer to the actual loss $\ell(\boldsymbol{\pi}(\boldsymbol{x}), y) = -\log \pi_y(\boldsymbol{x})$. Precisely, we use the objective

$$\mathcal{L}^{\mathrm{lp}} = -\frac{1}{n} \sum_{i=1}^{n} \log \pi_{y^{(i)}}\left(\boldsymbol{x}^{(i)}\right) + \lambda \left(u^{\mathrm{lp}}\left(\boldsymbol{x}^{(i)}\right) + \log \pi_{y^{(i)}}\left(\boldsymbol{x}^{(i)}\right)\right)^2, \tag{2}$$

where the risk predictor loss (squared Euclidean distance) is traded off with the label predictor loss (cross-entropy) with a hyperparameter $\lambda \in \mathbb{R}_+$.

Note that $Y \mid X = x$ is a random variable in the presence of aleatoric uncertainty. In expectation, Eq. (2) encourages $u^{\mathrm{lp}}(\boldsymbol{x})$ to approximate the true *pointwise risk* $\mathcal{R}(\boldsymbol{\pi}, \boldsymbol{x}) = \mathbb{E}_{p_{\mathrm{data}}(y \mid \boldsymbol{x})}[\ell(\boldsymbol{\pi}(\boldsymbol{x}), y)]$ at each input $\boldsymbol{x} \in \mathcal{X}$.

### A.1.2  Correctness Prediction

Correctness prediction is a variant of risk prediction that, instead of aiming to predict the risk of the network on input $\boldsymbol{x} \in \mathcal{X}$, estimates the true probability of correctness $p_{\mathrm{data}}\left(\arg\max_{c \in \{1,\ldots,C\}} f_c(\boldsymbol{x}) = y \mid \boldsymbol{x}\right)$ on input $\boldsymbol{x} \in \mathcal{X}$. This is achieved by using a sigmoid correctness predictor head $h$ and using the objective

$$\mathcal{L}^{\mathrm{cp}} = -\frac{1}{n} \sum_{i=1}^{n} \log \pi_{y^{(i)}}\left(\boldsymbol{x}^{(i)}\right) - \lambda \left(l_i \log h\left(\boldsymbol{x}^{(i)}\right) + (1 - l_i) \log \left(1 - h\left(\boldsymbol{x}^{(i)}\right)\right)\right), \tag{3}$$

where $l = \left[\arg\max_{c \in \{1,\ldots,C\}} f_c(\boldsymbol{x}) = y\right] \ \forall i \in \{1, \ldots, n\}$, and the correctness predictor loss (binary cross-entropy) is traded off with the label predictor loss (cross-entropy) with a hyperparameter $\lambda \in \mathbb{R}_+$. The uncertainty estimate is $u^{\mathrm{cp}}(\boldsymbol{x}) = 1 - h(\boldsymbol{x})$ (i.e., the probability of making an error).

### A.1.3  Deterministic Uncertainty Quantification

The deterministic uncertainty quantification (DUQ) method of Van Amersfoort et al. [53] learns a latent mixture-of-RBF density on the training set with a strictly proper scoring rule to capture the uncertainty in the prediction based on the Euclidean distance of the input's embedding to the mixture means. The training objective is

$$\mathcal{L}^{\mathrm{duq}} = -\frac{1}{n} \sum_{i=1}^{n} \sum_{c=1}^{C} \mathbf{one\_hot}\left(y^{(i)}\right)_c \log K^c\left(\boldsymbol{x}^{(i)}\right) + \left(1 - \mathbf{one\_hot}\left(y^{(i)}\right)_c\right) \log \left(1 - K^c\left(\boldsymbol{x}^{(i)}\right)\right), \tag{4}$$

where $K^c(\boldsymbol{x}) = \exp\left(-\frac{1}{2\gamma}\|\boldsymbol{f}(\boldsymbol{x}) - \boldsymbol{m}^c\|\right)$ is the RBF value corresponding to class $c \in \{1, \ldots, C\}$ identified by its mean vector $\boldsymbol{m}^c$ in the latent space. To facilitate minibatch training, Van Amersfoort et al. [53] employ an exponential moving average (EMA) to learn the mean vector using the following update rules:

$$n^c \leftarrow \gamma \cdot n^c + (1 - \gamma)|\mathcal{B}_c| \tag{5}$$

$$\boldsymbol{M}^c \leftarrow \gamma \cdot \boldsymbol{M}^c + (1 - \gamma) \sum_{(\boldsymbol{x},y)\in\mathcal{B}_c} \boldsymbol{W}^c \boldsymbol{f}(\boldsymbol{x}) \tag{6}$$

$$\boldsymbol{m}^c \leftarrow \frac{\boldsymbol{M}^c}{n^c}, \tag{7}$$

where $\mathcal{B}$ is a minibatch of samples and $\mathcal{B}_c = \{(\boldsymbol{x}, y) \in \mathcal{B} \mid y = c\}\ \forall c \in \{1, \ldots, C\}$. $\gamma$ is the EMA parameter, and $\boldsymbol{W}^c$ characterizes a linear mapping of the logits for each class.

To regularize the latent density and prevent feature collapse, Van Amersfoort et al. [53] use the following gradient penalty added to $\nabla_{\boldsymbol{\theta}}\mathcal{L}$:

$$\lambda \cdot \left(\left\|\nabla_{\boldsymbol{x}} \sum_{c=1}^{C} K^c(\boldsymbol{x})\right\|_2^2 - 1\right)^2 \tag{8}$$

Each RBF component in the latent space corresponds to one class. The confidence output of the method is the maximal RBF value of the input over all classes. Therefore, the *uncertainty* estimate can be calculated as $u^{\text{duq}}(\boldsymbol{x}) = 1 - \max_{c\in\{1,\ldots,C\}} K^c(\boldsymbol{x})$.

The predicted class of the trained network is $\arg\max_{c\in\{1,\ldots,C\}} K^c(\boldsymbol{x})$.

### A.1.4 Mahalanobis

The Mahalanobis method [30] builds a post-hoc latent density for the training set in the latent space by calculating per-class means and covariances, and using the induced mixture-of-Gaussians as the latent density estimate. Such latent densities are estimated in multiple layers of the network. One layer's confidence estimate is the maximal Mahalanobis score (Gaussian log-likelihood) $K^\ell(\boldsymbol{x})$ over all classes:

$$K^{\ell,c}(\boldsymbol{x}) = -\left(\boldsymbol{f}^\ell(\boldsymbol{x}) - \boldsymbol{\mu}^{\ell,c}\right)^\top \boldsymbol{\Sigma}_\ell^{-1} \left(\boldsymbol{f}^\ell(\boldsymbol{x}) - \boldsymbol{\mu}^{\ell,c}\right) \tag{9}$$

$$K^\ell(\boldsymbol{x}) = \max_{c\in\{1,\ldots,C\}} K^{\ell,c}(\boldsymbol{x}). \tag{10}$$

The centroid of the Gaussian for class $c \in \{1, \ldots, C\}$ in layer $\ell \in \{1, \ldots, L\}$ is

$$\boldsymbol{\mu}^{\ell,c} = \frac{1}{n_c} \sum_{i=1}^{n} \left[y^{(i)} = c\right] \boldsymbol{f}^\ell\left(\boldsymbol{x}^{(i)}\right), \tag{11}$$

where $n_c$ is the number of samples with label $c$, and

$$\boldsymbol{\Sigma}_\ell = \frac{1}{n} \sum_{c=1}^{C} \sum_{i=1}^{n} \left[y^{(i)} = c\right] \left(\boldsymbol{f}^\ell(\boldsymbol{x}) - \boldsymbol{\mu}^{\ell,c}\right) \left(\boldsymbol{f}^\ell(\boldsymbol{x}) - \boldsymbol{\mu}^{\ell,c}\right)^\top \tag{12}$$

is the tied covariance matrix used for all classes in layer $\ell \in \{1, \ldots, L\}$.

To make the differences of latent embeddings of ID and OOD samples more pronounced, all samples are adversarially perturbed w.r.t. the maximal Mahalanobis score for each layer's confidence score:

$$\hat{\boldsymbol{x}}^\ell = \boldsymbol{x} + \epsilon\, \mathbf{sgn}\left(\nabla_{\boldsymbol{x}} K^\ell(\boldsymbol{x})\right). \tag{13}$$

This perturbed sample is used to compute $K^\ell\left(\hat{\boldsymbol{x}}^\ell\right)$. Finally, a logistic regression OOD detector is learned on a held-out validation set of a balanced mix of ID and OOD samples to learn weights $w_\ell$ for each layer $\ell \in \{1, \ldots, L\}$ using the $L$-dimensional inputs $\left[K_1\left(\hat{\boldsymbol{x}}^1\right), \ldots, K_L\left(\hat{\boldsymbol{x}}^L\right)\right]^\top$. The final *uncertainty* estimate becomes $u^{\text{Mah}}(\boldsymbol{x}) = \sum_{\ell=1}^{L} w_\ell K^\ell\left(\hat{\boldsymbol{x}}^\ell\right)$.

This is the only method in our benchmark that requires a mixed ID-OOD validation set for training the logistic regression OOD detector.

### A.1.5 Temperature Scaling

Temperature scaling [18] post-hoc calibrates the predictive softmax distribution $\boldsymbol{\pi}(\boldsymbol{x})$ by learning a temperature parameter $\tau \in \mathbb{R}_+$ on a held-out ID validation set after training and setting $\boldsymbol{\pi}(\boldsymbol{x}) := \mathbf{softmax}\left(\boldsymbol{f}(\boldsymbol{x})/\tau\right)$. Guo et al. [18] show that temperature scaling leads to improvements on both the ECE score and strictly proper scoring rules. To determine the optimal $\tau$, we perform a grid search over $\tau \in \{0.1, 0.2, 0.3, \ldots, 10.1\}$ and choose the one that leads to the lowest NLL loss, following [37].

### A.1.6 Deep Deterministic Uncertainty

The Deep Deterministic Uncertainty (DDU) method [37] applies the spectral normalization of SNGPs (Appendix A.2.1) to the hidden weights to establish a distance-aware latent space. It then fits a mixture of Gaussians to this latent space based on (ID) training set statistics. Unlike the Mahalanobis method, it

1. does not use adversarial perturbations;
2. only builds a latent density in the pre-logit layer;
3. does not tie the covariance matrix across classes:

$$\pi^c = \frac{n_c}{n}; \tag{14}$$

$$\boldsymbol{\mu}^c = \frac{1}{n_c} \sum_{i=1}^{n} \left[ y^{(i)} = c \right] \boldsymbol{f}^{L-1}\left(\boldsymbol{x}^{(i)}\right); \tag{15}$$

$$\boldsymbol{\Sigma}^c = \frac{1}{n_c - 1} \sum_{i=1}^{n} \left[ y^{(i)} = c \right] \left( \boldsymbol{f}^{L-1}\left(\boldsymbol{x}^{(i)}\right) - \boldsymbol{\mu}^c \right) \left( \boldsymbol{f}^{L-1}\left(\boldsymbol{x}^{(i)}\right) - \boldsymbol{\mu}^c \right)^\top \tag{16}$$

for $c \in \{1, \ldots, C\}$ where $\boldsymbol{f}^{L-1}(\boldsymbol{x})$ denotes the output of the pre-logit layer on input $\boldsymbol{x} \in \mathcal{X}$. Finally, it uses a held-out ID validation set to apply temperature scaling to the logits.

Unlike the other methods we evaluate, the DDU method uses two uncertainty estimators, one for epistemic uncertainty and one for aleatoric uncertainty. The epistemic estimator is the negative log probability of the pre-logit on sample $\boldsymbol{x} \in \mathcal{X}$ under the MoG:

$$u_{\mathrm{eu}}^{\mathrm{ddu}}(\boldsymbol{x}) = -\log p\left( \boldsymbol{f}^{L-1}(\boldsymbol{x}) \Big| \{\pi^c\}_{c=1}^C, \{\boldsymbol{\mu}^c\}_{c=1}^C, \{\boldsymbol{\Sigma}^c\}_{c=1}^C \right).$$

The aleatoric estimator is the entropy of the softmax predictive distribution:

$$u_{\mathrm{au}}^{\mathrm{ddu}}(\boldsymbol{x}) = \mathbb{H}\left( \boldsymbol{\pi}(\boldsymbol{x}) \right).$$

Mukhoti et al. [37] do not provide a predictive uncertainty estimator, and the sum of the aleatoric and epistemic estimator is not a performant choice for this task, as the magnitude of the epistemic part is usually much larger than that of the aleatoric part in practice.

During training, we employ the cross-entropy loss to match the network's predicted probabilities to the (one-hot) ground-truth labels.

For a fair comparison of DDU with the other methods, we use the epistemic estimator for the OOD detection task following Mukhoti et al. [37] and the best-performing one from Appendix D otherwise.

### A.1.7 Cross-Entropy Baseline

As a baseline, we also benchmark a deterministic single-point network trained with the cross-entropy loss. While this is a deterministic method, one can also equate it to a degenerate Dirac delta distribution in parameter space: $q(\boldsymbol{\theta}') = \delta(\boldsymbol{\theta} - \boldsymbol{\theta}')$, making it the simplest possible distributional method.

### A.2 Distributional Methods

Distributional methods output a second-order input-conditional probability distribution over probability vectors $q(\boldsymbol{\pi} \mid \boldsymbol{x})$.

### A.2.1 Spectral Normalized Gaussian Process

Spectral normalized Gaussian processes (SNGP) [32] give an approximate Bayesian treatment to obtain uncertainty estimates using spectral normalization of the parameter tensors and a last-layer Gaussian process approximated by Fourier features. For an input $x \in \mathcal{X}$, it predicts a multivariate Gaussian distribution

$$\mathcal{N}\left(\boldsymbol{B}\boldsymbol{\phi}(\boldsymbol{x}), \boldsymbol{\phi}(\boldsymbol{x})^\top \left(\boldsymbol{\Phi}^\top \boldsymbol{\Phi} + \boldsymbol{I}\right)^{-1} \boldsymbol{\phi}(\boldsymbol{x})\boldsymbol{I}\right), \tag{17}$$

where $\boldsymbol{B}$ is a learned parameter matrix that maps from the pre-logits to the logits, and $\boldsymbol{\phi}(\boldsymbol{x}) = \cos\left(\boldsymbol{W}\boldsymbol{f}^{L-1}(\boldsymbol{x}) + \boldsymbol{b}\right)$ is a random feature embedding of the input $\boldsymbol{x} \in \mathcal{X}$ with $\boldsymbol{f}^{L-1}(\boldsymbol{x})$ being a pre-logit embedding, $\boldsymbol{W}$ a fixed semi-orthogonal random matrix, and $\boldsymbol{b}$ a fixed random vector sampled from $\text{Uniform}(0, 2\pi)$. $\boldsymbol{\Phi}^\top \boldsymbol{\Phi}$ is the (unnormalized) empirical covariance matrix of the pre-logits of the training set. This is calculated during the last epoch. The multivariate Gaussian presented above can be Monte-Carlo sampled to obtain $M$ logit vectors. We use $M = 1000$ Monte-Carlo samples and did not notice differences between using $M \in \{10, 100, 1000, 10000\}$ samples. Unless noted otherwise, we use $M = 1000$ for all other method that require Monte-Carlo sampling aswell. During training, we calculate the BMA from the set of logits and use the cross-entropy loss to fit the BMA to the (one-hot) labels.

The method also applies spectral normalization to the hidden weights in each layer to satisfy input distance awareness. We treat whether to apply spectral normalization to the batch normalization modules and whether to use layer normalization in the GP layer as hyperparameters. We benchmark both SNGPs and their non-spectral-normalized variants (denoted by GP).

### A.2.2 Latent Heteroscedastic Classifier

Latent heteroscedastic classifiers (HET-XL) [6] construct a heteroscedastic Gaussian distribution in the pre-logit layer to model per-input uncertainties: $\mathcal{N}(\boldsymbol{\phi}(\boldsymbol{x}), \boldsymbol{\Sigma}(\boldsymbol{x}))$, where $\boldsymbol{\phi}(\boldsymbol{x})$ is the learned input-conditional pre-logit mean and

$$\boldsymbol{\Sigma}(\boldsymbol{x}) = \boldsymbol{V}(\boldsymbol{x})^\top \boldsymbol{V}(\boldsymbol{x}) + \text{diag}(\boldsymbol{d}(\boldsymbol{x})) \tag{18}$$

is an input-conditional full-rank covariance matrix. Both the low-rank term's $\boldsymbol{V}(\boldsymbol{x})$ and the diagonal term's $\boldsymbol{d}(\boldsymbol{x})$ are calculated as a linear function of the layer's output before the pre-logit layer.

One can Monte-Carlo sample the pre-logits from the above Gaussian distribution and obtain a set of logits by transforming each using the last linear layer of the network. During training, this set is used to calculate the Bayesian Model Average (BMA), the argmax of which is the final prediction.

HET-XL uses a temperature parameter to scale the logits before calculating the BMA. This is chosen using a validation set. During training, we sample a set of logits, calculate the BMA, and use the cross-entropy loss to fit the BMA to the (one-hot) labels.

### A.2.3 Laplace Approximation

The Laplace approximation [8] approximates a Gaussian posterior $q(\boldsymbol{\theta} \mid \mathcal{D})$ over the network parameters for a Gaussian prior $p(\boldsymbol{\theta})$ and likelihood defined by the network architecture. It uses the maximum a posteriori (MAP) estimate as the mean and the inverse Hessian of the loss evaluated at the MAP as the covariance matrix:

$$\mathcal{N}\left(\boldsymbol{\theta}_{\text{MAP}}, \left(\frac{\partial^2 \mathcal{L}(\mathcal{D}; \boldsymbol{\theta})}{\partial \theta_i \partial \theta_j}\bigg|_{\boldsymbol{\theta}_{\text{MAP}}}\right)^{-1}\right). \tag{19}$$

This is a post-hoc method applied to a point estimate network. Following the recommendation of Daxberger et al. [8], we employ a last-layer KFAC Laplace approximation and find the prior variance using cross-validation. We draw network outputs using the GLM predictive on CIFAR-10, and the NN predictive on ImageNet because of the infeasibility of calculating the network Jacobian for the GLM due to extreme memory requirements ($\approx 450$ GB VRAM).

### A.2.4 SWAG

The Stochastic Weight Averaging–Dropout [34] method takes a model that has either converged or is close to converging and fine-tunes it for a certain number of epochs while taking checkpoints

of it at evenly spaced points. It keeps track of the averaged checkpoint weights and their low-rank covariance matrix. Once the fine-tuning is over, the method fits a Gaussian over the parameter space with mean and covariance matrix from the collected checkpoints. During inference, it samples the parameter-space posterior and uses these samples to make multiple predictions per input. Following [34], we sample $M = 30$ parameters from the Gaussian posterior before evaluation and re-calculate the batch normalization statistics for each of them on a $0.1$ fraction of the training dataset.

### A.2.5 MC Dropout

MC Dropout [48] has been shown to be a variational approximation to a deep Gaussian process [12]. During training, only one logit vector per input is sampled, and the cross-entropy loss is used. MC Dropout in the realm of uncertainty quantification remains active during inference and is used to sample $M$ logits by performing $M$ forward passes. Therefore, it directly samples from $q(\boldsymbol{\pi} \mid \boldsymbol{x})$ without characterizing it.

### A.2.6 HetClassNN

The Heteroscedastic Classification Neural Network (HetClassNN) [23] employs an output head that predicts input-conditional heteroscedastic logit variance vectors. During training, the method MC approximates the integral of the softmax probabilities with respect to the logit-space Gaussian and trains this Bayesian Model Average (BMA) with the cross-entropy loss. We call the logit samples from the Gaussian 'internal logits.' During inference, there is another meta MC sampling step of $M = 30$ samples w.r.t. random dropout masks. This results in $M$ outputs from the method, each of which is a Bayesian Model average of the logit-space Gaussian w.r.t a different dropout mask. We refer to the logarithms of these outputs as the 'external logits' or just 'logits.'

### A.2.7 Deep Ensemble

Deep ensembles [29] are approximate model distributions that give rise to a mixture of Dirac deltas in parameter space: $q(\boldsymbol{\theta}) = \frac{1}{M} \sum_{i=1}^{M} \delta(\boldsymbol{\theta} - \boldsymbol{\theta}^{(i)})$. Predominantly used to reduce the variance in the predictions and improve model accuracy, deep ensembles can also be used as approximators to the true distribution $p(\boldsymbol{\theta})$ induced by the randomness over datasets $\mathcal{D} := \left\{ \left( \boldsymbol{x}^{(i)}, y^{(i)} \right) \mid i \in \{1, \ldots, n\}, \boldsymbol{x}^{(i)} \in \mathcal{X}, y^{(i)} \in \{1, \ldots, C\} \right\}$ in the generative process $p_{\text{data}}(\boldsymbol{x}, y)$.

We obtain a set of logits by performing a forward pass over all models. Similarly to MC Dropout, deep ensembles do not explicitly parameterize the distribution over the predictions – they only sample from it. We ensemble five independently trained cross-entropy models.

### A.2.8 Shallow Ensemble

Shallow ensembles [31] are lightweight approximations of deep ensembles. They use a shared backbone and $M$ output heads (often referred to as "experts"). With a single forward pass, one obtains $M$ logit vectors per input. During training, the BMA of the $M$ predictions is calculated and matched to the ground-truth labels.

### A.2.9 Evidential Deep Learning

The seminal evidential deep learning method of Sensoy et al. [46] (denoted by EDL following Ulmer et al. [50]) directly learns a second-order predictive distribution $q(\boldsymbol{\pi} \mid \boldsymbol{x})$ using closed-form Bayesian inference. In particular, it learns an input-conditional Dirichlet posterior $q(\boldsymbol{\pi} \mid \boldsymbol{x}) = \text{Dir}(\boldsymbol{\pi}; \boldsymbol{\beta}(\boldsymbol{x}))$ with a fixed Dirichlet (conjugate) prior $\text{Dir}(\mathbf{1})$ representing a total lack of information and a categorical distribution over the classes as the likelihood. The logits of the network, $\boldsymbol{f}(\boldsymbol{x})$, are turned into pseudo-counts $\boldsymbol{\alpha}(\boldsymbol{x}) \in \mathbb{R}_+^C$ using either the exp or the softplus activation function. The posterior distribution is obtained in closed form by setting $\boldsymbol{\beta}(\boldsymbol{x}) = \boldsymbol{\alpha}(\boldsymbol{x}) + \mathbf{1}$. The components of the IT decomposition can also be derived in closed form; see Ulmer et al. [50] for details.

The loss of the EDL method has three components:

$$\mathcal{L}^{\text{edl}} = \frac{1}{n} \sum_{i=1}^{n} \left\| \mathbf{one\_hot}\left(y^{(i)}\right) - \frac{\boldsymbol{\beta}\left(\boldsymbol{x}^{(i)}\right)}{S\left(\boldsymbol{x}^{(i)}\right)} \right\|^2 + \sum_{c=1}^{C} \frac{\beta\left(\boldsymbol{x}^{(i)}\right)_c \left(S\left(\boldsymbol{x}^{(i)}\right) - \beta\left(\boldsymbol{x}^{(i)}\right)_c\right)}{S^2\left(\boldsymbol{x}^{(i)}\right)\left(S\left(\boldsymbol{x}^{(i)}\right) + 1\right)}$$
$$+ \lambda_t D_{\text{KL}}\left(\text{Dir}\left(\bar{\boldsymbol{\beta}}\left(\boldsymbol{x}^{(i)}, y^{(i)}\right)\right) \,\middle\|\, \text{Dir}(\mathbf{1})\right), \tag{20}$$

where $S(\boldsymbol{x}) = \sum_{c=1}^{C} \beta\left(\boldsymbol{x}^{(i)}\right)_c$ and $\bar{\boldsymbol{\beta}}(\boldsymbol{x}, y) = \mathbf{one\_hot}(y) + (1 - \mathbf{one\_hot}(y)) \odot \boldsymbol{\beta}$ is the Dirichlet parameter vector after removing the prediction corresponding to the label's index. The first term matches the mean of the Dirichlet posterior to the (one-hot) GT labels. The second term reduces the summed variance of each index $c \in \{1, \dots, C\}$ of the random variable distributed as the Dirichlet posterior. These two terms concentrate the Dirichlet density onto the one-hot label. The third term is a regularizer that drives all dimensions of the Dirichlet parameter vector toward a complete lack of knowledge except the one corresponding to the GT label. $\lambda_t$ is the scheduled trade-off factor at step $t$. We use a linear up-scaling of $\lambda_t$ from 0 to $\lambda_{\max} \leq 1$. On CIFAR-10, $\lambda_{\max} = 1$ is used following Sensoy et al. [46]. Om ImageNet, this led to an overly strong regularizer that prohibited learning (as the regularizer's magnitude depends on the number of classes). We found $\lambda_{\max} = 0.001$ to be a performant maximum trade-off factor for ImageNet.

### A.2.10  PostNet

The PostNet method of Charpentier et al. [4] builds upon the EDL method. PostNet also keeps the prior parameters fixed to $\mathbf{1}$, but instead of directly predicting pseudo-counts $\boldsymbol{\alpha}(\boldsymbol{x})$, they are calculated as $\alpha(\boldsymbol{x})_c = n_c \cdot p_\phi(\boldsymbol{z}(\boldsymbol{x}) \mid c)$ where $\boldsymbol{z}(\boldsymbol{x})$ is the latent embedding of input $\boldsymbol{x} \in \mathcal{X}$, $n_c$ is the number of training samples of class $c \in \{1, \dots, C\}$ and $p_\phi(\boldsymbol{z}(\boldsymbol{x}) \mid c)$ is a class-conditional normalizing flow with parameters $\phi$. Intuitively, the class-conditional normalizing flows give soft class membership indicators to each input, and their indicators are weighted by the class size.

The PostNet method is trained with a regularized Uncertain Cross-Entropy (UCE) loss:

$$\mathcal{L}^{\text{postnet}} = \frac{1}{n} \sum_{i=1}^{n} \mathbb{E}_{\text{Dir}\left(\boldsymbol{\pi}; \boldsymbol{\beta}\left(\boldsymbol{x}^{(i)}\right)\right)} \left[ \text{CE}\left(\boldsymbol{\pi}, \mathbf{one\_hot}\left(y^{(i)}\right)\right) \right] - \lambda \mathbb{H}\left(\text{Dir}\left(\boldsymbol{\beta}\left(\boldsymbol{x}^{(i)}\right)\right)\right). \tag{21}$$

While the first term drives $\text{Dir}(\boldsymbol{\beta}(\boldsymbol{x}))$ toward a Dirac distribution concentrated at the one-hot label, the second term maximizes the entropy of the Dirichlet posterior. The effect of each is determined by the trade-off factor $\lambda$.

## B  Definition and Further Results of the Bregman Decomposition

Bregman decompositions [42, 19, 28, 17] use not only the second-order predictive distribution $q(\boldsymbol{\pi} \mid \boldsymbol{x})$ but also take the ground-truth (GT) generative process $p_{\text{data}}(\boldsymbol{x}, y)$ into account. Bregman decompositions break up the expected loss of a model over all possible training datasets. This variability is approximated by $q(\boldsymbol{\pi} \mid \boldsymbol{x})$:

$$\underbrace{\mathbb{E}_{q(\boldsymbol{\pi}|\boldsymbol{x}), p_{\text{data}}(y|\boldsymbol{x})} \left[ D_F \left[ \mathbf{one\_hot}(y) \,\|\, \boldsymbol{\pi} \right] \right]}_{\text{predictive}} = \underbrace{\mathbb{E}_{p_{\text{data}}(y|\boldsymbol{x})} \left[ D_F \left[ \mathbf{one\_hot}(y) \,\|\, \boldsymbol{\pi}^*(\boldsymbol{x}) \right] \right]}_{\text{aleatoric}}$$
$$+ \underbrace{\mathbb{E}_{q(\boldsymbol{\pi}|\boldsymbol{x})} \left[ D_F \left[ \tilde{\boldsymbol{\pi}}(\boldsymbol{x}) \,\|\, \boldsymbol{\pi} \right] \right]}_{\text{epistemic}} \tag{22}$$
$$+ \underbrace{D_F \left[ \boldsymbol{\pi}^*(\boldsymbol{x}) \,\|\, \tilde{\boldsymbol{\pi}}(\boldsymbol{x}) \right]}_{\text{bias}}$$

The loss $D_F$ is a Bregman divergence induced by the strictly convex function $F$, like the Euclidean distance or the KL divergence. Since $\boldsymbol{\pi}^*(\boldsymbol{x}) = \mathbb{E}_{p_{\text{data}}(y|\boldsymbol{x})} \left[ \mathbf{one\_hot}(y) \right]$ is the Bayes predictor, the aleatoric uncertainty is the Bayes risk of the generative process, which is irreducible and independent of the $q(\boldsymbol{\pi} \mid \boldsymbol{x})$ distribution. As this process is unknown in practice, we *estimate* the aleatoric term by $\mathbb{E}_{q(\boldsymbol{\pi}|\boldsymbol{x})} \left[ \mathbb{H}\left(\boldsymbol{\pi}\right) \right]$ to create estimators – but not for evaluation. Similarly to the IT decomposition, the epistemic uncertainty is the average distance of predictions $\boldsymbol{\pi} \sim q(\boldsymbol{\pi} \mid \boldsymbol{x})$ from their centroid, the

dual BMA $\tilde{\boldsymbol{\pi}}(\boldsymbol{x}) = \arg\min_{\boldsymbol{z} \in \Delta^{C-1}} \mathbb{E}_{q(\boldsymbol{\pi}|\boldsymbol{x})}[D_F[\boldsymbol{z} \parallel \boldsymbol{\pi}]]$. This average is calculated in a dual space, but in certain cases, it is equal to the BMA [19]. The Bregman decomposition has an additional term, the bias – an uncertainty source that subsumes the uncertainty about the function class [54].

## B.1 Special Form of the Bregman Decomposition for the Kullback-Leibler Divergence

When choosing $F(\cdot) = -\mathbb{H}(\cdot)$, we obtain $D_F[\cdot \parallel \cdot] = D_{\mathrm{KL}}(\cdot \parallel \cdot)$. Consider the predictive uncertainty term. A one-hot vector's entropy is zero; therefore, the predictive uncertainty becomes $\mathbb{E}_{q(\boldsymbol{\pi}|\boldsymbol{x}),p_{\mathrm{data}}(y|\boldsymbol{x})}[\mathrm{CE}(\mathbf{one\_hot}(y), \boldsymbol{\pi})]$. The aleatoric term takes a convenient form:

$$\mathbb{E}_{p_{\mathrm{data}}(y|\boldsymbol{x})}[D_{\mathrm{KL}}(\mathbf{one\_hot}(y) \parallel \boldsymbol{\pi}^*(\boldsymbol{x}))] = \mathbb{E}_{p_{\mathrm{data}}(y|\boldsymbol{x})}\left[\sum_{c=1}^C \mathbf{one\_hot}(y)_c \log \frac{y_c}{\pi_c^*(\boldsymbol{x})}\right]$$

$$= -\sum_{c=1}^C \pi_c^*(\boldsymbol{x}) \log \pi_c^*(\boldsymbol{x}) = \mathbb{H}(\boldsymbol{\pi}^*(\boldsymbol{x})). \qquad (23)$$

On datasets with multiple labels per input, this quantity is precisely the entropy of the (normalized) label distribution corresponding to the labeler votes.

To calculate $\tilde{\boldsymbol{\pi}}(\boldsymbol{x})$, we can proceed as follows.

$$\tilde{\boldsymbol{\pi}}(\boldsymbol{x}) = \underset{\boldsymbol{z} \in \Delta^{C-1}}{\arg\min} \, \mathbb{E}_{q(\boldsymbol{\pi}|\boldsymbol{x})}[D_{\mathrm{KL}}(\boldsymbol{z} \parallel \boldsymbol{\pi}(\boldsymbol{x}))] \qquad (24)$$

$$= \underset{\boldsymbol{z} \in \Delta^{C-1}}{\arg\min} \sum_{c=1}^C z_c \log z_c - \sum_{c=1}^C z_c \log\left(\exp\left(\mathbb{E}_{q(\boldsymbol{\pi}|\boldsymbol{x})}[\log \pi_c]\right)\right) \qquad (25)$$

$$= \underset{\boldsymbol{z} \in \Delta^{C-1}}{\arg\min} \sum_{c=1}^C z_c \log z_c - \sum_{c=1}^C z_c \log\left(\exp\left(\mathbb{E}_{q(\boldsymbol{\pi}|\boldsymbol{x})}[\log \pi_c]\right)\right)$$

$$\qquad\qquad + \sum_{c=1}^C z_c \log\left(\sum_{c'=1}^C \exp\left(\mathbb{E}_{q(\boldsymbol{\pi}|\boldsymbol{x})}[\log \pi_{c'}]\right)\right) \qquad (26)$$

$$= \underset{\boldsymbol{z} \in \Delta^{C-1}}{\arg\min} \sum_{c=1}^C z_c \log z_c - \sum_{c=1}^C z_c \log \underbrace{\frac{\exp\left(\mathbb{E}_{q(\boldsymbol{\pi}|\boldsymbol{x})}[\log \pi_c]\right)}{\sum_{c'=1}^C \exp\left(\mathbb{E}_{q(\boldsymbol{\pi}|\boldsymbol{x})}[\log \pi_{c'}]\right)}}_{p_c :=} \qquad (27)$$

$$= \underset{\boldsymbol{z} \in \Delta^{C-1}}{\arg\min} \, D_{\mathrm{KL}}(\boldsymbol{z} \parallel \boldsymbol{p}) \qquad (28)$$

$$= \boldsymbol{p}. \qquad (29)$$

Therefore, $\tilde{\boldsymbol{\pi}}(\boldsymbol{x}) = \mathbf{softmax}\left(\mathbb{E}_{q(\boldsymbol{\pi}|\boldsymbol{x})}[\log \boldsymbol{\pi}]\right)$, where $\log$ is applied elementwise.

## B.2 DEUP is a Special Case of Bregman

As mentioned in Appendix B, a closely related formula to Bregman is the risk decomposition of Lahlou et al. [28] where the predictive uncertainty is directly equated to the risk of a deterministic predictor $\boldsymbol{\pi} \colon \mathcal{X} \to \Delta^{C-1}$, not an expectation of risks over datasets or hypothesis distributions:

$$\underbrace{\mathcal{R}(\boldsymbol{\pi}, \boldsymbol{x})}_{\text{predictive}} = \underbrace{\mathcal{R}(\boldsymbol{\pi}^*, \boldsymbol{x})}_{\text{aleatoric}} + \underbrace{\mathcal{R}(\boldsymbol{\pi}, \boldsymbol{x}) - \mathcal{R}(\boldsymbol{\pi}^*, \boldsymbol{x})}_{\text{bias}} \qquad (30)$$

where $\mathcal{R}(\boldsymbol{\pi}, \boldsymbol{x}) = \mathbb{E}_{p(y|\boldsymbol{x})}[\mathcal{L}(\boldsymbol{\pi}(\boldsymbol{x}), y)]$ is the pointwise risk of $\boldsymbol{\pi}$ at $\boldsymbol{x} \in \mathcal{X}$. When choosing $\mathcal{L}$ to be the squared Euclidean distance or the Kullback-Leibler divergence, Equation 30 becomes a special case of Equation 22 for a Dirac distribution $q(\boldsymbol{\pi}' \mid \boldsymbol{x}) = \delta(\boldsymbol{\pi}' - \boldsymbol{\pi}(\boldsymbol{x}))$ at the deterministic prediction $\boldsymbol{\pi}(\boldsymbol{x})$. This formulation is desired when one wants the predictive uncertainty to be aligned with the risk of one particular predictor and not the expected risk over a hypothesis distribution.

## B.3 Correlation of Components and Limitations

Let us carry out the same experiments for Bregman as we did for the IT decomposition in Section 3.1 of the main paper. As the Bregman and DEUP decompositions (Equations 22 and 30) consider the

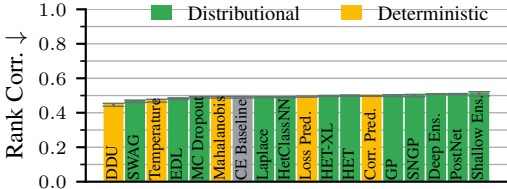 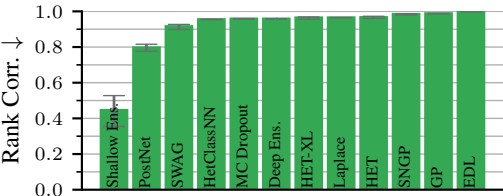

Figure B.1: *Left.* On ImageNet-ReaL, the rank correlation of the Bregman aleatoric and bias terms is between $0.45$ and $0.52$ for all distributional methods we benchmark. Note that the maximal rank correlation is less than one due to ties in the GT aleatoric uncertainties. *Right.* The Bregman decomposition shows similar rank correlation results to the IT decomposition between the estimated aleatoric uncertainty and the epistemic component on the ImageNet validation dataset (Fig. 2a).

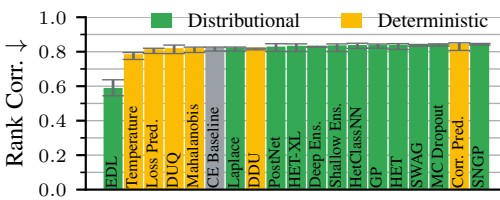 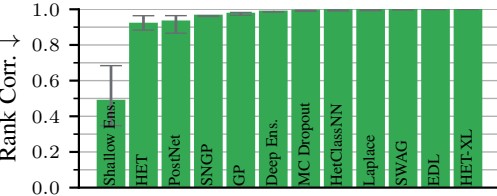

Figure B.2: *Left.* The rank correlation of the Bregman aleatoric and bias GT terms is above $0.59$ for all methods we benchmark on CIFAR-10. Note that the maximal rank correlation is less than one due to ties in the GT aleatoric uncertainties. *Right.* On CIFAR-10, the Bregman decomposition shows similarly strong rank correlation results between the estimated aleatoric uncertainty and the epistemic component as the IT decomposition does in Fig. 2b.

*ground-truth* label distribution as the aleatoric component, we use the IT aleatoric uncertainty as an estimator of it. The Bregman decomposition includes a bias component, whose correlations we also investigate.

### B.3.1 ImageNet

On the right of Fig. B.1, we can see that the correlation between the aleatoric and epistemic components of the Bregman decomposition is similarly high as for the IT decomposition in the main paper. The Bregman decomposition also has a bias term. On the left of Fig. B.1, we show that there is a considerable rank correlation between the Bregman ground-truth aleatoric and bias components. However, this is not severe enough to prevent the theoretical possibility of disentangling them via estimators.

### B.3.2 CIFAR-10

We see in Fig. B.2 (right) that the results using Bregman are virtually the same as those of the IT decomposition: most distributional methods exhibit very high rank correlations. As for the aleatoric and bias components, Fig. B.2 (left) shows that they have a high correlation on CIFAR-10 even when we use the GT values. Hence, there seems to be a fundamental limitation in disentangling them, no matter which estimators are used to approximate the GT values.

### B.4 Disentangling Epistemic and Aleatoric Uncertainty via Decomposition Formulas Is Feasible

In this section, we present the rank correlation between the *ground-truth* aleatoric uncertainty and the models' epistemic uncertainties. This serves to capture an inevitable level of correlation between these uncertainty sources, as measured (and defined) by the Bregman decomposition.

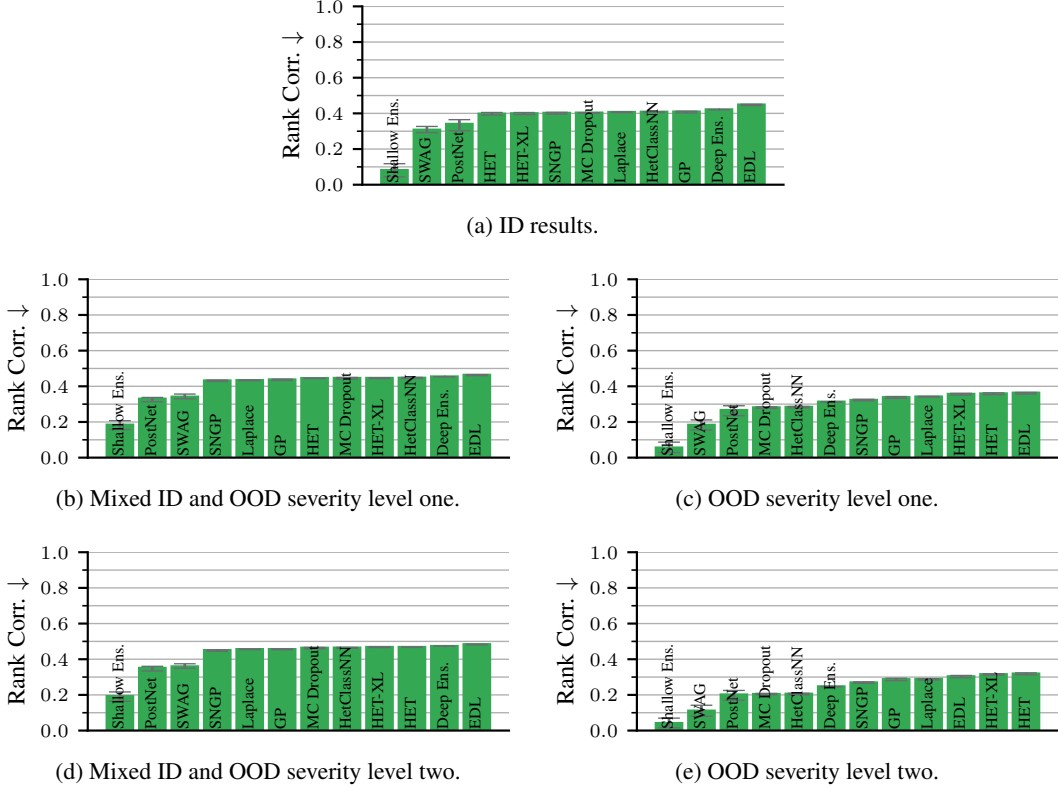

(a) ID results.

(b) Mixed ID and OOD severity level one.

(c) OOD severity level one.

(d) Mixed ID and OOD severity level two.

(e) OOD severity level two.

Figure B.3: On ImageNet, we find a positive rank correlation between the (ground-truth) aleatoric and epistemic components of the Bregman decomposition, implying that some level of correlation is inevitable when using this decomposition formula. However, this correlation is considerably lower than that between the aleatoric and epistemic *estimates* in Fig. 2a. This holds even if we increase the epistemic uncertainty in the dataset via ImageNet-C corruptions. We only show severity levels one and two here, as the GT aleatoric uncertainty values from the soft ImageNet-ReaL labels are only valid for these corruption levels – higher corruption would possibly change the soft label votes.

### B.4.1 ImageNet

ImageNet results on the correlation of the ground-truth aleatoric uncertainty and epistemic uncertainties of methods are shown in Fig. B.3. There is a positive rank correlation of up to $0.45$ between these quantities, implying that some level of correlation is inevitable (but not such extreme values displayed in Fig. 2a). The shallow ensemble, which already had the lowest *actual* correlation, also has the lowest *feasible* decorrelation. Since in the feasible decorrelation experiment, we replace the estimated AU with the GT AU and keep the epistemic component, this indicates that the epistemic estimate of the shallow ensemble is already quite disentangled. It is also high-performing, as seen in Fig. 3a. One way to improve this further was discussed in Section 3.3 of the main paper. We find that the Mahalanobis epistemic estimates and the CE baseline aleatoric estimates lead to reasonably well-performing yet decorrelated uncertainties. We hypothesize that the main reason is the explicit measurement of aleatoric and epistemic uncertainty at different parts of the computation graph. This, e.g., is lacking for the BMA decomposition: the aleatoric and epistemic estimates are generated from the same set of logits, which limits diverse behaviors across estimators.

### B.4.2 CIFAR-10

CIFAR-10 results on the correlation of the ground-truth aleatoric uncertainty and epistemic uncertainties of methods are shown in Fig. B.4a. Similar to ImageNet, there is a positive rank correlation between these quantities, implying a (low) inevitable level of correlation between the uncertainty sources. However, even for the most correlated second-order distribution of SWAG, with $0.39$ this

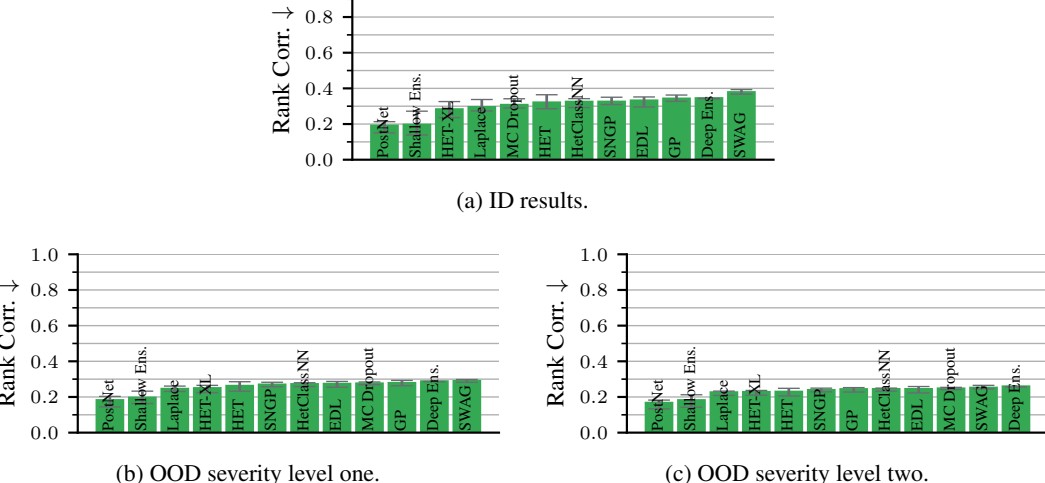

(a) ID results.

(b) OOD severity level one.

(c) OOD severity level two.

Figure B.4: On CIFAR-10, we find a positive rank correlation between the (ground-truth) aleatoric and epistemic components of the Bregman decomposition, implying that some level of correlation is inevitable when using this decomposition formula. However, this correlation is considerably lower than that between the aleatoric and epistemic *estimates* in Fig. 2b. This holds even if we increase the epistemic uncertainty in the dataset via CIFAR-10C corruptions. We only show severity levels one and two here, as the GT aleatoric uncertainty values from the soft CIFAR-10H labels are only valid for these corruption levels – higher corruption would possibly change the human annotators' CIFAR-10H soft label votes.

stays far below the rank correlations of the estimated components in Fig. 2b, which is above 0.99 for SWAG.

## B.5 Alignment of Methods with the Bregman Bias

### B.5.1 ImageNet

The rank correlation of benchmarked methods with the bias component of the Bregman decomposition is shown in Fig. B.5 for ID and OOD with severity two. Most methods exhibit a high rank correlation ($\geq 0.8$). This suggests that uncertainty estimators, to some extent, capture the model bias in terms of the Bregman formulation. All methods become less correlated with bias with increasing severity.

### B.5.2 CIFAR-10

The rank correlation of benchmarked methods with the bias component of the Bregman decomposition is shown in Fig. B.6. EDL is strongly correlated with the Bregman bias component, indicating that its uncertainty captures a notion of model bias. All methods become more highly correlated with bias with increasing OOD perturbation severity.

## C Further Results of the Information-Theoretical Decomposition

### C.1 Special Form on the Information-Theoretical Decomposition for Discrete Posteriors

Below, we show that the information-theoretical (IT) decomposition [11] separates the entropy of the BMA into an expected entropy term and a Jensen-Shannon divergence term when considering discrete uniform distributions $q(\boldsymbol{\pi} \mid \boldsymbol{x}) = \frac{1}{M} \sum_{m=1}^{M} \delta(\boldsymbol{\pi} - \boldsymbol{\pi}^{(m)})$ with $\boldsymbol{\pi}^{(m)} \sim q(\boldsymbol{\pi} \mid \boldsymbol{x})$. Note that our formulation uses sampling $M$ probability vectors from $q(\boldsymbol{\pi} \mid \boldsymbol{x})$ for each input $\boldsymbol{x} \in \mathcal{X}$, but the results also hold in the case of having a set of $M$ predictors $\{\boldsymbol{\pi}^{(m)}(\cdot)\}_{m=1}^{M}$.

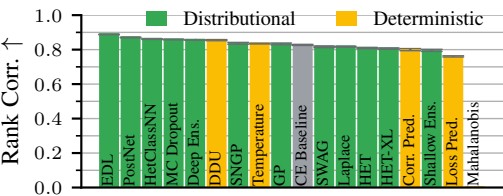

(a) ID rank correlation of methods with the Bregman decomposition's bias component.

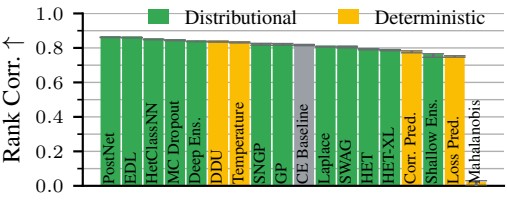

(b) Mixed ID and OOD rank correlation of methods with the Bregman bias using severity-one perturbations.

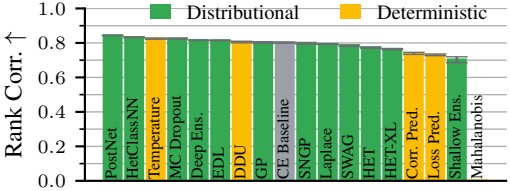

(c) OOD rank correlation of methods with the Bregman bias using severity-one perturbations.

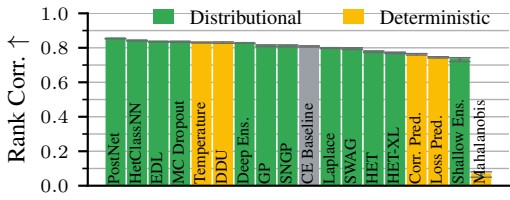

(d) Mixed ID and OOD rank correlation of methods with the Bregman bias using severity-two perturbations.

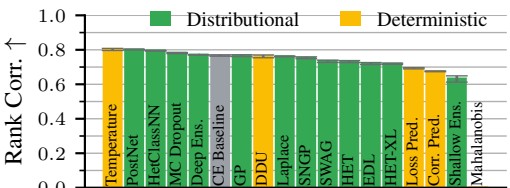

(e) OOD rank correlation of methods with the Bregman bias using severity-two perturbations.

Figure B.5: Rank correlation with the Bregman bias component on the ImageNet validation dataset. Most methods exhibit a high rank correlation ($\geq 0.8$). When going more OOD, all methods become less correlated with bias. Only severity levels one and two are shown, as the GT bias values from the soft ImageNet-ReaL labels are only valid for these corruption levels – higher corruption would possibly lead to a shift in labeler votes.

The IT decomposition treats the entropy of the predictive distribution $p(y \mid \boldsymbol{x}) = \int p(y \mid \boldsymbol{\pi}, \boldsymbol{x}) \, \mathrm{d}q(\boldsymbol{\pi} \mid \boldsymbol{x})$ as the predictive uncertainty metric and decomposes it into

$$\underbrace{\mathbb{H}_{p(y|\boldsymbol{x})}(y)}_{\text{predictive}} = \underbrace{\mathbb{E}_{q(\boldsymbol{\pi}|\boldsymbol{x})}\left[\mathbb{H}_{p(y|\boldsymbol{\pi},\boldsymbol{x})}(y)\right]}_{\text{aleatoric}} + \underbrace{\mathbb{I}_{p(y,\boldsymbol{\pi}|\boldsymbol{x})}(y;\boldsymbol{\pi})}_{\text{epistemic}}, \tag{31}$$

where $\mathbb{H}$ is the entropy and $\mathbb{I}$ is the mutual information.

Under a discrete uniform approximate distribution $q(\boldsymbol{\pi} \mid \boldsymbol{x})$, the predictive uncertainty is still the entropy of the BMA, and the aleatoric uncertainty also stays the expected entropy of the probability vectors of non-zero measure. We only have to show that the mutual information takes the convenient form of the Jensen-Shannon divergence under such an approximate posterior. Using $p(y, \boldsymbol{\pi} \mid \boldsymbol{x}) =$

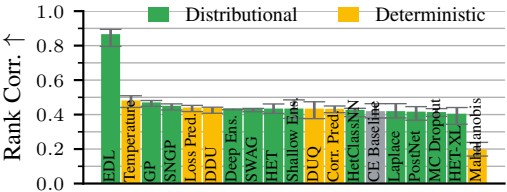

(a) ID rank correlation of methods with the Bregman decomposition's bias component.

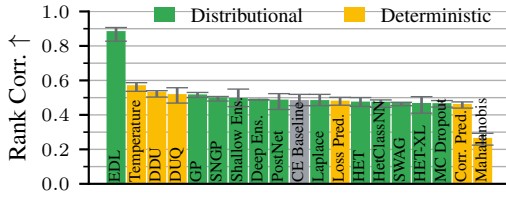

(b) Mixed ID and OOD rank correlation of methods with the Bregman bias using severity-one perturbations.

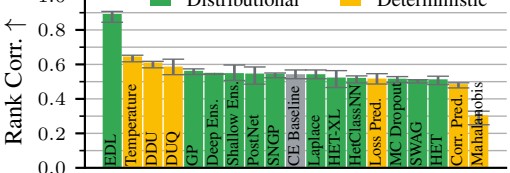

(c) OOD rank correlation of methods with the Bregman bias using severity-one perturbations.

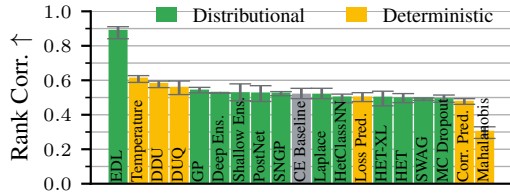

(d) Mixed ID and OOD rank correlation of methods with the Bregman bias using severity-two perturbations.

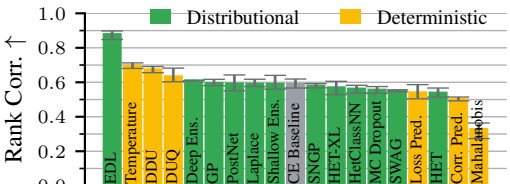

(e) OOD rank correlation of methods with the Bregman bias using severity-two perturbations.

Figure B.6: Rank correlations of the methods with the GT Bregman bias on CIFAR-10. EDL has a strong correlation, indicating that it estimates something close to the bias. When going more OOD, all methods become more highly correlated with the bias. Only severity levels one and two are shown, as the GT bias values from the soft CIFAR-10H labels are only valid for these corruption levels – higher corruption would possibly lead to a shift in labeler votes.

$p(y \mid \boldsymbol{\pi}, \boldsymbol{x}) q(\boldsymbol{\pi} \mid \boldsymbol{x})$, we have

$$\mathbb{I}_{p(y, \boldsymbol{\pi} \mid \boldsymbol{x})}(y; \boldsymbol{\pi}) = \sum_{y=1}^{C} \int \log \frac{p(y, \boldsymbol{\pi} \mid \boldsymbol{x})}{p(y \mid \boldsymbol{x}) q(\boldsymbol{\pi} \mid \boldsymbol{x})} \, \mathrm{d}p(y, \boldsymbol{\pi} \mid \boldsymbol{x}) \tag{32}$$

$$= \frac{1}{M} \sum_{m=1}^{M} \sum_{y=1}^{C} p\left(y \mid \boldsymbol{\pi}^{(m)}\right) \log \frac{p\left(y \mid \boldsymbol{\pi}^{(m)}\right)}{p(y \mid \boldsymbol{x})} \tag{33}$$

$$= -\frac{1}{M} \sum_{m=1}^{M} \mathbb{H}\left(\boldsymbol{\pi}^{(m)}\right) - \sum_{y=1}^{C} \frac{1}{M} \sum_{m=1}^{M} p\left(y \mid \boldsymbol{\pi}^{(m)}\right) \log p(y \mid \boldsymbol{x}) \tag{34}$$

$$= \mathbb{H}\left(\frac{1}{M} \sum_{m=1}^{M} \boldsymbol{\pi}^{(m)}\right) - \frac{1}{M} \sum_{m=1}^{M} \mathbb{H}\left(\boldsymbol{\pi}^{(m)}\right) \tag{35}$$

which is the Jensen-shannon divergence of the distributions $p\left(y \mid \boldsymbol{\pi}^{(m)}\right)$, $\boldsymbol{\pi}^{(m)} \sim q(\boldsymbol{\pi} \mid \boldsymbol{x})$, $m \in \{1, \ldots, M\}$.

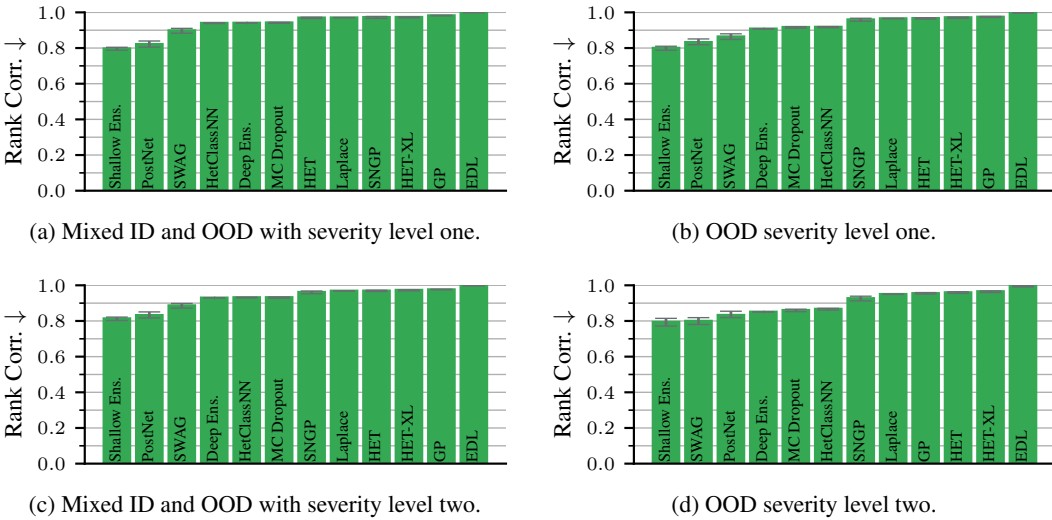

(a) Mixed ID and OOD with severity level one.

(b) OOD severity level one.

(c) Mixed ID and OOD with severity level two.

(d) OOD severity level two.

Figure C.1: Rank correlation of the aleatoric and epistemic components of the IT decomposition when increasing the epistemic uncertainty by going OOD on ImageNet-ReaL. Increasing the epistemic uncertainty of the datasets only slightly decreases the internal correlation of the estimates.

## C.2 Entanglement on Datasets with Increased Epistemic Uncertainty

The main paper showed in Section 3.1 that the IT decomposition's aleatoric and epistemic components are highly correlated across all distributional methods. In this section, we introduce epistemic uncertainty by going OOD via ImageNet-C distribution to find if this decorrelates the components.

### C.2.1 ImageNet

Fig. C.1 shows the results at severity levels one and two. The estimates generally become slightly less correlated as we go more OOD, but the correlations do not lower considerably.

### C.2.2 CIFAR-10

Results for severity levels one and two are shown in Fig. C.2. The increased epistemic uncertainty does not help decorrelate the components. Quite the opposite; the previously most uncorrelated estimates on the Shallow Ensemble become more highly correlated.

## C.3 Performance of Decorrelated Methods using the Information-Theoretical Components

In Sections 3.2 and 3.3, we use the best-performing aggregator for predicting aleatoric and epistemic uncertainty to ensure each second-order method has the best possible chances. In this section, we solely use the aggregators dictated by the IT decomposition.

### C.3.1 ImageNet

In Fig. C.3, we replace the aggregators of the previously best-performing distributional method, Shallow Ensemble, with the aggregators dictated by the IT decomposition. This reduces its performance to the CE baseline level. This shows that the choice of the aggregator counts and that it is not always the intuitively expected aggregator that performs best.

### C.3.2 CIFAR-10

As on ImageNet, replacing the aggregators of Shallow Ensemble with the IT decomposed ones on CIFAR-10 also lowers its performance on both the aleatoric and epistemic task to or below the CE baseline level in Fig. C.4.

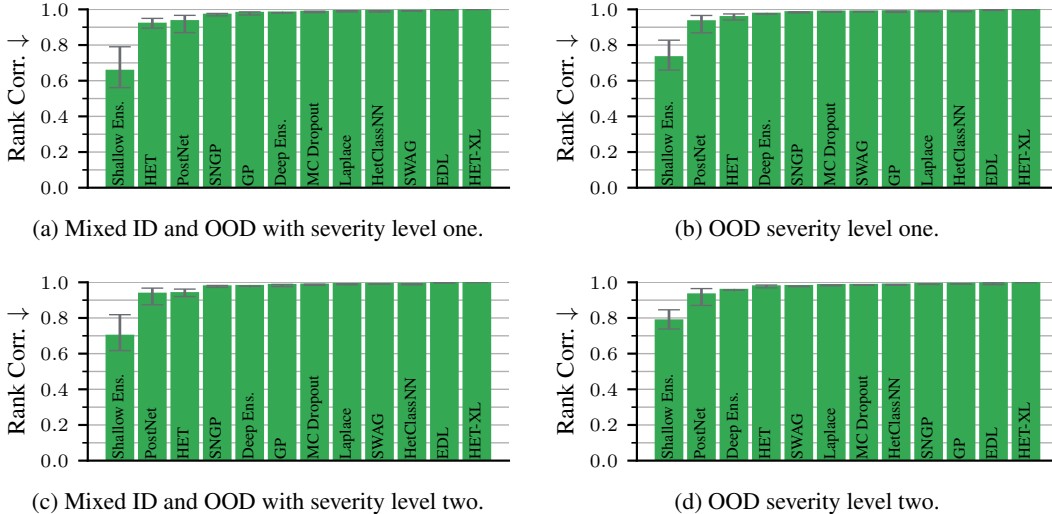

(a) Mixed ID and OOD with severity level one.

(b) OOD severity level one.

(c) Mixed ID and OOD with severity level two.

(d) OOD severity level two.

Figure C.2: Rank correlation of the aleatoric and epistemic components of the IT decomposition when increasing the epistemic uncertainty by going OOD on CIFAR-10H. The increased epistemic uncertainty does not decorrelate the components. Quite the opposite, it leads to even more highly correlated components.

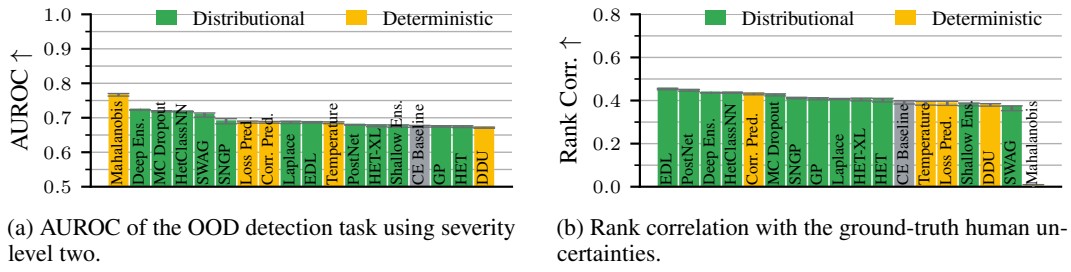

(a) AUROC of the OOD detection task using severity level two.

(b) Rank correlation with the ground-truth human uncertainties.

Figure C.3: On ImageNet, replacing the aggregators of the least-correlated distributional method, Shallow Ensemble, with the ones that the IT decomposition proposes drastically lowers its performance. All other methods are equipped with their best-performing estimator for the respective tasks, showing that specialized estimators work better.

## C.4 Pearson Correlation Results on ImageNet

Fig. C.5 shows Pearson correlation results between the IT decomposition's AU and EU term. In addition to the estimates having a strong monotonic relationship, they are also often linearly correlated.

## C.5 Cross-Evaluation of the IT Decomposition's Components on ImageNet

In Appendix C.5, we cross-evaluate the epistemic and aleatoric terms of the information-theoretical decomposition on the opposite task, which leads to two conclusions. (i) The epistemic estimates perform notably well on aleatoric uncertainty evaluation, which contradicts the common claim that epistemic estimates are not useful ID. (ii) The aleatoric estimates are almost always better than the epistemic ones on OOD detection, going against the wide belief that aleatoric estimates are not to be trusted OOD. This is another consequence of the entanglement that we uncover in our paper.

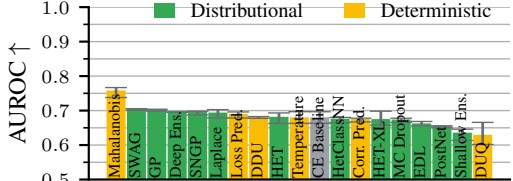
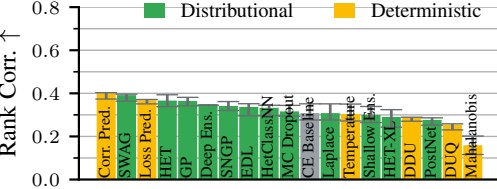

(a) AUROC of OOD detection performance of methods using perturbations of severity level two.

(b) Rank correlation between methods and the Bregman aleatoric component.

Figure C.4: Shallow ensemble underperforms the cross-entropy baseline on CIFAR-10 when using the estimators of the IT decomposition for the OOD detection and human uncertainty alignment tasks. All other methods are equipped with their best-performing estimator for the respective tasks, showing that the IT decomposition is not practically beneficial.

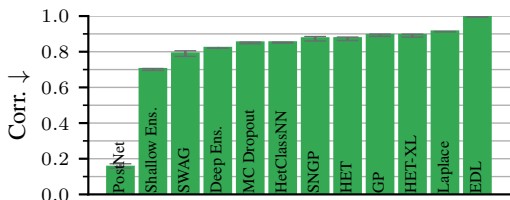

Figure C.5: Pearson correlation results of the IT decomposition's aleatoric and epistemic terms on ImageNet. The relationship is often not only monotonic but also linear, with PostNets being a notable exception.

# D   Definitions and Results of the Benchmarked Aggregators

In practical applications, distributional methods output a discrete set of probability vectors $\{\boldsymbol{\pi}^{(m)}\}_{m=1}^{M} \sim q(\boldsymbol{\pi} \mid \boldsymbol{x})$ per input $\boldsymbol{x} \in \mathcal{X}$.[4] This set can be aggregated in several ways to construct an uncertainty estimate $u(\boldsymbol{x})$. Commonly used aggregators are the Bayesian Model Average (BMA):

$$\bar{\boldsymbol{\pi}}(\boldsymbol{x}) = \frac{1}{M} \sum_{m=1}^{M} \boldsymbol{\pi}^{(m)}, \tag{36}$$

and the Bregman decomposition's central prediction term (Appendix B):

$$\tilde{\boldsymbol{\pi}}(\boldsymbol{x}) = \mathbf{softmax}\left(\frac{1}{M} \sum_{m=1}^{M} \log \boldsymbol{\pi}^{(m)}\right), \tag{37}$$

followed by taking their maximum probability, entropy, mutual information, or expected divergence [37, 11, 57, 19, 17]. Similarly, one can take the expected maximum probability and expected entropy over the set of probability vectors [37]. These possible choices are detailed below with pointers to their use in the literature.

## D.1   Entropy-Based Aggregators

According to the Source Coding Theorem, the entropy of the code is a fundamental and tight lower bound on the expected code word length for prefix-free symbol codes [58]. The entropy is an expectation over the length of per-symbol codewords. For general distributions $p(\boldsymbol{x})$, it intuitively measures the spread or the "amount of surprise" in $p(\boldsymbol{x})$: a higher entropy indicates more stochasticity in the distribution. We consider three entropy-based aggregators of $\{\boldsymbol{\pi}^{(m)}\}_{m=1}^{M} \sim q(\boldsymbol{\pi} \mid \boldsymbol{x})$ per input

---

[4]Note that our formulation uses sampling $M$ probability vectors from $q(\boldsymbol{\pi} \mid \boldsymbol{x})$ for each input $\boldsymbol{x} \in \mathcal{X}$, but the results also hold in the case of having a set of $M$ predictors $\{\boldsymbol{\pi}^{(m)}(\cdot)\}_{m=1}^{M}$.

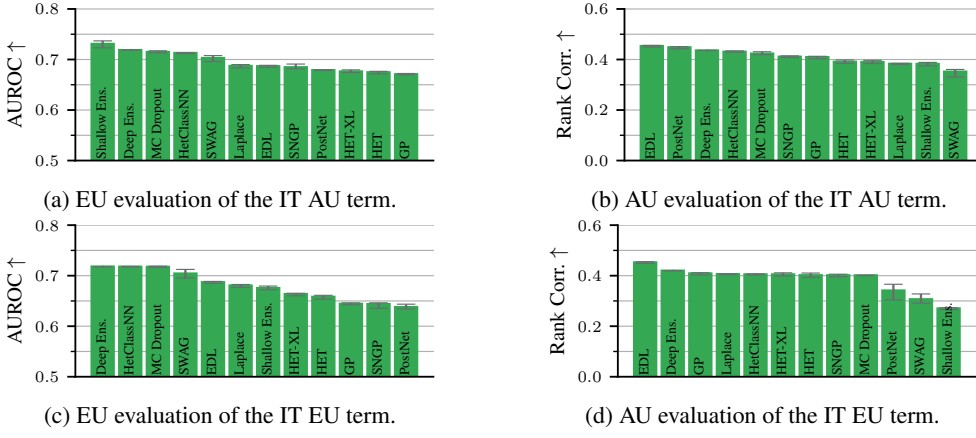

(a) EU evaluation of the IT AU term.

(b) AU evaluation of the IT AU term.

(c) EU evaluation of the IT EU term.

(d) AU evaluation of the IT EU term.

Figure C.6: Cross-evaluation of the IT decomposition's terms on ImageNet.

$x \in \mathcal{X}$:

$$u(\boldsymbol{x}) = \mathbb{H}\left(\bar{\boldsymbol{\pi}}(\boldsymbol{x})\right) \tag{38}$$

$$u(\boldsymbol{x}) = \mathbb{H}\left(\tilde{\boldsymbol{\pi}}(\boldsymbol{x})\right) \tag{39}$$

$$u(\boldsymbol{x}) = \frac{1}{M}\sum_{m=1}^{M}\mathbb{H}\left(\boldsymbol{\pi}^{(m)}\right) \tag{40}$$

The entropy appears both in the IT and Bregman decompositions (Eq. (1), Eq. (22)). Eq. (38) is often cited to capture predictive (or total) uncertainty, whereas Eq. (40) is known to capture aleatoric uncertainty [37, 11, 57]. As $\tilde{\boldsymbol{\pi}}(\boldsymbol{x})$ is a central predictor similar to $\bar{\boldsymbol{\pi}}(\boldsymbol{x})$, its entropy aligns well with a notion of predictive uncertainty.

### D.2 Maximum-Probability-Based Aggregators

Maximum-probability-based aggregators are similar to entropy-based ones: a small maximum probability value in the prediction vector necessarily means that all entries are small, leading to a high spread and entropy. As uncertainty estimates are higher when the model is more uncertain by convention, one usually takes one minus the maximum probability as a notion of uncertainty. We consider three maximum-probability-based aggregators of $\{\boldsymbol{\pi}^{(m)}\}_{m=1}^{M}\, q(\boldsymbol{\pi} \mid \boldsymbol{x})$ per input $\boldsymbol{x} \in \mathcal{X}$:

$$u(\boldsymbol{x}) = 1 - \max_{c \in \{1,\dots,C\}} \bar{\pi}_c(\boldsymbol{x}) \tag{41}$$

$$u(\boldsymbol{x}) = 1 - \max_{c \in \{1,\dots,C\}} \tilde{\pi}_c(\boldsymbol{x}) \tag{42}$$

$$u(\boldsymbol{x}) = 1 - \frac{1}{M}\sum_{m=1}^{M} \max_{c \in \{1,\dots,C\}} \pi_c^{(m)} \tag{43}$$

The maximum-probability-based aggregators are restricted to the $[0, 1]$ range. This is particularly important for (strictly) proper scoring rules for the correctness of prediction [36] and the notion of calibration, including the ECE and the reliability diagram [39]. Similarly to the entropy-based aggregators, Eq. (41) and Eq. (42) align with a notion of predictive uncertainty, whereas Eq. (43) is more aligned with a notion of aleatoric uncertainty.

### D.3 Disagreement-Based Aggregators

One can directly use the epistemic components of the Bregman and IT decompositions as they do not require a ground truth. In particular, one can use

$$u(\boldsymbol{x}) = \mathbb{H}\left(\bar{\boldsymbol{\pi}}(\boldsymbol{x})\right) - \frac{1}{M}\sum_{m=1}^{M}\mathbb{H}\left(\boldsymbol{\pi}^{(m)}\right), \quad \boldsymbol{\pi}^{(m)} \sim q(\boldsymbol{\pi} \mid \boldsymbol{x})\, \forall m \in \{1,\dots,M\}, \tag{44}$$

the (discretized) epistemic part of the IT decomposition, which is the Jensen-Shannon Divergence (see Appendix C.1), or

$$u(\boldsymbol{x}) = \frac{1}{M} \sum_{m=1}^{M} \left[ D_{\mathrm{KL}} \left( \tilde{\boldsymbol{\pi}}(\boldsymbol{x}) \,\middle\|\, \boldsymbol{\pi}^{(m)} \right) \right], \quad \boldsymbol{\pi}^{(m)} \sim q(\boldsymbol{\pi} \mid \boldsymbol{x}) \, \forall m \in \{1, \dots, M\}, \quad (45)$$

the (discretized) epistemic part of the Bregman decomposition, which is the expected divergence from the central predictor (see Appendix B.1). As both aggregators are divergences, they capture disagreement among a set of models. Thus, they are usually cited to be aligned with epistemic uncertainty [37, 11, 57, 19, 17].

Kendall and Gal [23] propose the expected variance of the logit or probability vectors as a measure of epistemic uncertainty:

$$u(\boldsymbol{x}) = \frac{1}{C} \sum_{c=1}^{C} \left[ \frac{1}{M} \sum_{m=1}^{M} f_c^{(m)}(\boldsymbol{x})^2 - \left( \frac{1}{M} \sum_{m=1}^{M} f_c^{(m)}(\boldsymbol{x}) \right)^2 \right], \quad (46)$$

$$u(\boldsymbol{x}) = \frac{1}{C} \sum_{c=1}^{C} \left[ \frac{1}{M} \sum_{m=1}^{M} \pi_c^{(m)}(\boldsymbol{x})^2 - \left( \frac{1}{M} \sum_{m=1}^{M} \pi_c^{(m)}(\boldsymbol{x}) \right)^2 \right]. \quad (47)$$

For the HetClassNN method, we also calculate these estimates using the internal logits.

Unless stated otherwise, we use the best-performing alternative for each distributional method in the benchmarks. For these methods, the model's prediction is always the most confident class of the BMA. For deterministic methods, we use their "canonical" uncertainty estimator introduced in Appendix A.1.

### D.4 Dempster-Shafer Value

The Dempster-Shafer (D-S) value Sensoy et al. [46] is an outlier: it does not fit into any of the aforementioned aggregator categories. In the framework of Evidential Deep Learning (see Appendix A.2.9), the D-S value is inversely proportional to the Dirichlet strength $S(\boldsymbol{x}) = \sum_{c=1}^{C} \beta(\boldsymbol{x})_c$ for input $\boldsymbol{x} \in \mathcal{X}$:

$$u(\boldsymbol{x}) = \frac{C}{S(\boldsymbol{x})}. \quad (48)$$

Informally, the more evidence the predictor has, the less uncertain it is about the input $\boldsymbol{x} \in \mathcal{X}$.

The D-S value can be extended to non-evidential methods by treating the exponentiated logits as pseudo-counts for the individual classes and setting

$$\boldsymbol{\beta}(\boldsymbol{x}) = \exp\left(\boldsymbol{f}(\boldsymbol{x})\right) + \mathbf{1}. \quad (49)$$

where the exponentiation is applied elementwise, and the $+$ operator denotes vector addition.

### D.5 The Behavior of the Aggregators Does Not Align With What the Literature Suggests

In this section, we collect per-aggregator results of specific methods to highlight that the best-performing aggregator often goes against what these aggregators intuitively aim to capture as described in Appendix D. Below, we provide a list of abbreviations used in the figures and connect them to the formulas in Appendix D. The "it" and "b" superscripts refer to the IT and Bregman decompositions, respectively. "AU", "PU", "EU", and "B" are shorthands for aleatoric, predictive,

epistemic uncertainty, and bias, respectively.

$$\text{PU}^{\text{it}} \equiv \mathbb{H}\left(\bar{\boldsymbol{\pi}}(\boldsymbol{x})\right) \tag{50}$$

$$\text{AU}^{\text{it}} \equiv \frac{1}{M}\sum_{m=1}^{M}\mathbb{H}\left(\boldsymbol{\pi}^{(m)}\right) \tag{51}$$

$$\text{EU}^{\text{it}} \equiv \mathbb{H}\left(\bar{\boldsymbol{\pi}}(\boldsymbol{x})\right) - \frac{1}{M}\sum_{m=1}^{M}\mathbb{H}\left(\boldsymbol{\pi}^{(m)}\right) \tag{52}$$

$$\text{PU}^{\text{b}} \equiv \frac{1}{M}\sum_{m=1}^{M}\left[\text{CE}\left(\boldsymbol{\pi}^*(\boldsymbol{x}), \boldsymbol{\pi}^{(m)}\right)\right] \tag{53}$$

$$\text{AU}^{\text{b}} \equiv \mathbb{H}\left(\boldsymbol{\pi}^*(\boldsymbol{x})\right) \tag{54}$$

$$\text{EU}^{\text{b}} \equiv \frac{1}{M}\sum_{m=1}^{M}\left[D_{\text{KL}}\left[\tilde{\boldsymbol{\pi}}(\boldsymbol{x}) \| \boldsymbol{\pi}^{(m)}\right]\right] \tag{55}$$

$$\text{B}^{\text{b}} \equiv D_{\text{KL}}\left[\boldsymbol{\pi}^*(\boldsymbol{x}) \| \tilde{\boldsymbol{\pi}}(\boldsymbol{x})\right] \tag{56}$$

$$\mathbb{H}\left(\tilde{\boldsymbol{\pi}}\right) \equiv \mathbb{H}\left(\tilde{\boldsymbol{\pi}}(\boldsymbol{x})\right) \tag{57}$$

$$1 - \mathbb{E}\left[\max\boldsymbol{\pi}\right] \equiv 1 - \frac{1}{M}\sum_{m=1}^{M}\max_{c\in\{1,\dots,C\}}\pi_c^{(m)} \tag{58}$$

$$1 - \max\bar{\boldsymbol{\pi}} \equiv 1 - \max_{c\in\{1,\dots,C\}}\bar{\pi}_c(\boldsymbol{x}) \tag{59}$$

$$1 - \max\tilde{\boldsymbol{\pi}} \equiv 1 - \max_{c\in\{1,\dots,C\}}\tilde{\pi}_c(\boldsymbol{x}) \tag{60}$$

$$\text{D-S} \equiv \frac{C}{S(\boldsymbol{x})} \tag{61}$$

$$\mathbb{E}\left[\text{var}\,\boldsymbol{f}\right] \equiv \frac{1}{C}\sum_{c=1}^{C}\left[\frac{1}{M}\sum_{m=1}^{M}f_c^{(m)}(\boldsymbol{x})^2 - \left(\frac{1}{M}\sum_{m=1}^{M}f_c^{(m)}(\boldsymbol{x})\right)^2\right] \tag{62}$$

$$\mathbb{E}\left[\text{var}\,\boldsymbol{\pi}\right] \equiv \frac{1}{C}\sum_{c=1}^{C}\left[\frac{1}{M}\sum_{m=1}^{M}\pi_c^{(m)}(\boldsymbol{x})^2 - \left(\frac{1}{M}\sum_{m=1}^{M}\pi_c^{(m)}(\boldsymbol{x})\right)^2\right] \tag{63}$$

where $\boldsymbol{\pi}^{(m)} \sim q(\boldsymbol{\pi} \mid \boldsymbol{x})$, $\boldsymbol{\pi}^{(m)} = \textbf{softmax}(\boldsymbol{f})\ \forall m \in \{1,\dots,M\}$.

### D.5.1 Aleatoric and predictive aggregators are often best for OOD detection

Let us consider the binary prediction task of distinguishing ID and OOD samples. Fig. D.1 and Fig. D.2 show the per-aggregator results on the OOD detection task for the GP and MC Dropout methods, respectively. These figures highlight two important observations:

1. the best aggregator for the task varies among different methods and

2. the disagreement-based epistemic uncertainty aggregators are often not the best for detecting OOD samples, against their original intuition.

Both results show that the choice of the estimator should be treated pragmatically based on performance because there are no intuitions that would consistently give the best estimator in each scenario.

### D.5.2 Methods are not equally sensitive to the choice of aggregator on correctness prediction

Let us turn to the binary *correctness* prediction task. Fig. D.3 and Fig. D.4 show the per-aggregator results on the correctness prediction detection task for the HET-XL and Deep Ensemble methods, respectively. These figures show that HET-XL is considerably less sensitive to the choice of the

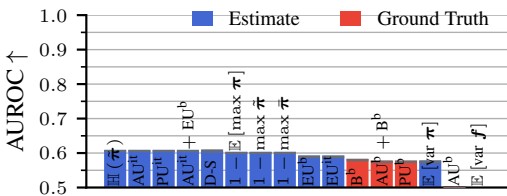

(a) GP OOD detection AUROC with severity level one.

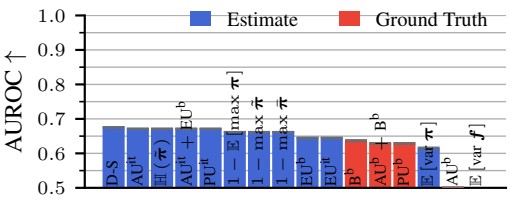

(b) GP OOD detection AUROC with severity level two.

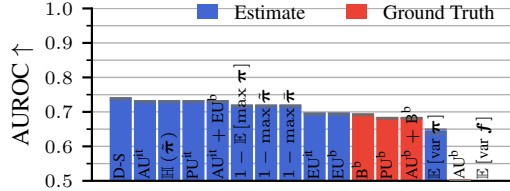

(c) GP OOD detection AUROC with severity level three.

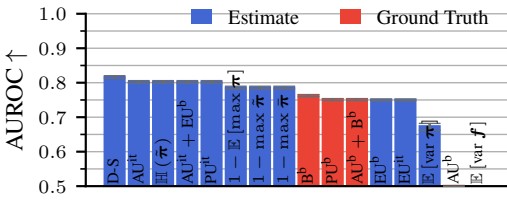

(d) GP OOD detection AUROC with severity level four.

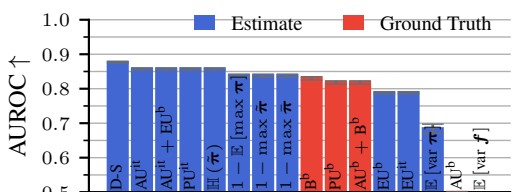

(e) GP OOD detection AUROC with severity level five.

Figure D.1: OOD detection results of the GP method on the ImageNet validation dataset measured by the AUROC metric. The OOD detection performance of all aggregators increases steadily as we increase the severity of the perturbed half of the mixed dataset. However, the disagreement-based epistemic aggregators, $EU^{it}$ and $EU^b$, notably underperform the $AU^{it}$ aggregator, even though the epistemic aggregators are deemed more suitable for OOD detection in the literature.

aggregator, and the ranking of the aggregators is inconsistent between the two methods. The epistemic aggregators are among the worst-performing estimates for deep ensembles. One might say that this is in line with their intuitions, as they are being used ID, but on HET-XL, they perform as strongly as predictive estimators. This reinforces that intuitions should not be used to guide the aggregator choice and to rather treat it as a hyperparameter for optimal performance.

# E   Goals of Disentanglement

What does it mean to have disentangled uncertainty estimators? Consider two estimators $u^{(a)}(\boldsymbol{x}), u^{(e)}(\boldsymbol{x})$ and *ground-truth* aleatoric and epistemic uncertainties $U^{(a)}(\boldsymbol{x}), U^{(e)}(\boldsymbol{x})$ for each input $\boldsymbol{x}_i$. The estimators $u^{(a)}$ and $u^{(e)}$ are decorrelated if

1. $u^{(a)}$ has low rank correlation with $U^{(e)}$ and
2. $u^{(e)}$ has low rank correlation with $U^{(a)}$.

Importantly, $u^{(a)}$ and $u^{(e)}$ having a severely high rank correlation prohibits disentanglement. Further, they are well-performing if

3. $u^{(a)}$ has high rank correlation with $U^{(a)}$ and
4. $u^{(e)}$ has high rank correlation with $U^{(e)}$.

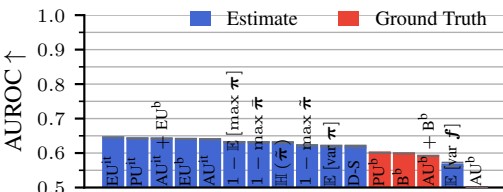

(a) MC Dropout OOD detection AUROC with severity level one.

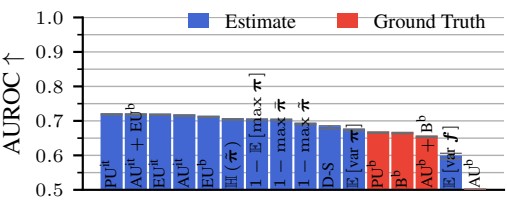

(b) MC Dropout OOD detection AUROC with severity level two.

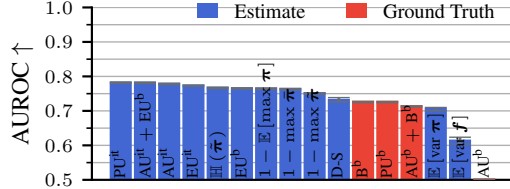

(c) MC Dropout OOD detection AUROC with severity level three.

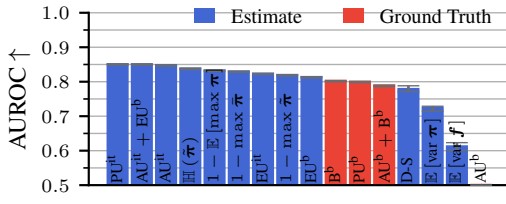

(d) MC Dropout OOD detection AUROC with severity level four.

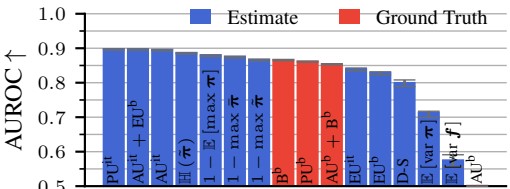

(e) MC Dropout OOD detection AUROC with severity level five.

Figure D.2: OOD detection results of the MC Dropout method on the ImageNet validation dataset measured by the AUROC metric. Contrarily to the GP method in Fig. D.1, the disagreement-based epistemic aggregators are on par with the predictive and aleatoric aggregators but lose their edge as the severity level increases.

We call uncertainty estimators disentangled when they are simultaneously decorrelated and well-performing.

Inspired by generalized bias-variance decompositions [42, 17], one may treat the training dataset $\mathcal{D}$ as a random variable sampled from the generative process $p(\boldsymbol{x}, y)$ and record the variability of the trained predictor under dataset change. Following the Bregman decomposition, one may then define

$$U^{(e)}(\boldsymbol{x}) := \mathbb{E}_{p(\mathcal{D})}\left[D_F\left[\tilde{\boldsymbol{\pi}}(\boldsymbol{x}) \parallel \boldsymbol{\pi}_{\mathcal{D}}(\boldsymbol{x})\right]\right] \tag{64}$$

with the corresponding central predictor $\tilde{\boldsymbol{\pi}}(\boldsymbol{x}) = \arg\min_{\boldsymbol{z} \in \Delta^{C-1}} \mathbb{E}_{p(\mathcal{D})}\left[D_F\left[\boldsymbol{z} \parallel \boldsymbol{\pi}_{\mathcal{D}}(\boldsymbol{x})\right]\right]$. As this is impossible to obtain in practical setups (or is too noisy to MC-approximate), we instead consider the proxy task of OOD detection for evaluating the disentanglement of aleatoric and epistemic uncertainties.

The above definition generalizes to any pair of different uncertainty sources, e.g., the Bregman bias and aleatoric uncertainties. The workaround of choosing a proxy task is not needed to evaluate the Bregman bias and aleatoric components' disentanglement.

# F  Design Choices

In this section, we provide the rationale behind which metrics we report for each section.

### F.1 Disentanglement Experiments

In Appendix E, we motivate that disentangled uncertainty estimators for epistemic and aleatoric uncertainty should be decorrelated. To quantify this, we measure the Spearman rank correlation. The Spearman rank correlation can detect any monotonic dependency between two continuous variables. Hence, in this case it will allow us to detect whether the aleatoric uncertainties are always high when the epistemic uncertainties are high. We choose this over the Pearson correlation, as the Pearson correlation can only detect linear dependencies. Although we have seen in Fig. 1 that the dependencies are, in fact, often linear, we chose the Spearman correlation because it still works even if two estimators are beyond linear correlation.

### F.2 Epistemic Uncertainty Experiments

We use OOD detection as a prominent proxy task for epistemic uncertainty. In OOD detection, some samples are from the ID dataset, and some are OOD. In other words, the true uncertainty here surfaces in two classes, and we want the estimated uncertainty to be higher for OOD samples than for ID samples. Such cases of a binary outcome variable predicted with a continuous estimator in a given direction can be handled by the AUROC. It measures the probability that an OOD sample will have a higher estimated uncertainty than an ID sample.

### F.3 Aleatoric Uncertainty Experiments

In the aleatoric uncertainty experiments, we check whether the estimated uncertainty is predictive of the true aleatoric uncertainty. The estimated uncertainty is a continuous variable, as are the true aleatoric uncertainty values. Hence, we again need a correlation metric. We again decide on the Spearman rank correlation because it can detect non-linear, monotonic relationships, and there is no prior reason to believe that, e.g., the disagreement aggregator of ensembles as an uncertainty estimator should behave linearly the same as the aleatoric uncertainty ground-truths.

The aleatoric ground truths are, however, often equal to zero (when all annotators agree), and one can argue that the exact amount of disagreement between humans when it is non-zero is quite noisy. This is why we also report a binarized version of the aleatoric GT uncertainty in Appendix H.4. Now, the estimated uncertainty is continuous while the GT is binary (with a direction, i.e., that GT uncertain samples should have higher predicted uncertainties). So, like above, we measure the AUROC.

## G Main CIFAR-10 Experiments

This section mirrors Section 3 of the main paper on the CIFAR-10 dataset whenever they were not already studied in the main paper.

### G.1 Epistemic Uncertainty: Specialized Uncertainty Esimators detect OOD Inputs the Best

We use balanced mixtures of ID and OOD datasets to evaluate OOD detection performance as a proxy for epistemic uncertainty. OOD samples are perturbed ID samples with severity level two. The uncertainty estimators are tasked to predict which sample is OOD, i.e., OOD inputs should have higher uncertainty estimates. As for ImageNet in Section 3.2, the Mahalanobis method is by far the most performant in telling apart clean CIFAR-10 samples from perturbed ones. However, the ranking of the remaining methods is different from that of ImageNet.

### G.2 Aleatoric Uncertainty: No Method With Outstanding Performance

Let us now benchmark how much the estimators predict aleatoric uncertainties on CIFAR-10. Since we use the entropy of human annotator label distributions as ground truths, this could also be considered the alignment with human uncertainties. Fig. G.2 shows drastically different results from the ImageNet results in Section 3.3. Correctness prediction is most aligned *on average* with a notably small min-max error bar. This is reasonable since ID, the aleatoric uncertainty of the sample determines the network's correctness the most.[5] SWAG and Loss Prediction, methods that

---

[5]Note that we train with only one label per input.

performed worse than the CE baseline on ImageNet, now perform among the best. This again shows that rankings from CIFAR-10 are not very informative of ImageNet rankings, especially because the ground-truth soft-label distributions on CIFAR-10H and ImageNet-ReaL-H were collected with slightly different protocols.

### G.3 Predictive Uncertainty: Close to Saturation on CIFAR-10

Fig. G.3 shows that most uncertainty estimators perform within $\pm 0.02$ of the cross-entropy baseline when predicting correctness ID, and modern methods like HET-XL do not outperform older methods like deep ensembles. The best methods are saturated at very high AUROCs of $0.95$. The methods are also close to the CE baseline (but with a different ranking) when slightly altering the correctness metric to account for soft labels in Appendix H.1.

The saturation is even more pronounced on the abstained prediction task. All uncertainty methods apart from Mahalanobis obtain an AUC score greater than $0.99$ in Fig. G.4. Practically, this means that one can obtain a close-to-perfect classification accuracy by abstaining from prediction on a tiny set of samples. This is largely because the classification accuracy on CIFAR-10 is very high, so the AUAC, even of methods with worse ID uncertainty estimation like Mahalanobis, is close to perfect.

### G.4 Different Tasks Require Different Estimators

In the previous sections, we have hinted at the fact that the performance across methods is very similar on some tasks and dissimilar on others. In this section, we investigate the correlation among the previous practical tasks using a correlation matrix. To construct the matrix, we consider all benchmarked methods with all uncertainty aggregators (see Appendix D) and calculate the Pearson (Fig. G.5) and Spearman correlations (Fig. G.6) of performances on different metrics. Similar to ImageNet, we find consistent clusters between the Pearson and Spearman correlations. One similarity to ImageNet is also that the OOD cluster repels the other clusters. However, the aleatoric cluster on CIFAR-10H forms a unique cluster, whereas it was correlated with the Accuracy cluster on ImageNet. We believe this is due to the different ways the soft labels were obtained on ImageNet-ReaL and CIFAR-10H.

## H Further Practical Results

### H.1 Correctness Prediction

#### H.1.1 ImageNet

We show the correctness prediction performance of methods on OOD and mixed ID + OOD datasets in Fig. H.1. Like in the main paper, evidential deep learning methods, EDL and PostNet, dominate on the correctness prediction task across all severity levels and mixtures. The performance of Mahalanobis increases on mixed datasets as it can detect OOD samples well, which happen to also be incorrect more often.

#### H.1.2 CIFAR-10

We show the correctness prediction performance of methods on OOD and mixed ID + OOD datasets in Fig. H.2. We observe a consistent degradation of performance across methods on both dataset types.

### H.2 Performance Tendency for Increasing Severity

In Section 3.6 of the main paper, we show the performance of MC Dropout when going OOD, claiming that it is prototypical for other methods. Figs. H.3 and H.4 show for CIFAR-10 and ImageNet, respectively, that MC Dropout is not an outlier and other methods show very similar generalization capabilities. The only outlier is Mahalanobis, which has a bad correctness AUROC regardless of whether it is ID or OOD. In Figs. 7a, H.3 and H.4, we normalize the metrics using the formula $^{(\text{metric}-\text{rnd})}/_{(1-\text{rnd})}$ for direct comparability, where rnd is the base value that a random predictor achieves on that metric ($0.5$ for AUROC, $1/C$ for classification accuracy).

### H.3 OOD Detection

#### H.3.1 ImageNet

In Appendix G.1, we hint at the fact that nearly all methods show a steady increase in OOD detection performance as we increase the severity of the perturbed half of the dataset. Fig. H.5 shows how the performance of each method changes as we increase the severity level. We can see a steady increase in OOD detection performance for all methods. However, the specialized OOD detector, Mahalanobis, benefits less than the other methods. In particular, at severity level three, shallow and deep ensembles overtake Mahalanobis and remain the best estimators from then on. At severity levels four and five, Mahalanobis falls below the CE baseline. This may be because Mahalanobis was trained to detect samples at severity level two and cannot generalize as well to higher severity levels as the other methods.

#### H.3.2 CIFAR-10

Fig. H.6 shows the OOD detection performance of the benchmarked methods as we increase the severity level. We can see a steady increase in the performance for all methods. However, the relative distance of the specialized OOD detector, Mahalanobis, to the other methods shrinks. This could, however, be because it is the first to arrive at a higher, possibly saturated, performance.

### H.4 Ambiguous Input Detection

In this section, we consider an alternative task to evaluate the alignment of methods with aleatoric uncertainty. It is a binary prediction problem where the positive samples are ones with a non-deterministic GT soft label distribution, i.e., where annotators do not unanimously agree on the class of the sample. We evaluate the AUROC of uncertainty estimators on this binary task.

#### H.4.1 ImageNet

ImageNet ambiguous input detection results are shown in Fig. H.7. Method rankings are different from the continuous rankings in Fig. 3b. This indicates that, on ImageNet, the uncertainty estimators also differ in their performance on ranking within the ambiguous images.

#### H.4.2 CIFAR-10

Ambiguous input detection results on CIFAR-10 are shown in Fig. H.8. Interestingly, the ranking of methods is the same as when using rank correlations in Fig. G.2, except for MC Dropout. This could indicate that, on CIFAR-10H, most of the rankings in the rank correlations come from the methods telling apart samples with and without uncertainty, and that the ranking within the uncertain images does not make too much of a difference, or that none of the benchmarked methods (except possibly MC Dropout) is much better than the others in this.

### H.5 Abstained Prediction Results on the rAULC and E-AURC Metrics

In this section, we evaluate the benchmarked methods on the rAULC and E-AURC abstained prediction metrics that normalize the AUAC by the accuracy of the underlying model [43, 14].

#### H.5.1 ImageNet

ImageNet rAULC and E-AURC results are shown in Fig. H.9. Even though both metrics are normalized by the underlying model's accuracy, this normalization is done in different ways, and the rankings of methods are quite different.

#### H.5.2 CIFAR-10

Fig. H.10 shows CIFAR-10 rAULC and E-AURC results for abstained prediction. The ranking is similar to that of AUROC and non-normalized AUAC, as foreshadowed by the correlation matrix in Fig. G.5.

### H.6 Log Probability Proper Scoring Rule for Correctness Prediction

#### H.6.1 ImageNet

Results on the log probability proper scoring rule are shown in Fig. H.11. The EDL method consistently outperforms all other methods across all severity levels. The performance of all methods stays roughly constant as we increase the severity.

#### H.6.2 CIFAR-10

In Fig. H.12, we present the methods' results on the log probability proper scoring rule considering the CIFAR-10 dataset. We find that the Deep Ensemble consistently outperforms the others ID and on OOD in all but one scenario.

### H.7 Correlation Matrices of Metrics on ImageNet

All correlation matrices in this paper are constructed using the following procedure. First, the results of all methods on all metrics are collected. We consider three aggregators for all methods: $\max \bar{\pi}$, $\max \tilde{\pi}$, and $\mathbb{E}\left[\max \pi\right]$, using the notation introduced in Appendix D.5. These are the only aggregators that are restricted to the $[0, 1]$ interval, which is needed for the ECE metric and the proper scoring rules for correctness prediction. On a particular metric, each (method, aggregator) pair gives rise to one score. For each metric pair, we calculate the Pearson and Spearman correlations between the corresponding rankings of (method, aggregator) pairs.

Fig. 6 of the main paper shows Pearson correlation results on ImageNet among eight metrics, showcasing two clusters of metrics. Fig. H.13 shows the Spearman correlation results. The results are stable across these different metrics.

To give a more comprehensive overview of metric correlations, Fig. H.14 shows an extended correlation matrix. This matrix contains further abstinence and aleatoric uncertainty metrics, coupled with proper scoring rules for the models' predictions. Most of the added metrics fall into the bottom-right cluster. Interestingly, even though the rAULC and E-AURC metrics both evaluate abstained prediction performance, only the latter is highly correlated with the AUAC metric. The rAULC metric is more aligned with correctness prediction (corr. $= 0.98$) and the top-left cluster. The rank correlation results are, again, similar but more pronounced; see Fig. H.15.

### H.8 Correlations of Rankings Between Datasets

Table H.1 shows the correlation of rankings on CIFAR-10 and ImageNet. Nine of thirteen metrics have substantially different rankings (rank. corr $< 0.5$). On nine of the thirteen metrics, the best-performing method also differs between the two datasets, indicating that performance on CIFAR-10 should not be taken as an estimate for ImageNet performance.

### H.9 Training on Different Training Dataset Fractions of CIFAR-10

This subsection shows how the uncertainties reported by different distributional methods change as we vary the training dataset size. In particular, we train on $\{10\%, 50\%, 100\%\}$ of CIFAR-10 from scratch and evaluate the trained models on the CIFAR-10 test set.

Table H.2 shows that for the majority of the benchmarked methods, both the mean epistemic and aleatoric uncertainties of the IT decomposition decrease monotonically with an increasing training dataset size. Table H.3 shows the results using the estimators of Kendall and Gal [23], and Table H.4 shows the results of the Bregman decomposition. Fig. H.16 visualizes this tendency for the EDL method but also highlights the extreme correlation (0.9993) between the aleatoric and epistemic estimates.

Fig. H.17 shows that the severe rank correlations between the aleatoric and epistemic estimates remain when using the formulation of Kendall and Gal [23].

Table H.1: Rank correlations of method rankings on different metrics for all combinations of methods and aggregators between CIFAR-10 and ImageNet. The rankings of approaches are considerably different between these two datasets.

| Metric | Rank Corr. CIFAR-10 vs. ImageNet |
|---|---|
| Correctness AUROC | 0.503 |
| ECE | 0.659 |
| Correctness Brier | 0.193 |
| Correctness Log Prob. | 0.445 |
| rAULC | 0.554 |
| E-AURC | 0.482 |
| AUAC | 0.581 |
| Accuracy | 0.263 |
| Aleatoric Log Prob. | 0.484 |
| Aleatoric Brier | 0.426 |
| Aleatoric Rank Corr. | 0.013 |
| Aleatoric AUROC | 0.290 |
| OOD AUROC | 0.368 |

Table H.2: Aleatoric and epistemic uncertainty estimates of the IT decomposition for different methods and reduced training dataset fractions of CIFAR-10. We report the mean estimates over the test dataset.

| | Aleatoric | | | | | | | | | | | | Epistemic | | | | | | | | | | | |
|---|---|---|---|---|---|---|---|---|---|---|---|---|---|---|---|---|---|---|---|---|---|---|---|---|
| | Deep | EDL | GP | HET | HETClassNN | HET-XL | Laplace | MC Dropout | PostNet | Shallow | SNGP | SWAG | Deep | EDL | GP | HET | HETClassNN | HET-XL | Laplace | MC Dropout | PostNet | Shallow | SNGP | SWAG |
| 10% | 0.235 | 1.701 | 0.208 | 0.200 | 0.184 | 0.214 | 0.272 | 0.209 | 0.480 | 0.228 | 0.201 | 0.418 | 0.161 | 0.296 | 0.015 | 0.004 | 0.109 | 0.014 | 0.181 | 0.103 | 0.008 | 0.001 | 0.016 | 0.285 |
| 50% | 0.103 | 0.869 | 0.082 | 0.108 | 0.082 | 0.104 | 0.142 | 0.104 | 0.109 | 0.093 | 0.082 | 0.408 | 0.062 | 0.106 | 0.001 | 0.001 | 0.047 | 0.007 | 0.114 | 0.046 | 0.001 | 0.000 | 0.001 | 0.348 |
| 100% | 0.084 | 0.716 | 0.066 | 0.088 | 0.080 | 0.079 | 0.107 | 0.055 | 0.118 | 0.076 | 0.058 | 0.056 | 0.037 | 0.083 | 0.000 | 0.001 | 0.034 | 0.004 | 0.070 | 0.037 | 0.001 | 0.000 | 0.000 | 0.033 |

# I  Training and Implementation Details

For both datasets, we train and evaluate on an NVIDIA GeForce RTX 2080 Ti GPU. We only use an NVIDIA A100 Tensor Core GPU for the construction of the Laplace approximation on ImageNet, owing to the VRAM requirements of this method. Our code is based on the timm library [56], extended with a wide range of uncertainty quantification methods, evaluation metrics, ImageNet-C and CIFAR-10C corruptions, and general soft label support. The exact hyperparameter settings for each method are available in the README.md file of our published GitHub repository.

## I.1  CIFAR-10

For CIFAR-10, we follow the augmentations and training schedules of the uncertainty_baselines GitHub repository [38]. In particular, for non-post-hoc methods, we train a Wide ResNet 28-10 [61] for 200 epochs with a step decay schedule at $[60, 120, 160]$ epochs with decay rate 0.2 and one learning rate warmup epoch. The only exceptions are the SNGP-variants and the DDU method (250 epochs with a step decay schedule at $[75, 150, 200]$ epochs) [32, 37]. We use stochastic gradient descent with momentum 0.9 and a batch size of 128. Our training augmentation comprises a random crop using padding 2 and a random flip on the vertical axis with probability 0.5. We use $2/5$th of the CIFAR-10 test dataset as the validation split and the rest as the test split. The learning rate and weight decay hyperparameters are chosen by ten iterations of the Bayesian optimization scheme of Weights & Biases [3] based on the correctness prediction AUROC metric on the validation split. The additional hyperparameters of benchmarked methods are determined by either using values suggested by the original authors or including these in the hyperparameter sweep.

## I.2  ImageNet

On ImageNet, we fine-tune a pretrained ResNet 50 [20] using the `resnet50.a1_in1k` parameters from the timm library as initialization. We fine-tune for 50 epochs following a cosine learning rate schedule [33] using the LAMB optimizer [60] and a learning rate warmup period of 5 epochs. We use a batch size of 128 with 16 accumulation steps, resulting in an effective batch size of 2048,

Table H.3: Aleatoric and epistemic uncertainty estimates of the formulation of Kendall and Gal for different methods and reduced training dataset fractions of CIFAR-10. This formulation uses the expected variance of the probability vectors as epistemic and the expected entropy as aleatoric uncertainty. We report the mean estimates over the test dataset.

| | Aleatoric | | | | | | | | | | | | Epistemic | | | | | | | | | | | |
|---|---|---|---|---|---|---|---|---|---|---|---|---|---|---|---|---|---|---|---|---|---|---|---|---|
| | Deep | EDL | GP | HET | HetClassNN | HET-XL | Laplace | MC Dropout | PostNet | Shallow | SNGP | SWAG | Deep | EDL | GP | HET | HetClassNN | HET-XL | Laplace | MC Dropout | PostNet | Shallow | SNGP | SWAG |
| 10% | 0.235 | 1.701 | 0.208 | 0.200 | 0.184 | 0.214 | 0.272 | 0.209 | 0.480 | 0.228 | 0.201 | 0.418 | 1.064e-2 | 6.156e-3 | 8.322e-4 | 1.635e-4 | 6.030e-3 | 6.538e-4 | 7.133e-3 | 5.594e-3 | 8.382e-5 | 3.636e-5 | 8.725e-4 | 1.342e-2 |
| 50% | 0.103 | 0.869 | 0.082 | 0.108 | 0.082 | 0.104 | 0.142 | 0.104 | 0.109 | 0.093 | 0.082 | 0.408 | 4.234e-3 | 1.569e-3 | 6.243e-5 | 6.146e-5 | 2.606e-3 | 3.501e-4 | 4.185e-3 | 2.519e-3 | 1.033e-5 | 1.300e-7 | 6.033e-5 | 1.492e-2 |
| 100% | 0.084 | 0.716 | 0.066 | 0.088 | 0.080 | 0.079 | 0.107 | 0.055 | 0.118 | 0.076 | 0.058 | 0.056 | 2.484e-3 | 1.139e-3 | 2.261e-5 | 5.595e-5 | 1.855e-3 | 2.139e-4 | 2.885e-3 | 2.097e-3 | 7.470e-6 | 1.000e-7 | 2.020e-5 | 1.743e-3 |

Table H.4: Aleatoric and epistemic uncertainty estimates of the Bregman decomposition for different methods and reduced training dataset fractions of CIFAR-10. We report the mean estimates over the test dataset.

| | Aleatoric | | | | | | | | | | | | Epistemic | | | | | | | | | | | |
|---|---|---|---|---|---|---|---|---|---|---|---|---|---|---|---|---|---|---|---|---|---|---|---|---|
| | Deep | EDL | GP | HET | HetClassNN | HET-XL | Laplace | MC Dropout | PostNet | Shallow | SNGP | SWAG | Deep | EDL | GP | HET | HetClassNN | HET-XL | Laplace | MC Dropout | PostNet | Shallow | SNGP | SWAG |
| 10% | 0.258 | 0.403 | 1.589e-2 | 4.016e-3 | 0.146 | 1.408e-2 | 0.211 | 0.131 | 8.497e-3 | 7.965e-4 | 1.642e-2 | 0.379 | 0.235 | 1.701 | 0.208 | 0.200 | 0.184 | 0.214 | 0.272 | 0.209 | 0.480 | 0.228 | 0.201 | 0.418 |
| 50% | 0.093 | 0.129 | 1.130e-3 | 1.283e-3 | 0.061 | 6.921e-3 | 0.123 | 0.057 | 1.537e-3 | 1.970e-6 | 1.086e-3 | 0.496 | 0.103 | 0.869 | 0.082 | 0.108 | 0.082 | 0.104 | 0.142 | 0.104 | 0.109 | 0.093 | 0.082 | 0.408 |
| 100% | 0.050 | 0.101 | 4.152e-4 | 1.071e-3 | 0.041 | 4.318e-3 | 0.077 | 0.050 | 1.476e-3 | 1.590e-6 | 3.735e-4 | 0.039 | 0.084 | 0.716 | 0.066 | 0.088 | 0.080 | 0.079 | 0.107 | 0.055 | 0.118 | 0.076 | 0.058 | 0.056 |

following Tran et al. [49]. The hyperparameters are chosen identically to those on CIFAR-10 (see Appendix I.1).

## I.3 Runtime

Table I.1 and Table I.2 show statistics of the per-epoch runtime for each method on ImageNet and CIFAR-10, respectively. As Laplace, Mahalanobis, temperature scaling, and deep ensemble are post-hoc methods, their reported time comprises the construction of the method and its evaluation on various ID and OOD test datasets.

# J Visualization of Images and Label Distributions

This section displays both easy (low human uncertainty) ImageNet samples in Fig. J.1 and hard (high human uncertainty) ones in Fig. J.2 using the ImageNet-ReaL labels and ImageNet-C perturbations.

Figs. J.3 and J.4 give summary statistics of the label distributions of ImageNet-ReaL and CIFAR-10H, respectively.

Table I.1: Summary of average per-batch forward times for the benchmarked methods on CIFAR-10. Note that the rankings for methods with similar forward times are subject to many external factors, such as the cluster node's state or the uptime of the allocated GPU. However, computationally costly methods consistently require more time. Methods are sorted by increasing mean per-batch forward time.

| Method | Mean (s) | Min (s) | Max (s) |
|---|---|---|---|
| Temperature | 0.0614 | 0.0605 | 0.0629 |
| EDL | 0.0626 | 0.0616 | 0.0634 |
| DUQ | 0.0627 | 0.0619 | 0.0635 |
| CE Baseline | 0.0628 | 0.0622 | 0.0635 |
| Corr. Pred. | 0.0631 | 0.0625 | 0.0639 |
| HET | 0.0635 | 0.0625 | 0.0640 |
| Shallow Ens. | 0.0636 | 0.0620 | 0.0668 |
| Loss Pred. | 0.0673 | 0.0619 | 0.0800 |
| Laplace | 0.0668 | 0.0664 | 0.0677 |
| SNGP | 0.0703 | 0.0665 | 0.0842 |
| GP | 0.0698 | 0.0625 | 0.0870 |
| HET-XL | 0.0728 | 0.0727 | 0.0730 |
| PostNet | 0.0838 | 0.0812 | 0.0882 |
| Mahalanobis | 0.7524 | 0.7215 | 0.8132 |
| DDU | 0.2038 | 0.1543 | 0.2685 |
| Deep Ens. | 0.3426 | 0.3426 | 0.3426 |
| HetClassNN | 1.9995 | 1.9938 | 2.0077 |
| MC Dropout | 2.0342 | 1.9832 | 2.1176 |
| SWAG | 3.2236 | 2.8773 | 4.3338 |

Table I.2: Summary of average per-batch forward times for the benchmarked methods on ImageNet. Note that the rankings for methods with similar forward times are subject to many external factors, such as the cluster node's state or the uptime of the allocated GPU. However, computationally costly methods consistently require more time. Methods are sorted by increasing mean per-batch forward time.

| Method | Mean (s) | Min (s) | Max (s) |
|---|---|---|---|
| HET | 0.0485 | 0.0464 | 0.0538 |
| HET-XL | 0.0576 | 0.0564 | 0.0596 |
| Loss Pred. | 0.0597 | 0.0585 | 0.0622 |
| Corr. Pred. | 0.0598 | 0.0591 | 0.0606 |
| CE Baseline | 0.0610 | 0.0592 | 0.0637 |
| Shallow Ens. | 0.0651 | 0.0590 | 0.0882 |
| EDL | 0.0764 | 0.0739 | 0.0790 |
| Temperature | 0.0765 | 0.0671 | 0.1039 |
| GP | 0.0818 | 0.0795 | 0.0836 |
| PostNet | 0.0846 | 0.0816 | 0.0942 |
| SNGP | 0.0924 | 0.0861 | 0.1168 |
| Laplace | 0.3933 | 0.3837 | 0.4005 |
| Deep Ens. | 0.4822 | 0.4822 | 0.4822 |
| Mahalanobis | 1.5409 | 1.4934 | 1.6099 |
| DDU | 2.3874 | 2.2113 | 2.5266 |
| MC Dropout | 2.1183 | 2.0455 | 2.3026 |
| HetClassNN | 2.2859 | 2.0830 | 2.4543 |
| SWAG | 2.9623 | 2.8433 | 3.1278 |

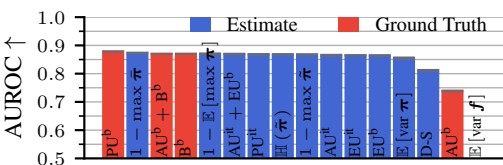

(a) HET-XL correctness AUROC on ID data.

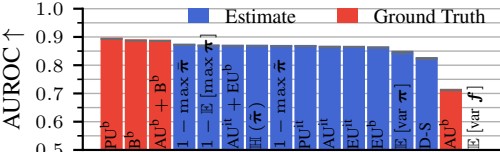

(b) HET-XL correctness AUROC on mixed ID and OOD data of severity level one.

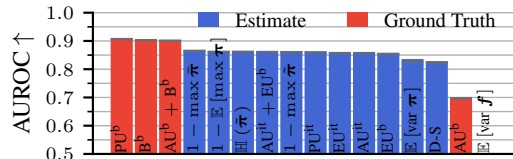

(c) HET-XL correctness AUROC on OOD data of severity level one.

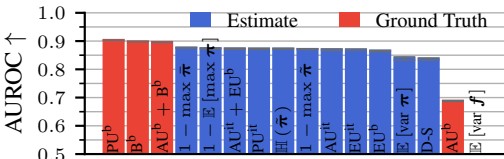

(d) HET-XL correctness AUROC on mixed ID and OOD data of severity level two.

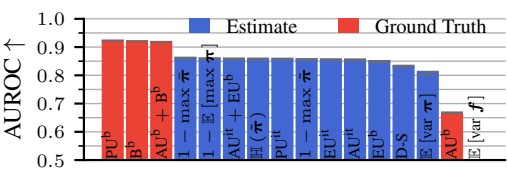

(e) HET-XL correctness AUROC on OOD data of severity level two.

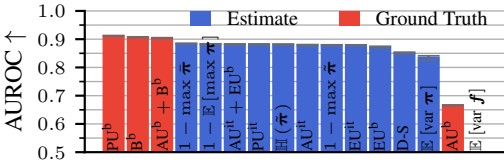

(f) HET-XL correctness AUROC on mixed ID and OOD data of severity level three.

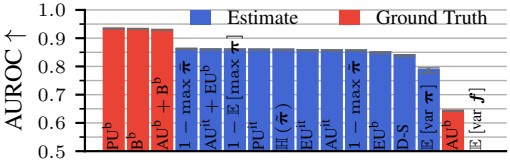

(g) HET-XL correctness AUROC on OOD data of severity level three.

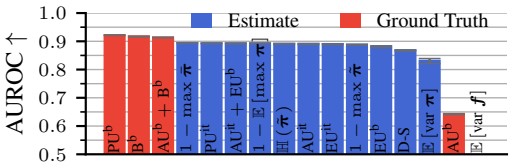

(h) HET-XL correctness AUROC on mixed ID and OOD data of severity level four.

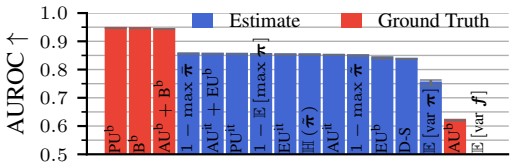

(i) HET-XL correctness AUROC on OOD data of severity level four.

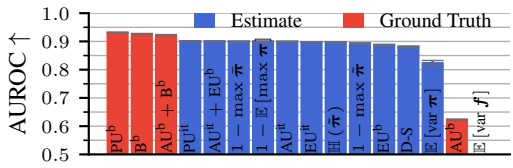

(j) HET-XL correctness AUROC on mixed ID and OOD data of severity level five.

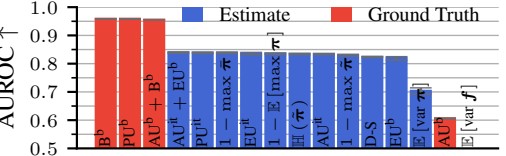

(k) HET-XL correctness AUROC on OOD data of severity level five.

Figure D.3: For the HET-XL method, the correctness prediction performance is saturated across aggregators on the ImageNet validation dataset. The disagreement-based epistemic aggregators, $EU^{it}$ and $EU^b$, are saturated among the other estimators, challenging the common assumption that epistemic aggregators perform poorly ID.

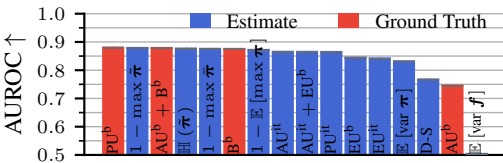

(a) Deep Ensemble correctness AUROC on ID data.

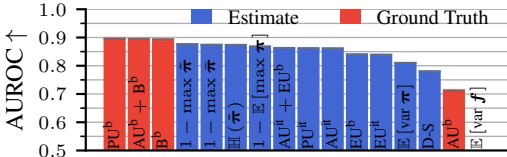

(b) Deep Ensemble correctness AUROC on mixed ID and OOD data of severity level one.

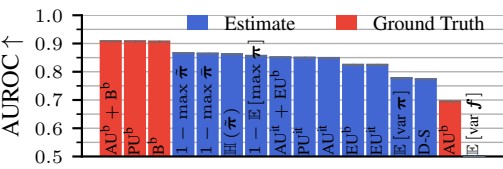

(c) Deep Ensemble correctness AUROC on OOD data of severity level one.

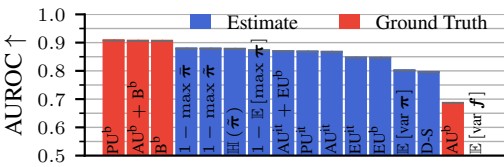

(d) Deep Ensemble correctness AUROC on mixed ID and OOD data of severity level two.

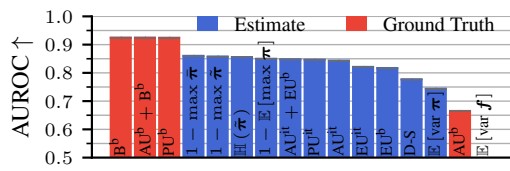

(e) Deep Ensemble correctness AUROC on OOD data of severity level two.

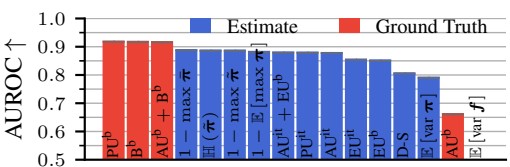

(f) Deep Ensemble correctness AUROC on mixed ID and OOD data of severity level three.

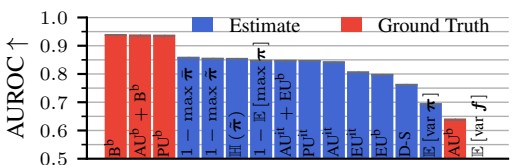

(g) Deep Ensemble correctness AUROC on OOD data of severity level three.

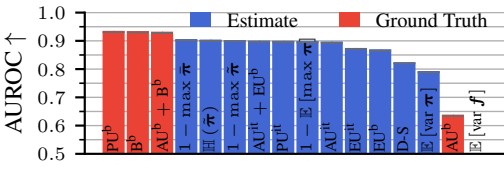

(h) Deep Ensemble correctness AUROC on mixed ID and OOD data of severity level four.

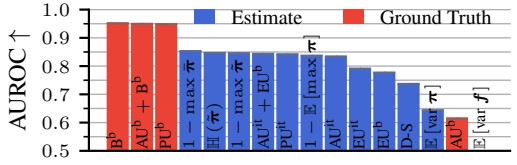

(i) Deep Ensemble correctness AUROC on OOD data of severity level four.

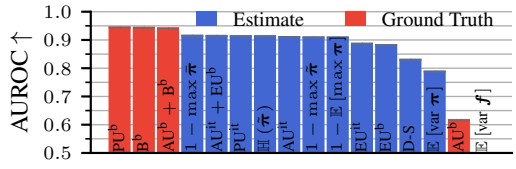

(j) Deep Ensemble correctness AUROC on mixed ID and OOD data of severity level five.

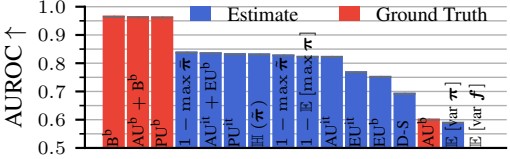

(k) Deep Ensemble correctness AUROC on OOD data of severity level five.

Figure D.4: For the Deep Ensemble method, the disagreement-based epistemic aggregators underperform the other aggregators on the correctness prediction task on the ImageNet validation set, which aligns with expectations.

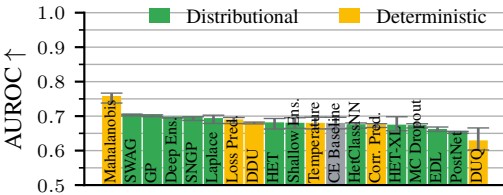

Figure G.1: OOD detection results on the CIFAR-10 test set, measured by the AUROC metric. OOD samples are perturbed by CIFAR-10C corruptions of severity level two. Mahalanobis, a method specially trained for this task, has an edge over the remaining methods.

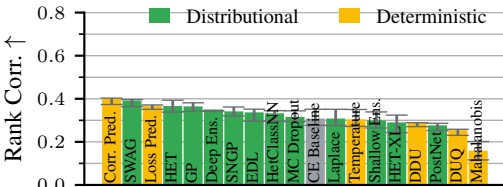

Figure G.2: Rank correlation with the soft input-conditional label distributions of CIFAR-10H corresponding to labeler votes.

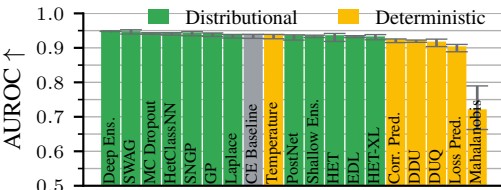

Figure G.3: On ID CIFAR-10 samples, the performance of methods on predicting correctness is close to saturated, although the more expensive distributional methods tend to perform better than the cheaper deterministic methods.

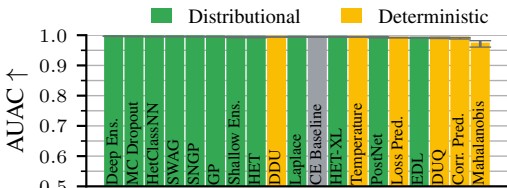

Figure G.4: ID abstained prediction results using the AUAC metric on the CIFAR-10 test dataset. On ID CIFAR-10 samples, most methods solve the abstinence task almost perfectly. This even holds for Mahalanobis. This is largely because the accuracy on CIFAR-10 is very high, and AUAC depends on it.

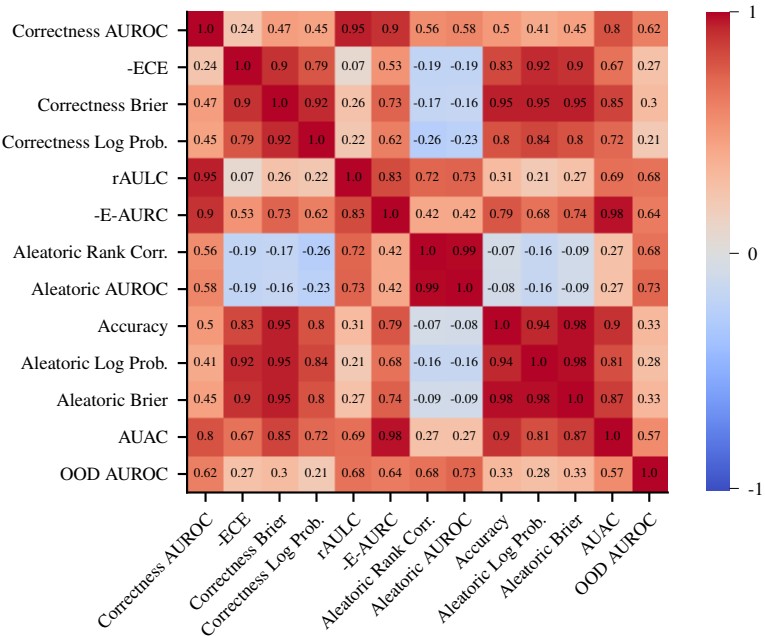

Figure G.5: Pearson correlation of metric pairs calculated over all (method, aggregator) pairs on CIFAR-10.

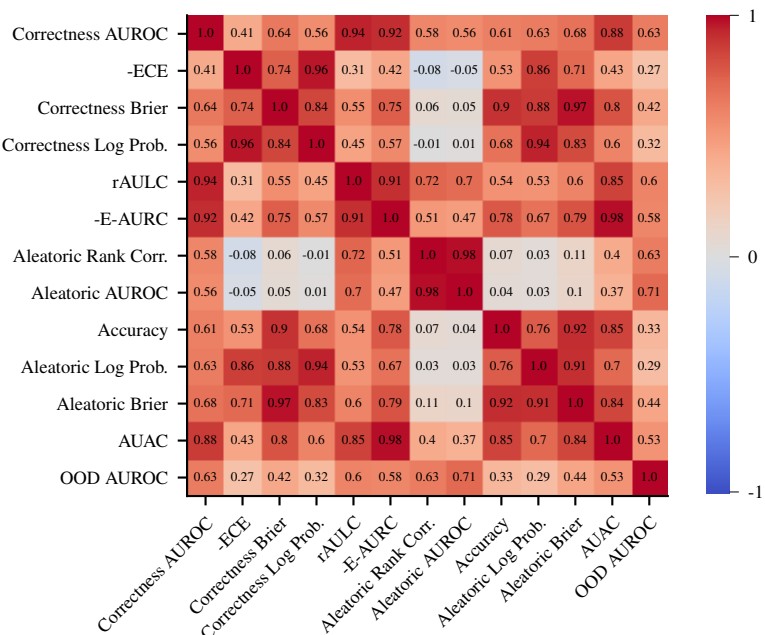

Figure G.6: Spearman rank correlation of metric pairs calculated over all (method, aggregator) pairs on CIFAR-10. The clusters are similar to those of the linear correlation in Fig. G.5, although there are more cases of higher correlations between the clusters. This discrepancy can be attributed to the saturation of methods on various metrics on CIFAR-10.

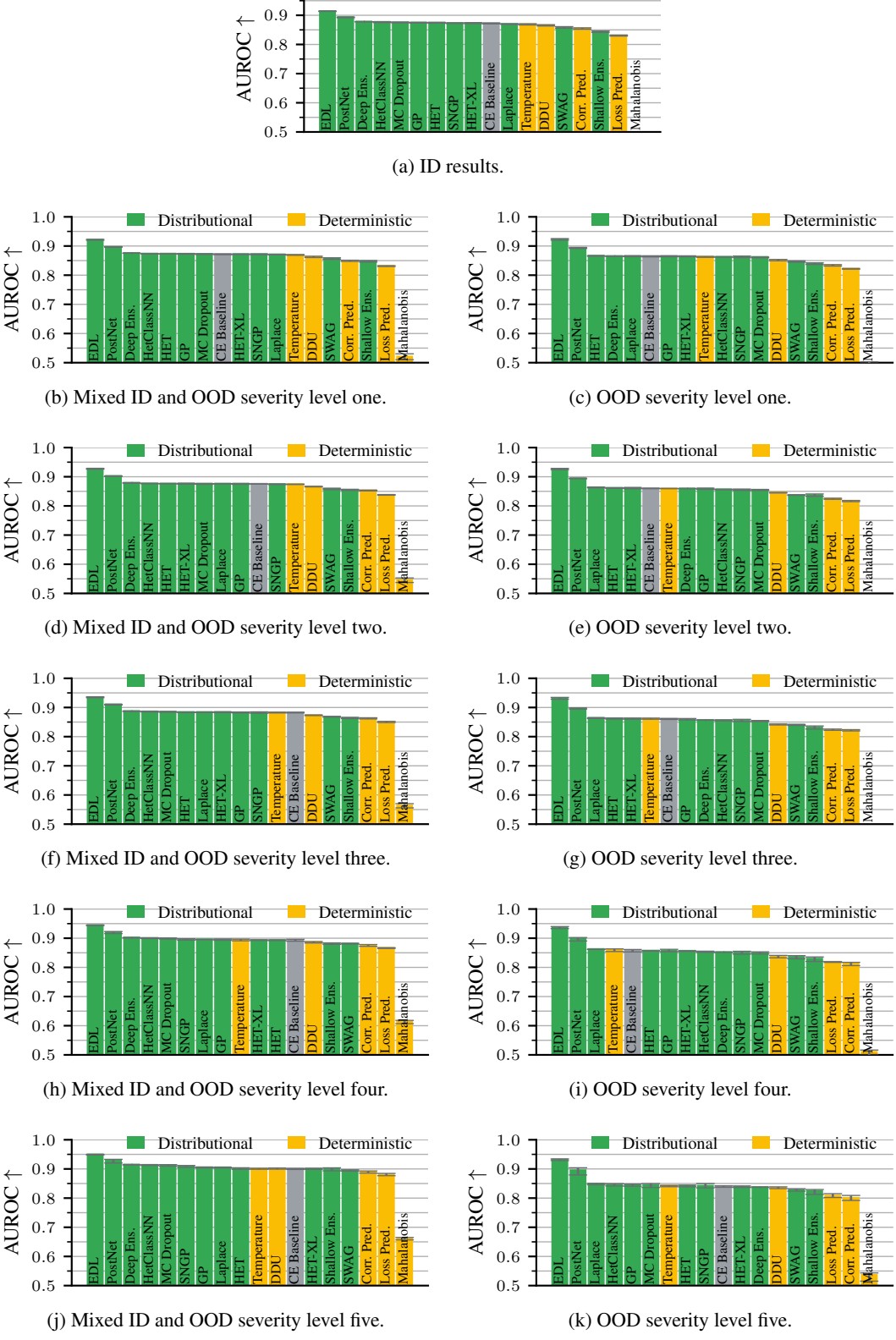

Figure H.1: On ImageNet, the evidential deep learning methods, EDL and PostNet, dominate on the correctness prediction task. The performance of Mahalanobis stably increases on mixed datasets, as models perform worse on OOD images than on ID ones, and it can detect OOD samples well.

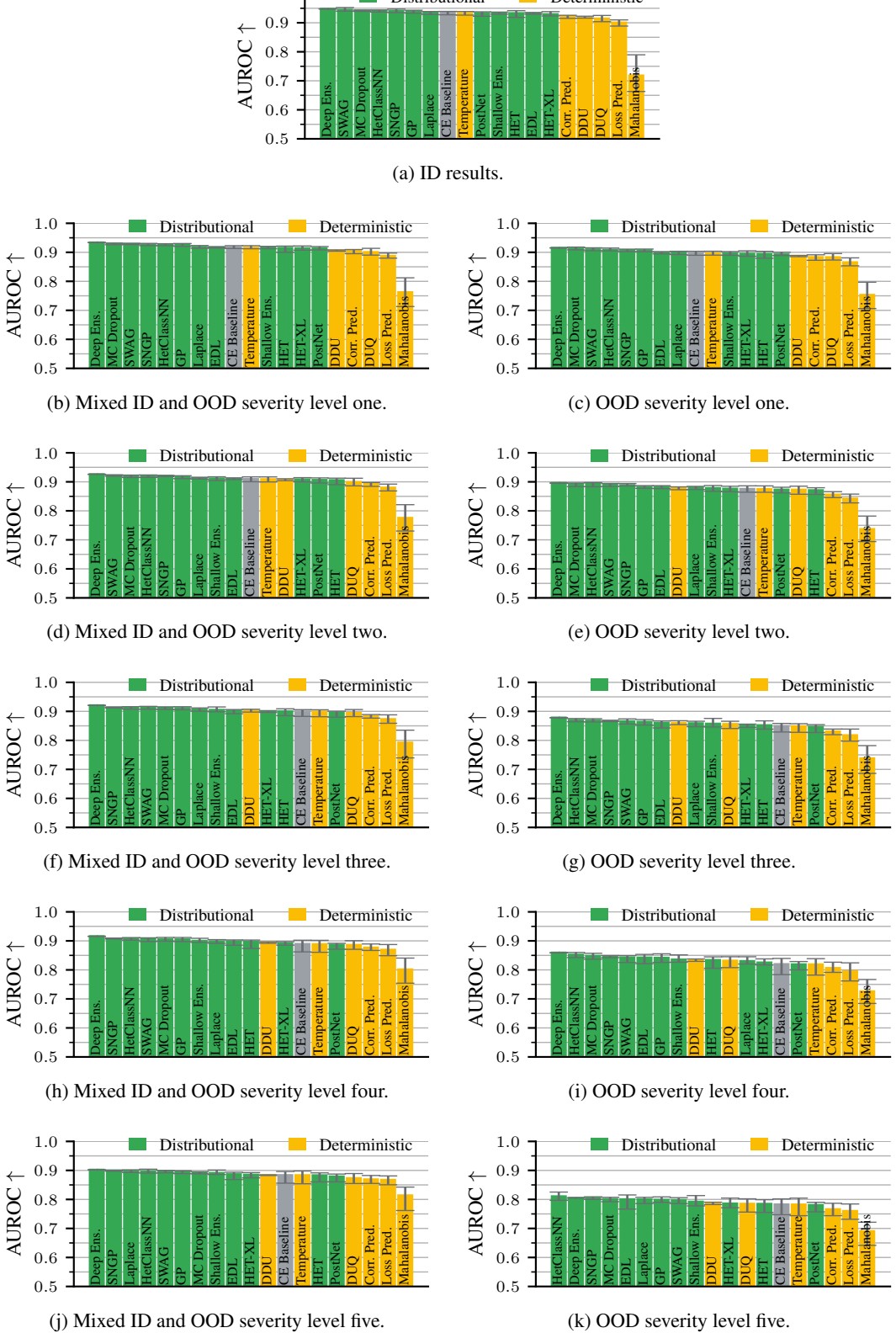

Figure H.2: On CIFAR-10, the performance of the methods consistently drops on the correctness prediction task, both on completely OOD datasets (right column) and on balanced mixtures of ID and OOD datasets (left column).

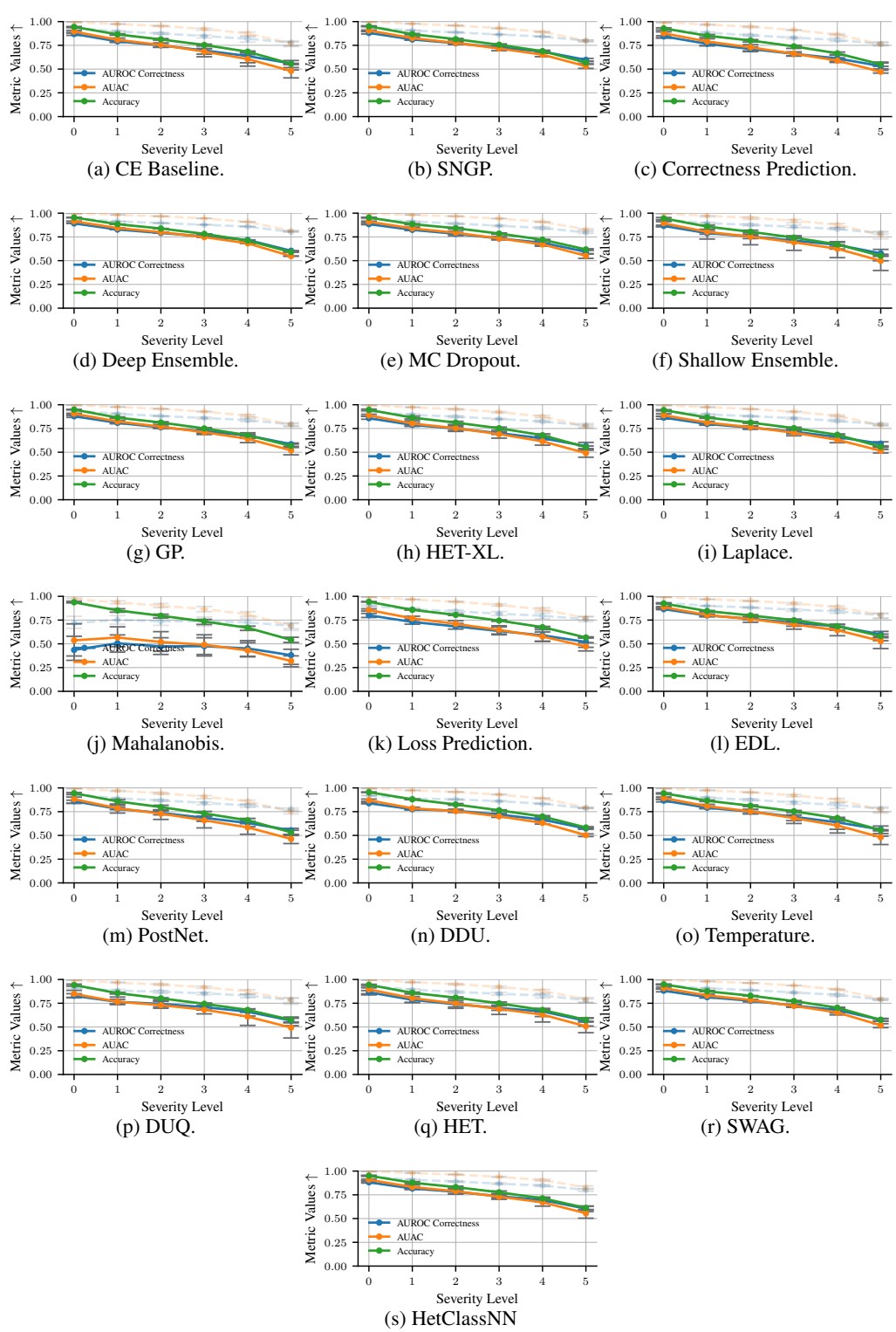

Figure H.3: On CIFAR-10, all methods' performance deteriorates at the same rate as the model's accuracy on the correctness and abstinence tasks. The only exception is Mahalanobis, which is a specialized OOD detector.

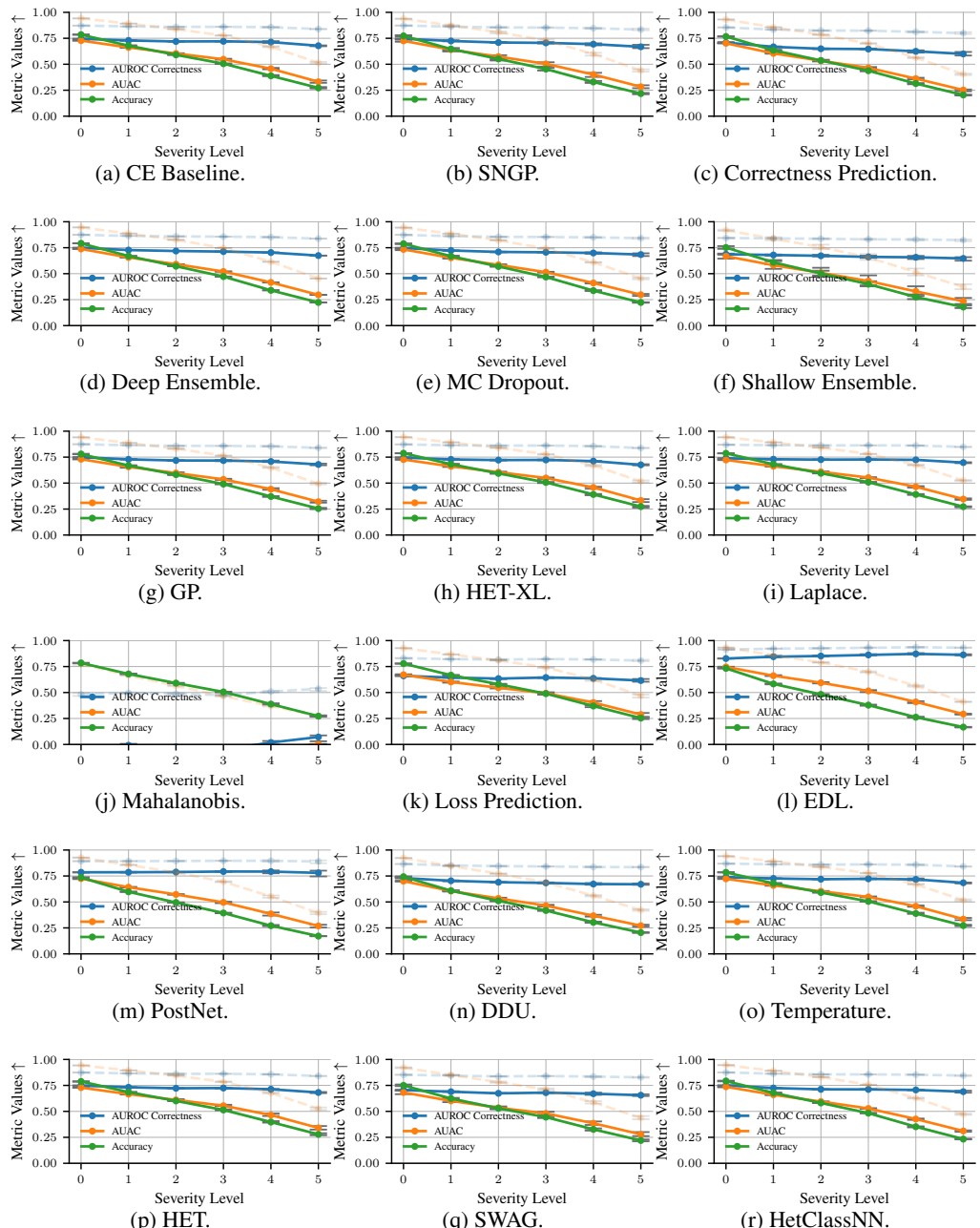

Figure H.4: On ImageNet, the estimate for predictive correctness is much more robust to OOD perturbations than the model's accuracy for all methods except Mahalanobis (a specialized OOD detector). The AUC abstinence score deteriorates at the same rate as the model's accuracy, which is an inherent property of the metric as the accuracy lower bounds the abstinence AUC metric.

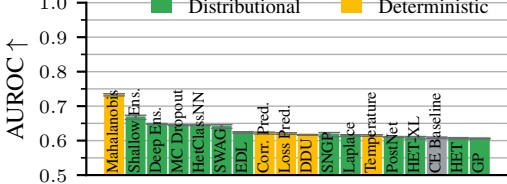

(a) OOD detection AUROC with severity level one.

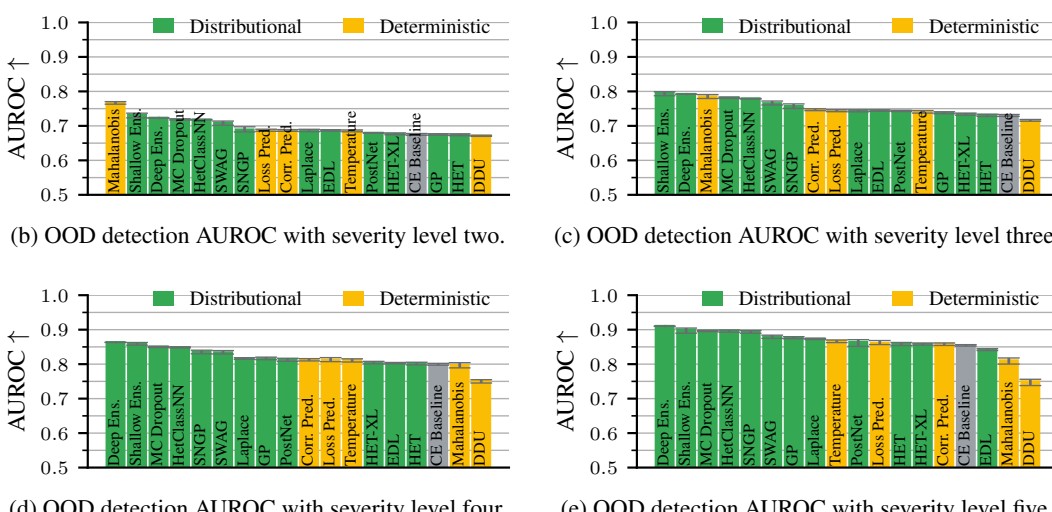

(b) OOD detection AUROC with severity level two.

(c) OOD detection AUROC with severity level three.

(d) OOD detection AUROC with severity level four.

(e) OOD detection AUROC with severity level five.

Figure H.5: The OOD detection performance of all methods increases steadily as we increase the severity of the perturbed half of the mixed dataset on the ImageNet validation dataset. However, the specialized OOD detector, Mahalanobis, generalizes worse than the other methods.

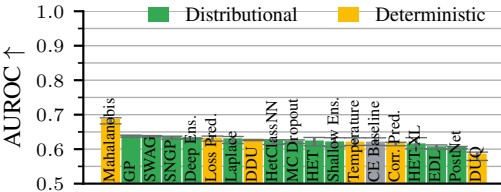

(a) OOD detection AUROC with severity level one.

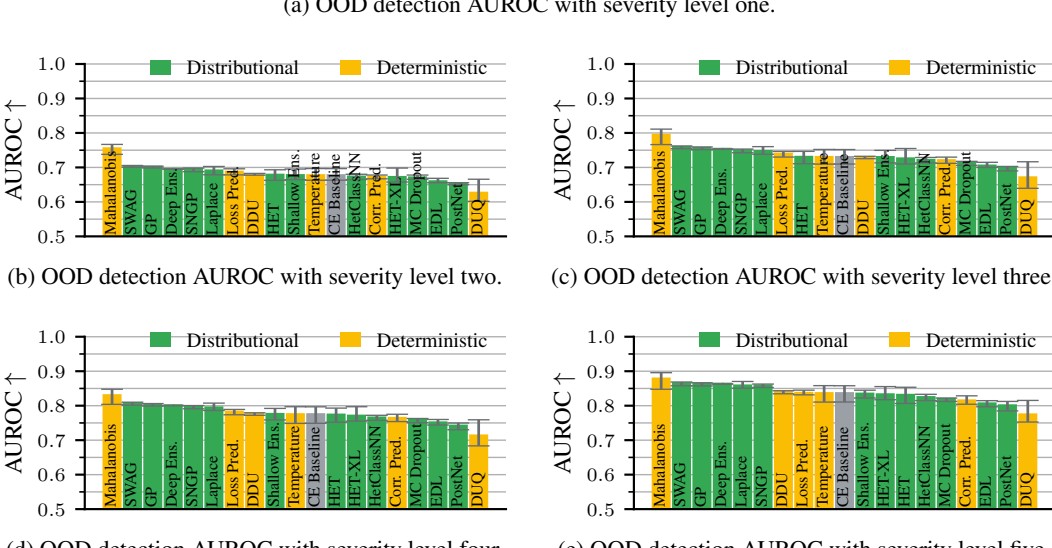

(b) OOD detection AUROC with severity level two.

(c) OOD detection AUROC with severity level three.

(d) OOD detection AUROC with severity level four.

(e) OOD detection AUROC with severity level five.

Figure H.6: On CIFAR-10, the OOD detection performance of all methods increases steadily as we increase the severity of the perturbed half of the mixed dataset. The relative distance between Mahalanobis and the other methods decreases with increasing severity level.

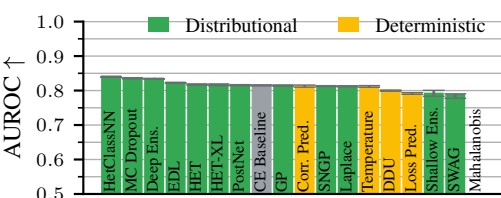

Figure H.7: ImageNet ambiguous input detection results. The ranking is considerably different from the ranking with the continuous ground truths in Fig. 3b.

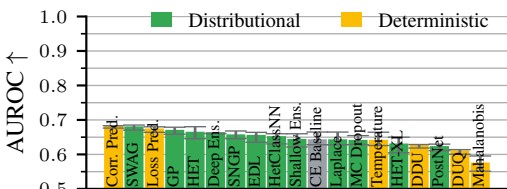

Figure H.8: CIFAR-10 ambiguous input detection results. The Correctness Prediction method is most aligned with human uncertainties. The methods are ranked the same as when using the rank correlation in Fig. G.2, except for MC Dropout.

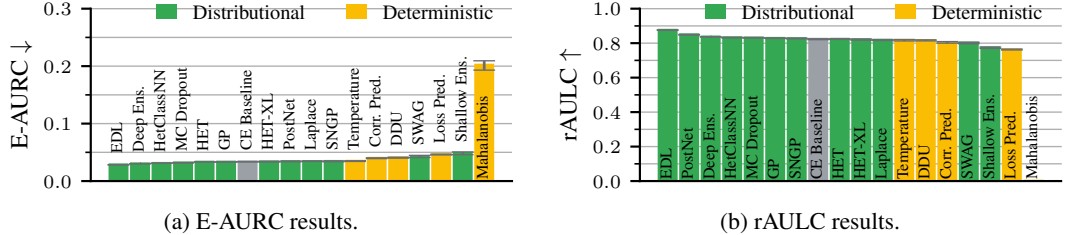

(a) E-AURC results.           (b) rAULC results.

Figure H.9: ImageNet abstained prediction evaluation on the E-AURC (left) and the rAULC (right) metric. Even though both metrics are normalized by the underlying model's accuracy, this normalization is done in different ways, and the rankings of methods are quite different, e.g., for PostNet and SNGP.

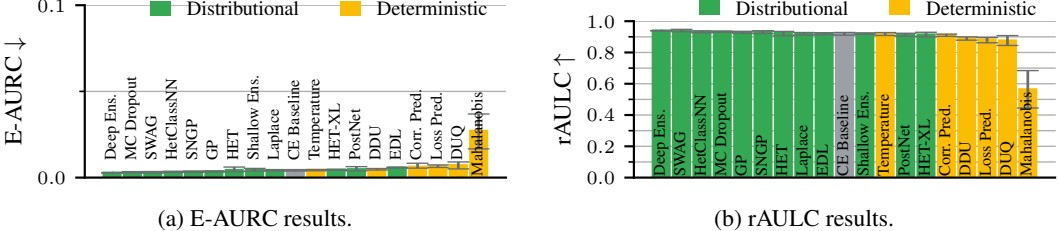

(a) E-AURC results.           (b) rAULC results.

Figure H.10: CIFAR-10 abstained prediction evaluation on the E-AURC (left) and the rAULC (right) metric. While deep ensembles are best according to both metrics, the rankings of methods are quite different.

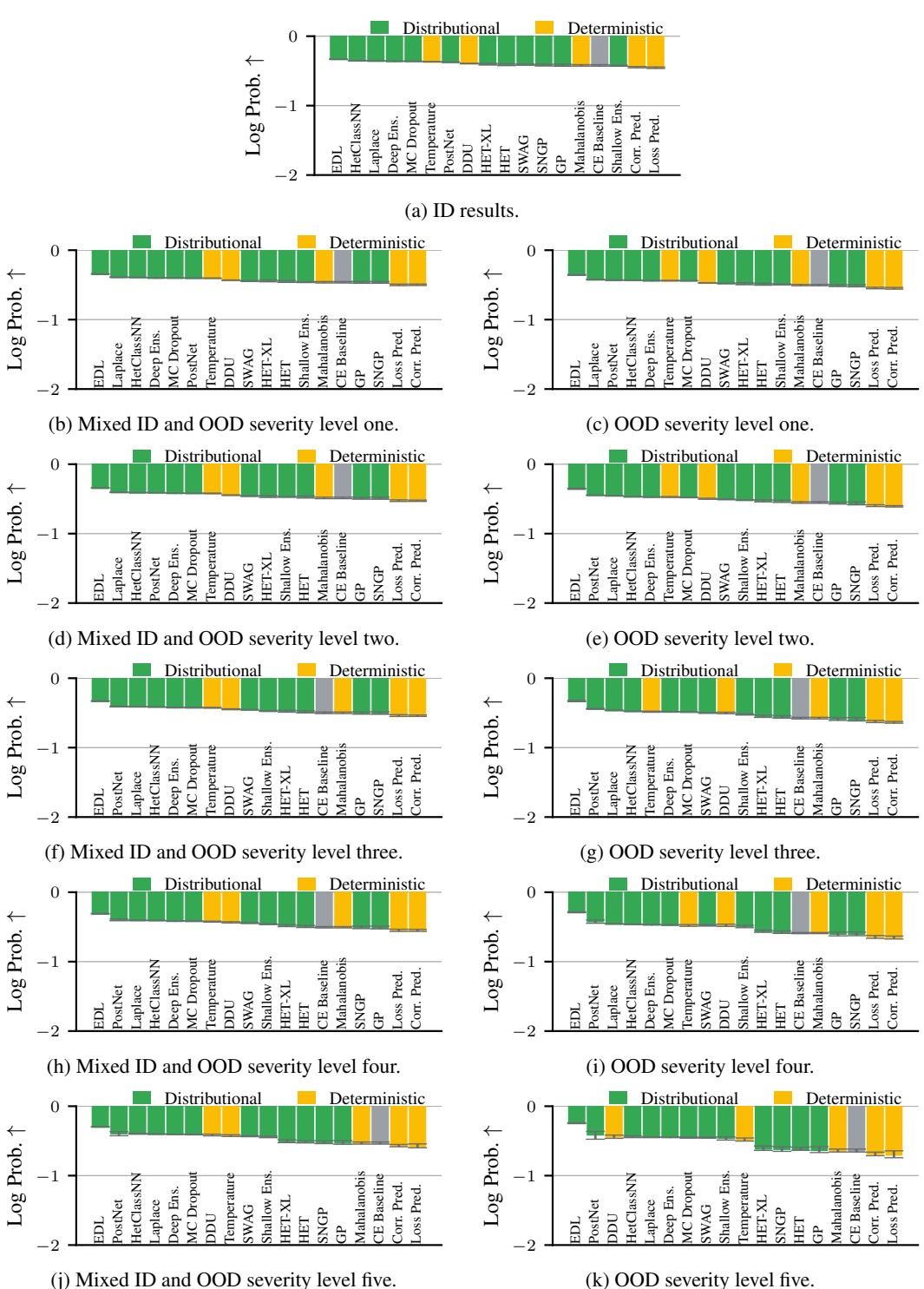

Figure H.11: On ImageNet, most methods consistently outperform the cross-entropy baseline on average, both ID and OOD for all severity levels when evaluating on the log probability proper scoring rule for correctness prediction. The EDL method performs best across all settings.

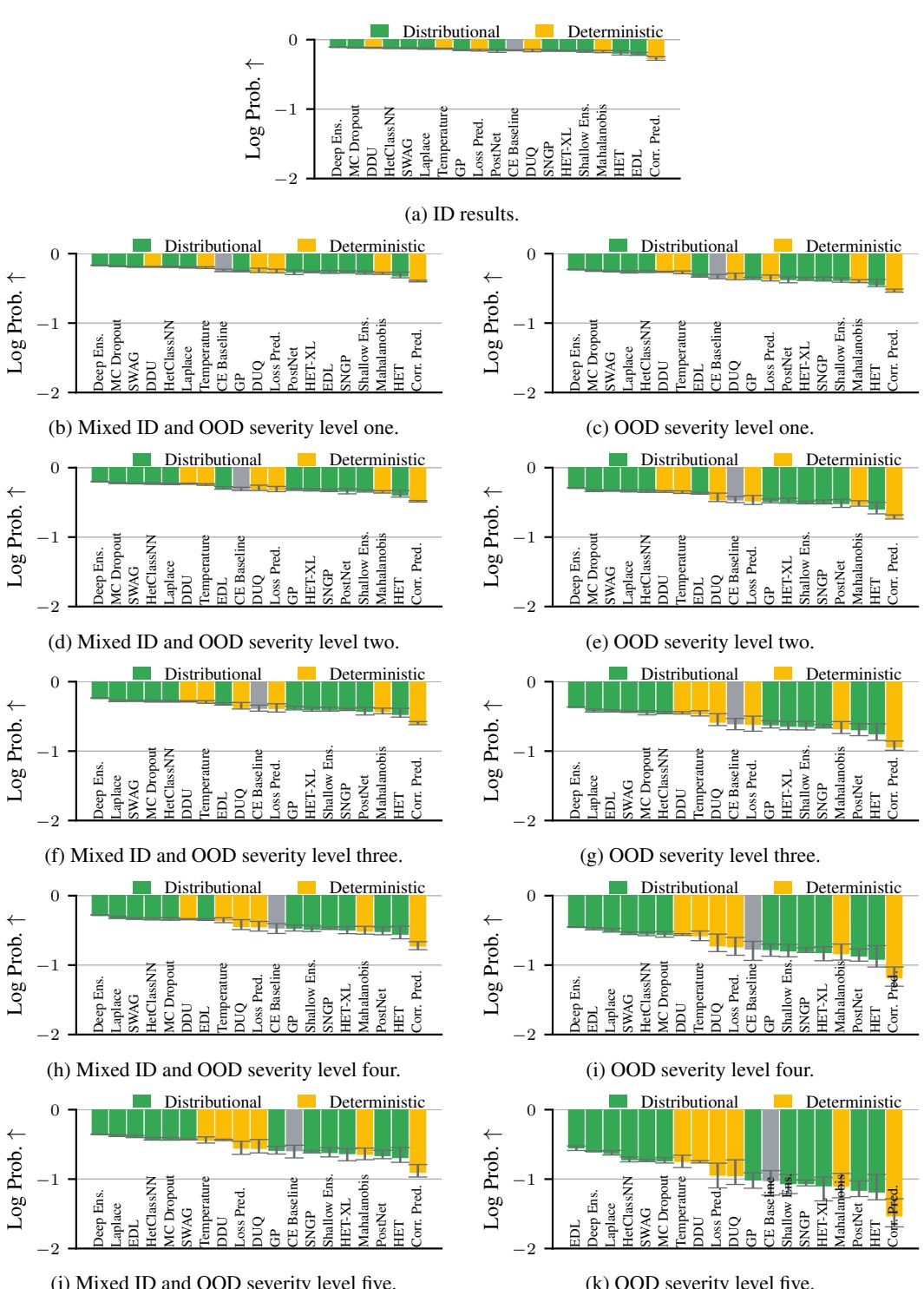

Figure H.12: On CIFAR-10, the deep ensemble is a consistently robust method both ID and OOD for all but one severity level when evaluating on the log probability proper scoring rule for correctness prediction.

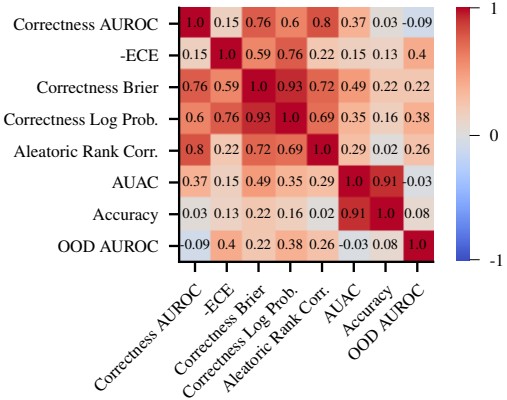

Figure H.13: Spearman correlation of metric pairs across all methods and aggregators on the ImageNet validation set. The correlations are similar to those of the linear correlation in Fig. 6. Only some of the considered metrics have a very high rank correlation among methods on the ImageNet validation dataset: most capture different aspects of uncertainty methods.

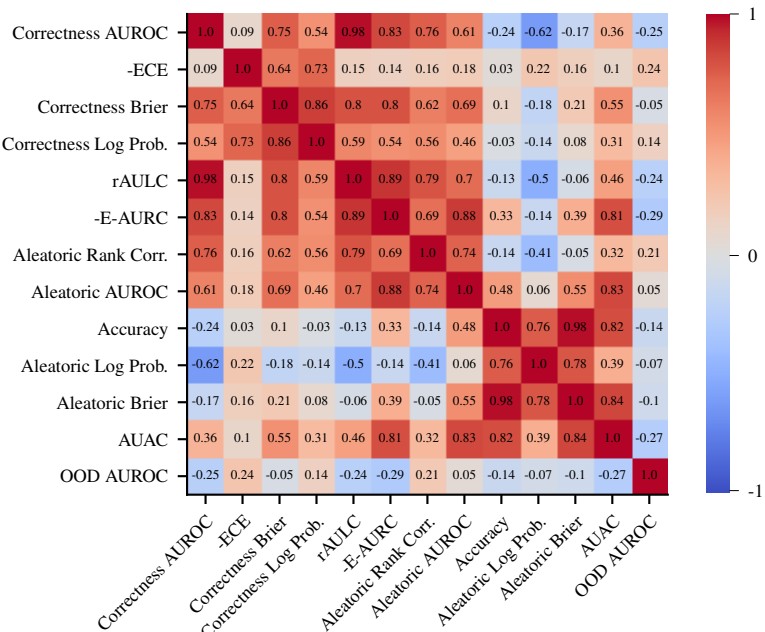

Figure H.14: Pearson correlation of metric pairs across all methods and aggregators on the ImageNet validation set. Extended version of Fig. 6.

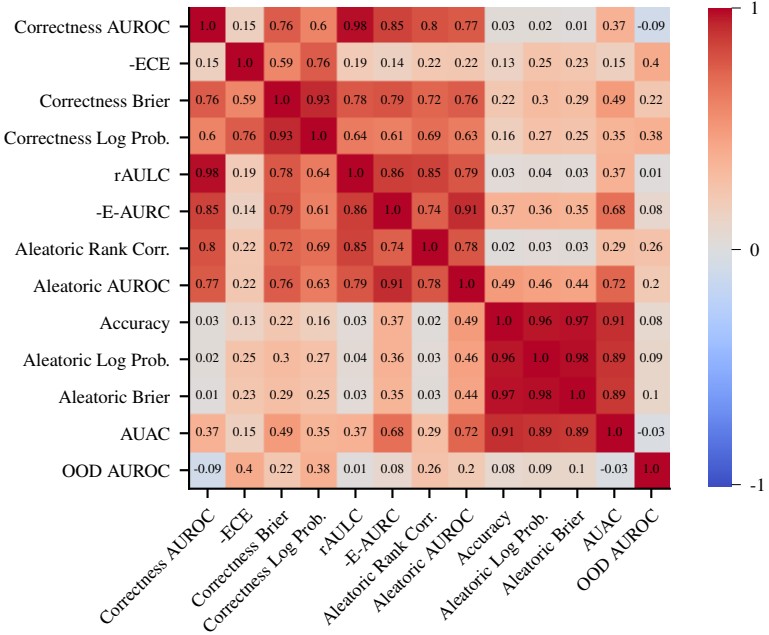

Figure H.15: Spearman correlation variant of Fig. H.14.

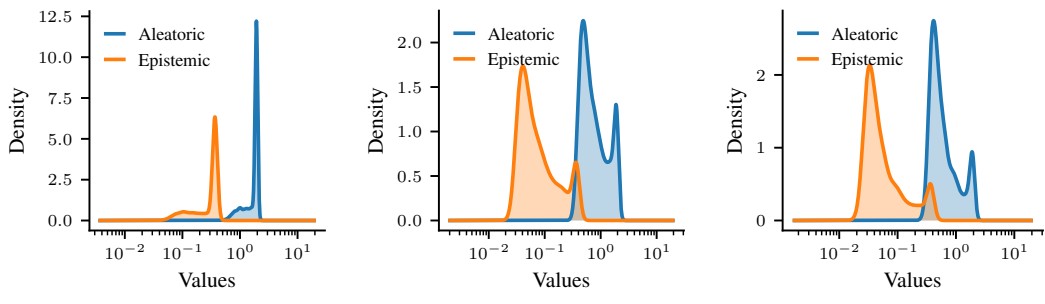

(a) Trained on 10% of CIFAR-10.    (b) Trained on 50% of CIFAR-10.    (c) Trained on 100% of CIFAR-10.

Figure H.16: Gaussian kernel density estimates of the EDL method's aleatoric and epistemic uncertainties of the IT decomposition when trained on different CIFAR-10 portions.

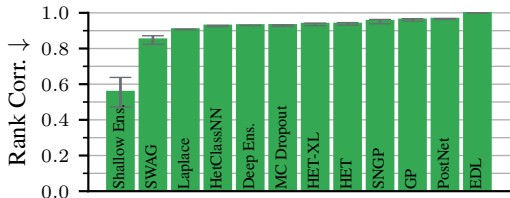

Figure H.17: Kendall and Gal Rank Correlation scores for different methods on ImageNet. This formulation uses the expected variance of the probability vectors as epistemic and the expected entropy as aleatoric uncertainty. We report the mean rank correlation across five seeds.

**Original Samples**          **Perturbed Samples**

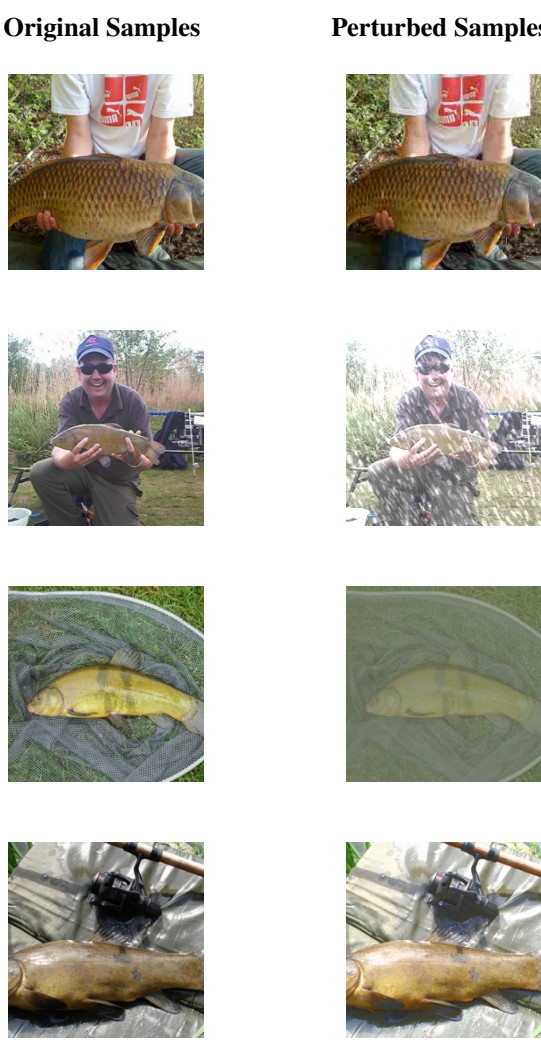

Figure J.1: Easy ImageNet-ReaL cases with no human disagreement on the labels. OOD samples are of severity two.

**Original Samples**  **Label Distributions**  **Perturbed Samples**

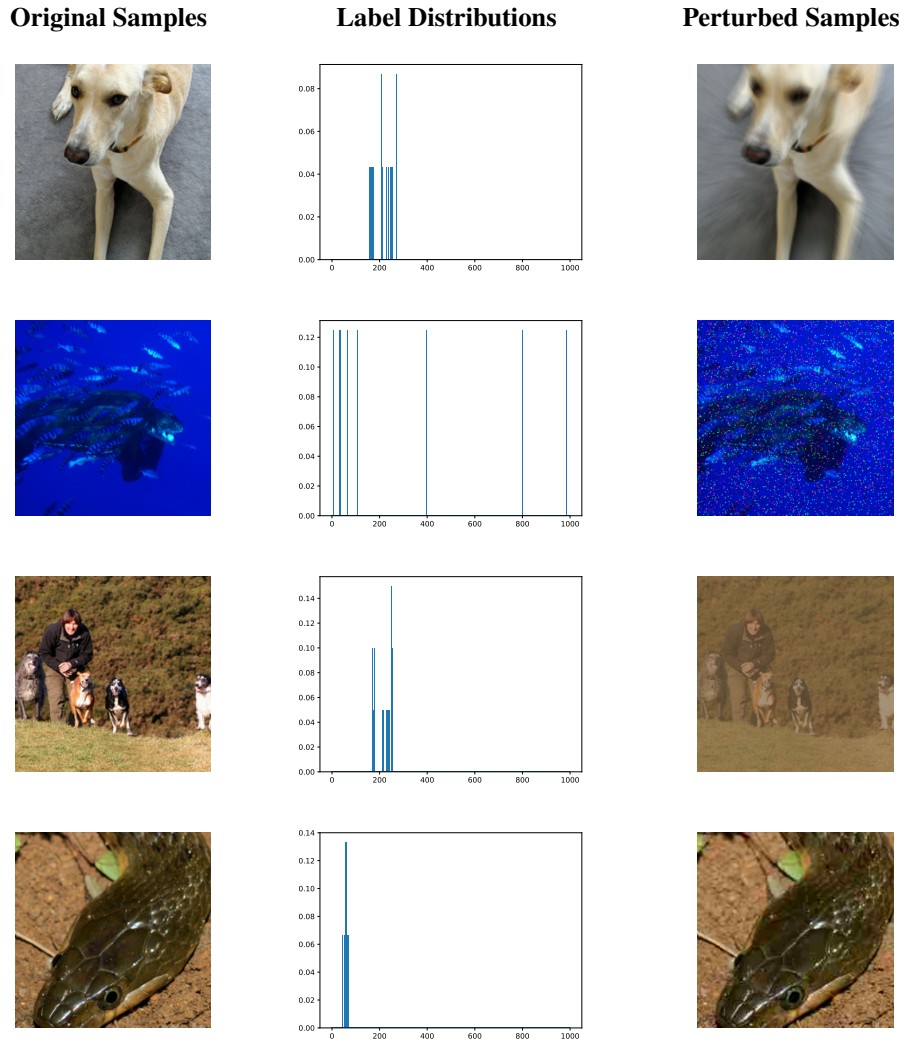

Figure J.2: Hard ImageNet-ReaL cases with high human uncertainty (i.e., high disagreement among annotators on the correct label). OOD samples are of severity two.

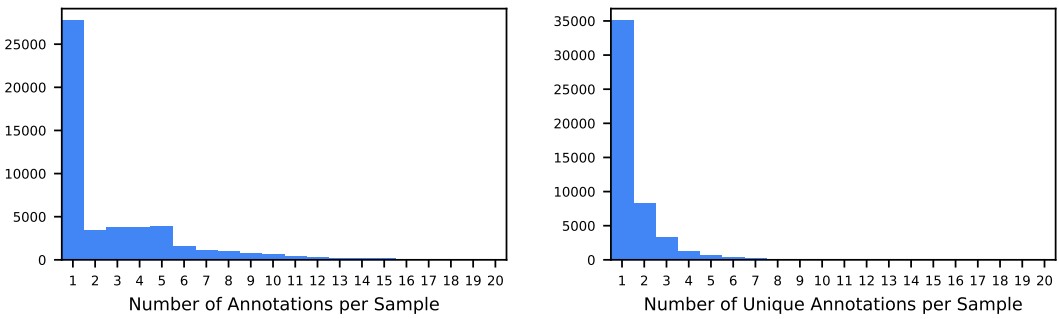

Figure J.3: Histograms of the label distributions of the ImageNet-ReaL validation set.

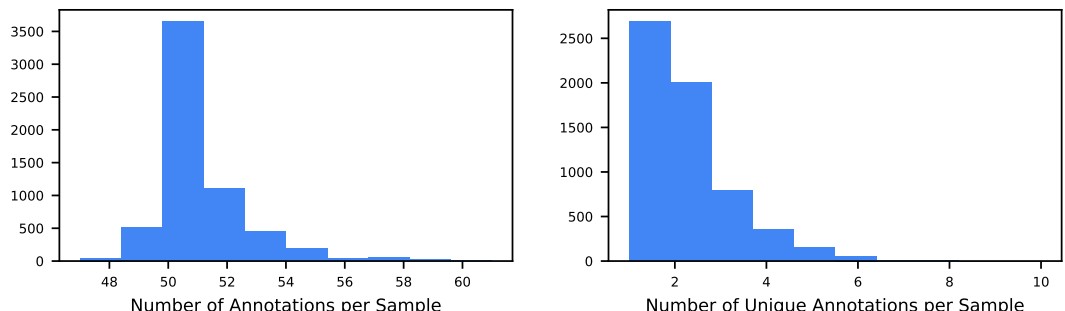

Figure J.4: Histograms of the label distributions of the CIFAR-10H validation set.

