# OpenReview forum: "Benchmarking Uncertainty Disentanglement: Specialized Uncertainties for Specialized Tasks"
_NeurIPS.cc/2024/Datasets_and_Benchmarks_Track — NeurIPS 2024 Track Datasets and Benchmarks Spotlight_

### Official Review · Reviewer_B7nH · 2024-07-24
**Informative benchmark for disentangled uncertainties in classification**

**Rating:** 7
**Confidence:** 4

**Review:**

- __Originality__: The focus on differentiating tasks that use aleatoric or epistemic is great. The use of image datasets and their variants (e.g., with corruption) is an interesting benchmark setting.
 - __Quality__: The quantity of methods considered as well as aggregators is useful for the analysis. Error bars were computed with repetitions for randomized learning methods. Considering both distributional method and deterministic methods is great.
- __Clarity__: Most of the papers are clear but the quantity of information (without even considering appendices...) can make it hard to follow. I found the titles of section 3 not particularly well phrases (seems like they are conclusions, acronyms could be explicit, and "OOD-ness" is not particularly nice to read). Having the Related work (Section 4) is relatively unusual I would move it earlier before diving into the results... Merging the Limitations and Conclusion sections is also a possibility, and splitting the conclusion into paragraphs.
- __Significance__: The overall benchmark is useful. The content of the main paper helps everyone understand the main conclusions of this work and the appendices can be interesting for a curious reader eager to dive into the details. The code is made available for reproducibility and extension of the benchmark.

**Strengths:**

The target problem "disentangled uncertainty" is a strength of this paper. The variety of UQ methods evaluated as well as aggregation strategies is an other good aspect of this work.

**Additional Feedback:**

N/A

**Clarity:**

- It is not clear to me as to why we look at **rank** correlation. Overall, the reading was quite challenging to follow even though the paper raises interesting questions.

- l. 155: "We measure via a binary classification AUROC if uncertainty estimates are higher on these OOD samples than on ID samples." Could you explain more about this? I understand that the binary classification is for example class 0 for an ID sample and 1 for an OOD sample. Is that correct? The sentence could be improved to clarify that.

- In the text, leaving "ground truth" instead of GT seems more clear to me.

- Figure 1: shows that Deep Ensemble is bad because high correlation between IT (information-theoretical) aleatoric and epistemic. Fig 2. a and b shows that Deep Ensembles are performing well on OOD and Ambiguous labels.... so what do we conclude?

- Section 3.5: Figure 4.b is not discussed when the results seems different compared to 4.a. Why is that?

**Correctness:**

- Fig. 1 and 2: I do not understand what the error bars mean. For example, Fig 2.a Mahalanobis, the top whisker aligns with the top of the bar. What are they (e.g., means, medians, quartiles, standard errors)?
- l. 165: "Laplace, the only potentially disentangled method, is the second-least able to detect epistemic uncertainties", on Fig 2.a, Laplace is placed 5th worst in AUROC.

**Documentation:**

The repository is relatively well documented (README, inline comments in the code) but could be improved.

**Limitations:**

Discussed by the authors but I would add that maybe the conclusion about small to large data (cifar10 to imagenet) is maybe more an observation cannot be yet generalized?

**Opportunities For Improvement:**

See my comments about clarity.

**Relation To Prior Work:**

The paper reviews some methods for uncertainty quantification. Conformal prediction methods are missing from the review. Are they applicable? For example, quantile-estimators are not mentioned either probably because of the classification setting.

**Summary And Contributions:**

The paper is about proposing a benchmark for uncertainty quantification disentanglement. The benchmark includes 17 UQ methods. The main finding of the benchmark is that no method disentangles aleatoric and epistemic UQ well. The benchmark is specific for classification and considered two image tasks ImageNet and Cifar10.

---

> ### Author Rebuttal · Authors · 2024-08-16
>
> Dear Reviewer,
>
> We greatly appreciate your acknowledgment of our benchmark. We address your remarks below.
>
> **L1: The conclusion about small to large data is more an observation that cannot be yet generalized.** The difference in the behavior of methods on CIFAR-10 and ImageNet is not a general claim but rather a note of caution. We are restricted to these two datasets because ImageNet-ReaL and CIFAR-10H are the only larger-scale datasets with multiple annotations per image, which are used for the aleatoric uncertainty evaluation. Once more such datasets become available, our proposed benchmark protocol can be directly applied to them. We have added this to our limitations section.
>
> **Corr. 1: Meaning of error bars.** The bar height corresponds to the mean performance of methods across five seeds, and the corresponding error bars show the minimum and maximum metric values, as mentioned in line 123 of our submission.
>
> **Corr. 2: Typo in line 165.** Thank you for pointing out the typo in line 165. We have updated the sentence in the revision.
>
> **Clarity 1: Use of acronyms.** We agree that a lighter use of acronyms improves the clarity of the paper. In particular, we have removed the word "OOD-ness," and we refer to the "ground truth" explicitly. The titles of Section 3 -- serving indeed as high-level takeaways -- have also been rephrased, and we add a summarizing sentence to each experiment in the revision.
>
> **Clarity 2: Moving the Related Works section earlier or merging the Limitations and Conclusion sections.** The role of the Related Works section in our paper is to review the conclusions of the experimental results in the light of existing work, making it natural to place it after the experiments. We agree that merging the Limitations and Conclusion sections improves the paper and updated the revision accordingly.
>
> **Clarity 3: Use of rank correlation.** A severe correlation between the epistemic and aleatoric uncertainty estimates prohibits them from capturing semantically different sources of uncertainty. To detect any kind of monotonic relation between the estimates, we report the rank correlation in the main paper. We have also added the Pearson correlation results to Appendix C of the revision and the table below. These mostly agree with the Spearman (rank) correlation results, except for PostNets. This is because the Pearson correlation fails to detect non-linear dependencies (see Fig. 1b of the rebuttal PDF) between the aleatoric and epistemic estimates.
>
> | Method | Spearman (Rank) Correlation | Pearson Correlation |
> |--------|-------------------------|-------------|
> | Laplace | 0.02418 ± 0.0062 | 0.01363 ± 0.0074 |
> | SNGP | 0.93860 ± 0.0016 | 0.81107 ± 0.0016 |
> | GP | 0.94154 ± 0.0014 | 0.83421 ± 0.0012 |
> | HET-XL | 0.97630 ± 0.0024 | 0.91313 ± 0.0024 |
> | MC-Dropout | 0.89813 ± 0.0012 | 0.82608 ± 0.0011 |
> | Shallow Ens. | 0.90814 ± 0.0076 | 0.81058 ± 0.0068 |
> | Deep Ens. | 0.91541 | 0.80855 |
> | HetClassNN | 0.93197 ± 0.0012 | 0.81492 ± 0.0012 |
> | PostNet | 0.90254 ± 0.0060 | 0.57699 ± 0.0070 |
> | HET | 0.97689 ± 0.0031 | 0.90375 ± 0.0031 |
> | EDL | 0.99932 ± 0.00002 | 0.99519 ± 0.00002 |
>
> **Clarity 4: OOD detection evaluation.** Your interpretation is correct: An equal split of ID and OOD samples is constructed, with ID samples getting class 0 and OOD samples getting class 1. In the paper, we report the AUROC of the uncertainty estimator on this binary prediction problem. We have updated the description in the main paper.
>
> **Clarity 5: Conclusion for deep ensembles.** The conclusion is that (i) the information-theoretical decomposition does not lead to disentangled uncertainties, even though (ii) deep ensembles can perform well at both epistemic and aleatoric tasks. The reason why these two conclusions are simultaneously possible is that while Fig. 1 uses the terms of the information-theoretical decomposition, in the practical experiments, we report the best-performing aggregator for each method (Section 2.1.1, last paragraph), which is not necessarily a component of the information-theoretical decomposition.
>
> **Clarity 6: Difference between Fig. 4.b and Fig. 4.a.** The main paper first discusses the ImageNet results and then highlights key differences between the small-scale CIFAR-10 and the large-scale ImageNet datasets. Fig. 4.b, containing results for CIFAR-10, is discussed in Section 3.7.
>
> **Rel. 1: Conformal prediction.** Your understanding is correct. Conformal prediction methods are not applicable to our benchmark on epistemic and aleatoric uncertainty disentanglement because conformal predictions are predictive (total) uncertainty estimators that do not separate the aleatoric and epistemic sources of uncertainty. You are also correct in your understanding that quantile estimators do not apply since they are not defined for classification settings.
>
> **Doc. 1: The documentation of the repository could be improved.** On top of the inline comments and the `README` file, we have updated our repository to include docstrings and clarifying comments for all public functions.
>
> We hope that our rebuttal appropriately addresses your remarks and are looking forward to a fruitful discussion of our work.

---

### Official Review · Reviewer_h1Bx · 2024-07-25
**Interesting paper about uncertainty disentanglement evaluation**

**Rating:** 7
**Confidence:** 5
**Clarity:** Yes, the paper is mostly well written.

**Review:**

This paper is in my opinion medium quality, while there are many results and an interesting benchmark, the main result of uncertainty disentanglement quality is only evaluated in terms of rank correlation, and other experiments only test one kind of uncertainty at a time, which is not completely correct and might miss the causes for disentanglement failure.

The pros are the large number of experiments, large number and selection of uncertainty estimation methods, and some of conclusions are interesting (OOD detection, aleatoric uncertainty estimation, correctness prediction, and relationship between tasks).

The cons are the non-standard use of rank correlation to measure uncertainty disentanglement quality, some unclear benchmark details, and not testing both aleatoric and epistemic uncertainty for all tasks, which would strengthen the paper and its conclusion that disentanglement often fails.

**Strengths:**

- The paper is clear and well written.
- The paper tests an excellent number and an excellent selection of uncertainty methods, and there are some clear conclusions.
- In particular I like the experiments on OOD detection, aleatoric uncertainty quality, and correctness prediction, and the comparison across different uncertainty tasks and metrics.
- I believe the evaluation looks correct, most but not all of the conclusions I believe also to be correct, and some results are significant, in particular the ones about aleatoric uncertainty quality, correctness prediction and uncertainty tasks and metrics.
- Experiments are done in ImageNet and CIFAR10, the large scale ensures conclusions are robust to transfer to real-world settings, and the paper points this out in some sections for specific settings, and also concludes that low scale experiments (CIFAR10) do not transfer to ImageNet, which is a valid conclusion.

After rebuttal, if authors include all changes from discussion (Kendall and Gal formulation evaluation, and evaluating epistemic uncertainty by varying the size of the training set), I change my score to accept.

**Additional Feedback:**

Some minor remarks/suggestions.
- Since rank correlation is presented as main metric for disentangled uncertainty quality, showing some samples/scatterplots of aleatoric and epistemic uncertainty, to see how these correlations look like visually.
- I get the impression that from Sec 3.2, the paper deviates from evaluating uncertainty disentanglement, as its not clear which uncertainty output is being evaluated, as both uncertainties are not evaluated equally, like in Figure 2, where epistemic uncertainty is evaluated using OOD detection in the left figure, but aleatoric uncertainty is evaluated on the right plot against a human label ground truth, I would have expected all experiments to evaluate both kinds of disentangled uncertainties equally, to make arguments on what each kind of uncertainty is really doing.

And two question for rebuttal:
- Have you considered that the monte carlo approximation for the mutual information in the information theoretic formulation is the source of the correlation between aleatoric and epistemic uncertainty?
- Why not evaluate the Gaussian logits formulation by Kendall and Gal?

**Correctness:**

As a benchmark, I believe the evaluation is mostly correct, my only complain is the use of rank correlation to evaluate disentangled uncertainty quality, which is not a standard way to evaluate such uncertainties, and the paper does not prove that this is a good way to make such comparison.

**Documentation:**

The benchmark and evaluation details are mostly documented in the supplementary material. I believe the benchmark details are not completely there, as the benchmark is basically implicitly described in the paper, it would be nice to write a short section to describe the benchmark, for future reproducibility.

**Ethics:**

No ethics issues.

**Limitations:**

The paper has a good limitations section, and I believe the limitations should be pointed out more strongly in the title, as the paper only evaluates classification tasks and mostly the information theoretic formulation, this could be pointed out in the papers' title.

Two additional limitations are that the gaussian logits formulation was not evaluated, and that human soft labels for aleatoric uncertainty, there are no guarantees that a model will follow human intuitions, so it is not a perfect ground truth.

**Opportunities For Improvement:**

- My main criticism of this paper is the use of rank correlation to measure quality of uncertainty disentanglement, the paper argues that uncertainties should not be rank correlated, but this is not a standard way to measure ucertainty disentanglement, and the paper does not demonstrate or prove that this is a good metric for quality in this case. There is agreement in the uncertainty estimation literature that one way to test for epistemic uncertainty is to vary the size of the training set, but this is not done in this paper, only OOD detection samples are used as a way to test epistemic uncertainty estimation.
- My second criticism is selection of uncertainty disentanglement formulation, as the information theoretical formulation is the most evaluated one, followed by the Bregman decomposition, but there is also the Gaussian logits decomposition (by Kendall and Gal) that is completely ignored in this paper.
- The information theoretical formulation is approximate, and I believe this might be a good explanation on why disentangled uncertainties are correlated, and would be very positive to explore this explanation, specially as the mutual information term is only estimated using monte carlo sampling, and the quality of such approximation is critical, it could be the source of correlation between model and data uncertainties that the authors observe.

**Relation To Prior Work:**

Yes, the paper is outstandingly well connected to the literature.

**Summary And Contributions:**

This paper presents a benchmark/evaluation for uncertainty estimation, focusing mostly but not only on uncertainty disentanglement, meaning making separate estimates of model and data uncertainty, but also focuses on other kinds of uncertainty and related tasks.

The benchmark presents some conclusions for the field across many uncertainty estimatin methods, that disentangling uncertainties seem to be rank correlated in most methods, OOD detection is hard, aleatoric uncertainty estimation is hard, and correctness prediction mostly works, all these evaluated on ImageNet and CIFAR10.

Contributions are:
- A benchmark/evaluation for uncertainty estimation, in particular about disentangled uncertainties and other tasks.
- Initial results for many UQ methods and aggregators, with some clear conclusions already drawn.
- Confirmation of previous results on more recent uncertainty estimation methods, in particular OOD detection degrading with corruption severity.

---

> ### Author Rebuttal · Authors · 2024-08-16
>
> Dear Reviewer,
>
> We sincerely appreciate your expert feedback and useful suggestions for improving our work. We address your concerns below and provide a detailed changelog in the general comment.
>
> **Con. 1: Non-standard use of rank correlation to measure uncertainty disentanglement quality.** To the best of our knowledge, we are the first to quantify uncertainty disentanglement instead of solely relying on qualitative visualizations (Mukhoti et al., 2023, Fig. 2; Valdenegro-Toro and Mori, 2022 Figs. 8-10; Kendall and Gal, 2017, Figs. 1,5,6), but if you have pointers to further metrics, we are happy to implement them.
>
> Rank correlation is used to detect any monotonic relationship between the epistemic and aleatoric components, not just linear ones. To motivate this choice of metric, we have added scatter plots of the aleatoric and epistemic estimates to the rebuttal PDF. Fig. 1a is a descriptive scatter plot of most methods, which shows that the epistemic and aleatoric components are highly correlated (most of the time, even linearly -- with PostNet being an outlier in Fig. 1b). In other words, they do not provide semantically different sources of uncertainty as required and are not disentangled. The rank correlation metric successfully captures the dependencies between the estimates. We have also added results for the Pearson correlation below, which agrees with these findings (except it fails to detect the non-linear relation for PostNet). Further, we have added a dedicated "Design Choices" section to the appendix of the revised paper that motivates the choice of rank correlation.
>
> | Method | Spearman (Rank) Correlation | Pearson Correlation |
> |--------|-------------------------|-------------|
> | Laplace | 0.02418 ± 0.0062 | 0.01363 ± 0.0074 |
> | SNGP | 0.93860 ± 0.0016 | 0.81107 ± 0.0016 |
> | GP | 0.94154 ± 0.0014 | 0.83421 ± 0.0012 |
> | HET-XL | 0.97630 ± 0.0024 | 0.91313 ± 0.0024 |
> | MC-Dropout | 0.89813 ± 0.0012 | 0.82608 ± 0.0011 |
> | Shallow Ens. | 0.90814 ± 0.0076 | 0.81058 ± 0.0068 |
> | Deep Ens. | 0.91541 | 0.80855 |
> | HetClassNN | 0.93197 ± 0.0012 | 0.81492 ± 0.0012 |
> | PostNet | 0.90254 ± 0.0060 | 0.57699 ± 0.0070 |
> | HET | 0.97689 ± 0.0031 | 0.90375 ± 0.0031 |
> | EDL | 0.99932 ± 0.00002 | 0.99519 ± 0.00002 |
>
> **Con. 2: Unclear documentation of benchmark details.** In addition to the benchmark details in Appendix I, we have also added a "Design Choices" section to the appendix that explains the rationale behind our evaluation (Con. 1). For future reproducibility, we have added additional docstrings and descriptions on top of the existing `README` file and inline comments in our code appendix.
>
> **Con. 3: Not testing both aleatoric and epistemic uncertainty for all tasks.**
>
> We compute the information-theoretical aleatoric and epistemic estimators, along with eight further aggregators (Appendix E), in all experiments. Outside the disentanglement experiments, we chose to always report only the best-performing aggregator because all ten aggregators would add complexity for the reader (see Section 2.1.1, last paragraph). We agree that the epistemic and aleatoric aggregators from the information-theoretical decomposition are of particular interest, so we share their results in the rebuttal PDF. In Fig. 2 of the rebuttal PDF, we cross-evaluate the epistemic and aleatoric terms of the information-theoretical decomposition on the opposite task, which leads to two conclusions. (i) The epistemic estimates perform notably well on aleatoric uncertainty evaluation, which contradicts the common claim that epistemic estimates are not useful in-distribution. (ii) The aleatoric estimates are almost always better than the epistemic ones on OOD detection, going against the wide belief that aleatoric estimates are not to be trusted OOD. This is another consequence of the entanglement that we uncover in our paper. We have added these insights to the disentanglement evaluation of the revised paper.
>
> **W1: Testing for epistemic uncertainty by varying the size of the training set.** Inspired by your remarks, we have added experimental results on {$10\%, 50\%, 100\%$} of the training data in Fig. 3 and Table 1 of the rebuttal PDF following Kendall and Gal (2017, Table 3b). We exchanged the 25\% fraction used by Kendall and Gal (2017) with a 10\% fraction to induce more epistemic uncertainty in the large-scale ImageNet dataset. Table 1 shows that for the majority of the benchmarked methods, both the mean epistemic and aleatoric uncertainties decrease monotonically with an increasing training dataset size, which is in line with the findings of Kang et al. (2024; ICLR). Fig. 3 visualizes this tendency for the EDL method but also highlights the extreme correlation (0.9993) between the aleatoric and epistemic estimates.
>
> **W2: The Gaussian logits decomposition by Kendall and Gal is ignored in the paper.** Following your valuable remark, we have implemented and evaluated the method of Kendall and Gal (2017; NeurIPS), increasing the number of benchmarked methods to eighteen. Their method (abbreviated as HetClassNN), coupled with their proposed aleatoric and epistemic estimators, shows a similarly severe rank correlation (0.9419). We get very similar results to the information-theoretical evaluation if we change the architecture of Kendall and Gal (2017) to any of the ten other distributional methods (but use their epistemic and aleatoric estimators), as shown in the table below.
>
> | Method | Kendall and Gal Rank Correlation |
> |--------|-------|
> | MC-Dropout | 0.8013 |
> | GP | 0.8294 |
> | Shallow Ens. | 0.8353 |
> | SNGP | 0.8576 |
> | Deep Ens. | 0.8732 |
> | Laplace | 0.001
> | HET | 0.9336 |
> | HET-XL | 0.9366 |
> | HetClassNN | 0.9419 |
> | PostNet | 0.9801 |
> | EDL | 0.9932 |
>
> In Fig. 2 and Table 1 of the rebuttal PDF, we also evaluate HetClassNN using the estimators of the information-theoretical decomposition. HetClassNN is on par with deep ensembles in capturing aleatoric uncertainty.

---

> > ### Author Rebuttal · Authors · 2024-08-16
> >
> > **W3: The information-theoretical formulation is approximate.** For the Evidential Deep Learning methods (EDL, PostNet), the terms of the information-theoretical formulation are evaluated in closed form. For all methods except MC-Dropout, our evaluation used 1000 Monte Carlo samples. As MC-Dropout requires one full forward pass for each sample, we used 100 samples. To observe the effect of Monte Carlo sampling on disentanglement, we evaluated the GP method with {$10, 100, 100, 10000$} samples. The results below show that the number of MC samples does not have a large effect on the rank correlation, and sampling more makes the correlation higher (as it reduces the independent noise).
> >
> > | Number of MC samples | Rank Correlation Between IT Aleatoric & Epistemic |
> > |----------------------|---------------------------------------------------|
> > | 10                   | 0.93806                                           |
> > | 100                  | 0.96407                                           |
> > | 1000                 | 0.96702                                           |
> > | 10000                | 0.96723                                           |
> >
> > **L1: The Gaussian logits formulation is not evaluated.** We have included the Gaussian logits formulation in W2 and the rebuttal PDF (Fig. 2 and Table 1). We also added the results to the revision. Please refer to W2 for our conclusions.
> >
> > **L2: There are no guarantees that a model will follow human intuitions.** Aleatoric uncertainty is a model-independent quantity that only depends on the generative process of the data. As human annotators are a part of this generative process (providing $p(y \mid x)$), the Bayes-optimal prediction is to perfectly align with the (normalized) annotator votes. While we agree that models might not follow human intuitions, we believe that the annotator label distributions are a ground-truth source of aleatoric uncertainty that is meaningful to compare against.
> >
> > **Corr. 1: The paper does not prove that the rank correlation is a good way to evaluate uncertainty disentanglement.** The "Design Choices" section of our revision and Con. 1 of our rebuttal discusses our choice of rank correlation in detail.
> >
> > **Doc. 1: It would be nice to write a short section to describe the benchmark, for future reproducibility.** We agree that such an explicit section improves reproducibility and have therefore included it in the "Design Choices" section of our revision.
> >
> > **Add. 1: Showing some scatter plots of aleatoric and epistemic uncertainty.** As referenced in Con. 1, we have added these scatter plots to Fig. 1 of the rebuttal PDF.
> >
> > **Add. 2: It is not clear which uncertainty output is being evaluated, as the uncertainties are not evaluated equally.** As discussed in Con. 3, in the practical experiments, we have added the cross-evaluation of the aleatoric and epistemic estimates to Fig. 2 of the rebuttal PDF and the disentanglement evaluation of the revision.
> >
> > **Q1: "Have you considered that the monte carlo approximation for the mutual information in the information theoretic formulation is the source of the correlation between aleatoric and epistemic uncertainty?"** We have used closed-form expressions for the PostNet and EDL methods and tested the effect of the Monte Carlo approximation's stochasticity for the others, which we found to be mild. Please refer to W3 for a detailed discussion.
> >
> > **Q2: "Why not evaluate the Gaussian logits formulation by Kendall and Gal?"** We have included this formulation in our revision and the rebuttal PDF. Please refer to W2 for the details.
> >
> > The reviewer's suggestions have greatly enhanced the paper's quality. We hope that our new experiments and explanations address the reviewer's concerns about our benchmark.

---

> > > ### Author Response · Authors · 2024-08-27
> > >
> > > Dear Reviewer h1Bx,
> > >
> > > The discussion period ends this week.
> > >
> > > We have added your requested experiments to the rebuttal (Gaussian logit decomposition, reduced dataset sizes, Pearson correlations, scatterplots), which we believe have significantly enriched our benchmark.
> > >
> > > We would greatly appreciate it if you could let us know whether the addition of these experiments addresses your concerns.
> > >
> > > Best wishes,
> > >
> > > The Authors

---

> > ### Comment · Reviewer_h1Bx · 2024-08-31
> >
> > Thanks for the detailed remarks and rebuttal, this looks good, I will increase my score accordingly.
> >
> > My only suggestion now is to do the variation of training set size for all uncertainty methods, as this is a recognized way to test epistemic uncertainty, and to test variation of training set also for the Kendall and Gal formulation.

---

> > > ### Author Response · Authors · 2024-09-01
> > >
> > > Dear Reviewer,
> > >
> > > We are grateful for your recognition of our work and the score increase. Below, we show the mean epistemic uncertainty estimates from the Kendall & Gall formulation when trained on {10%, 50%, 100%} of ImageNet. Methods whose epistemic uncertainties consistently decrease with an increasing dataset size are boldfaced. We will include these results in the revision, as well as the evaluation of all methods (including deterministic ones) on varying dataset sizes.
> > >
> > > | Training Data | Dropout  | EDL          | GP           | HET      | HET-XL   | Laplace  | PostNet      | Shallow  | SNGP         | HetClassNN   |
> > > | ------------- | -------- | ------------ | ------------ | -------- | -------- | -------- | ------------ | -------- | ------------ | ------------ |
> > > | 10%           | 1.327e-4 | **3.948e-7** | **6.928e-4** | 5.179e-4 | 4.159e-4 | 1.391e-4 | **8.695e-9** | 2.811e-4 | **6.977e-4** | **6.236e-5** |
> > > | 50%           | 9.493e-5 | **2.937e-7** | **2.322e-4** | 1.115e-4 | 1.049e-4 | 1.253e-4 | **7.731e-9** | 1.804e-4 | **2.311e-4** | **5.355e-5** |
> > > | 100%          | 8.159e-5 | **2.718e-7** | **1.145e-4** | 1.215e-4 | 1.079e-4 | 1.404e-4 | **5.585e-9** | 1.994e-4 | **1.342e-4** | **4.563e-5** |

---

### Official Review · Reviewer_hBr9 · 2024-08-06
**first comprehensive benchmark for studying uncertainty disentanglement**

**Rating:** 7
**Confidence:** 4
**Correctness:** Experiment design and analysis of the…
**Clarity:** The paper is well structured and is e…

**Review:**

The work is original and has a clear focus on uncertainty disentanglement.
Contributed benchmark including variety of tasks is useful for reproducibility and speeding the progress in this important direction.
This is the most comprehensive experimental study on this topic I have seen so far.
The paper is well-written

**Strengths:**

1) First comprehensive benchmark in this area.
2) Impressive number of considered approaches and uncertainty tasks; a OOD detection, aleatoric uncertainty estimation, and correctness prediction.  Relationships between tasks have been also analysed.
3) Useful insights from the extensive experimental study.

**Additional Feedback:**

n/a

**Documentation:**

The descriptions are clear and provide enough detail. I am satisfied.

**Limitations:**

nothing, besides the mentioned opportunities for improvement.

**Opportunities For Improvement:**

1) While the benchmarking study included many approaches and a diverse range of uncertainty tasks, I wonder whether other datasets but ImageNet and CIFAR10 should be included. If so, should those be considered as separate benchmarks of part of the current suit. How difficult would it be to expand the current dataset(s) x tasks?
2) Design choices can be motivated more meticulously, in particular the choice of rank correlation to assess the quality of uncertainty disentanglement.
3) While the paper is well-written overall. The results and conclusions we can or cannot derive from them could be described in more detail.

**Relation To Prior Work:**

I enjoyed reading literature analysis. It is clear, insightful, and complete. I am not aware of other other that would be critical to include.

**Summary And Contributions:**

The paper presents a comprehensive benchmarking of uncertainty disentanglement across a diverse range of uncertainty tasks.
Modeling uncertainty is a highly relevant and challenging topic. Contributed benchmark is an important milestone for pushing this area of research further.
The results obtained from the extensive experimental study are insightful for both practitioners and researchers interested in understanding SOTA and finding directions for future work.

---

> ### Author Rebuttal · Authors · 2024-08-16
>
> Dear Reviewer,
>
> We express our sincere gratitude for acknowledging our work and for your valuable feedback.
>
> **W1: Other datasets.** Our benchmark suite is easy to extend to more datasets due to its modular structure. We currently benchmark on ImageNet-ReaL and CIFAR-10H since these are the only larger-scale datasets with multiple annotations per image, which are used for the aleatoric uncertainty evaluation. Once more such datasets become available, our proposed benchmark protocol can be directly applied to them. Other task-specific evaluation criteria are also simple to add in the `evaluate.py` file. We hope that the extensive set of evaluation criteria and benchmark tasks we provide lead to a more thorough and standardized evaluation of uncertainty quantification methods.
>
> **W2: Motivation of design choices, particularly the choice of rank correlation.** When epistemic and aleatoric uncertainty estimates are severely correlated, they cannot measure semantically different sources of uncertainty. The Spearman (rank) correlation can detect arbitrary monotonic dependencies between two random variables and is thus a suitable metric to evaluate the correlation of aleatoric and epistemic estimates. Inspired by your suggestion, we have added a "Design Choices" section to the appendix that motivates our choice of evaluation criteria and experimental setup. We also report the Pearson correlations in Appendix C of the revision and the table below, which are in accordance with the Spearman correlations (except for PostNet, where the relationship is highly monotonic but non-linear; see Fig. 1b in the rebuttal PDF).
>
> | Method | Spearman (Rank) Correlation | Pearson Correlation |
> |--------|-------------------------|-------------|
> | Laplace | 0.02418 ± 0.0062 | 0.01363 ± 0.0074 |
> | SNGP | 0.93860 ± 0.0016 | 0.81107 ± 0.0016 |
> | GP | 0.94154 ± 0.0014 | 0.83421 ± 0.0012 |
> | HET-XL | 0.97630 ± 0.0024 | 0.91313 ± 0.0024 |
> | MC-Dropout | 0.89813 ± 0.0012 | 0.82608 ± 0.0011 |
> | Shallow Ens. | 0.90814 ± 0.0076 | 0.81058 ± 0.0068 |
> | Deep Ens. | 0.91541 | 0.80855 |
> | HetClassNN | 0.93197 ± 0.0012 | 0.81492 ± 0.0012 |
> | PostNet | 0.90254 ± 0.0060 | 0.57699 ± 0.0070 |
> | HET | 0.97689 ± 0.0031 | 0.90375 ± 0.0031 |
> | EDL | 0.99932 ± 0.00002 | 0.99519 ± 0.00002 |
>
> **W3: Description of results and conclusions.** To strengthen the paper further, in addition to the section titles that highlight high-level takeaways from each section, we have added a summarizing sentence to each benchmark type.
>
> We hope our additional experiments and implemented changes positively influence your evaluation.

---

### Author Rebuttal · Authors · 2024-08-16

Dear Reviewers,

We would like to thank you for your time volunteered for reviewing. We are delighted about your positive response, in which you all agree that our benchmark _tests an excellent number and an excellent selection of uncertainty methods_ (h1Bx, hBr9, B7nH), making it _the most comprehensive experimental study on this topic I have seen_ (hBr9).
We are happy that you find the novel problem of uncertainty disentanglement _a strength of this paper_ (B7nH, hBr7) and that we connect it _outstandingly well to the literature_ (h1Bx) so that you _enjoyed reading the literature analysis. It is clear, insightful, and complete_ (hBr9).

We are determined to constitute this benchmark as the all-embracing basis for developing disentangled uncertainties in the coming years. Thus, we happily implemented your requested experiments. In summary, we

- Reimplemented and benchmarked a further method (Gaussian logit decomposition) on top of our existing seventeen methods
- Added experiments that test epistemic uncertainty by varying the training dataset size
- Added Pearson correlation results and scatter plots for all disentanglement methods, which agree with our findings via the Spearman rank correlation
- Added explanations both in the main text and in a dedicated "Design Choices" appendix
- Added docstrings and descriptions to the existing inline comments and `README` file of our open-source code base so that it is an easy-to-extend fundament for developing disentangled uncertainties.

We attach a PDF with the additional results and discuss them in your individual comments.

---

> ### Author Response · Authors · 2024-09-01
>
> Dear Reviewers,
>
> We would like to thank you for your valuable suggestions once again. We are happy about the score increase of reviewer h1Bx and the agreeing acceptance scores. In total, we
> - Added new methods: HetClassNN (Kendall & Gall, 2017), SWAG (Maddox et al., 2019), and Laplace with higher precision;
> - Refactored the code base to be a modular extension of PyTorch Image Models (`timm`) to support all `timm` models that may be added in the future;
> - Added experiments that test the epistemic uncertainty of all methods and decompositions by varying the training dataset size;
> - Added Pearson correlation results and scatter plots for all disentanglement methods, which agree with our findings via the Spearman rank correlation;
> - Added further explanations both in the main text and in a dedicated "Design Choices" appendix;
> - Added further docstrings and descriptions to our open-source code base and its `README` file so that it is an easy-to-extend foundation for developing disentangled uncertainties.
>
> We hope that our benchmark suite standardizes and accelerates uncertainty disentanglement research.

---

### Decision · Program_Chairs · 2024-09-26

**Decision:**

Accept (Spotlight)

**Comment:**

The authors give the first benchmark for uncertainty disentanglement. Specifically, they evaluate 17 recent uncertainty quantification methods on two different datasets, showing whether they can give disentangled uncertainty estimates (as well as other uncertainty analyses). The authors give a number of insights based on their experiments.

The reviews were uniformly positive, specifically mentioning the following:
- The paper gives the first comprehensive benchmark for uncertainty disentanglement.
- The experimental section is very thorough, with many uncertainty quantification approaches, as well as quantifying different types of uncertainty predictions.
- The authors already show a number of interesting findings and analysis.
- Additionally, the authors release their code and give a thorough limitations section.

After the initial reviews, the authors made a significant update in order to address the reviews. This included experimental extensions, such as adding an 18th method and adding Pearson correlation, as well as clarity improvements, such as adding a “design choices” section.

After reading through the reviews and rebuttals, I agree with the reviewers about the high quality and significance of this work, especially after the authors’ updates.  I recommend acceptance.